# ECHOES OF BERT: DO MODERN LANGUAGE MODELS REDISCOVER THE CLASSICAL NLP PIPELINE?

## ABSTRACT

Large transformer-based language models dominate modern NLP, yet our understanding of how they encode linguistic information relies primarily on studies of early models like `BERT` and `GPT-2`. Building on prior BERTology work, we analyze 25 models spanning classical architectures (`BERT`, `DeBERTa`, `GPT-2`) to modern large language models (`Pythia`, `OLMo-2`, `Gemma-2`, `Qwen2.5`, `Llama-3.1`), probing layer-by-layer representations across eight linguistic tasks in English. Consistent with earlier findings, we find that hierarchical organization persists in modern models: early layers capture syntax, middle layers handle semantics and entity-level information, and later layers encode discourse phenomena. However, larger models compress this entire hierarchy toward earlier layer positions, suggesting they build richer representations more quickly. We dive deeper, conducting an in-depth multilingual analysis of two linguistic properties - lemma identity and inflectional features - that help disentangle form from meaning. We find that lemma information concentrates linearly in early layers but becomes increasingly nonlinear deeper in the network, while inflectional information remains linearly accessible throughout all layers. Additional analyses of attention mechanisms, steering vectors, and pretraining checkpoints reveal where this information resides within layers, how it can be functionally manipulated, and how representations evolve during pretraining. Taken together, our findings suggest that, even with substantial advances in LLM technologies, transformer models learn to organize linguistic information in similar ways, regardless of model architecture, size, or training regime, indicating that these properties are important for next token prediction.

## 1 INTRODUCTION

Large transformer-based language models (LMs) are widely used for tasks such as text generation, question answering, and code completion (Workshop, 2023; Groeneveld et al., 2024; Llama, 2024; Hui et al., 2024) However, how these models internally represent linguistic information remains an active research area. Prior work suggests a hierarchical organization where different layers specialize in capturing distinct levels of linguistic structure, from surface features to syntax and semantics (Jawahar et al., 2019; Tenney et al., 2019a; Rogers et al., 2020).

But these studies focus only on first-generation LMs such as `BERT` and `GPT-2` (Devlin et al., 2019; Radford et al., 2019). Since then, language technology has transformed dramatically - today's models differ in architecture (encoder-only, decoder-only, encoder–decoder), pretraining objectives (masked vs. causal language modeling), training data volume (billions vs. trillions of tokens), and post-training adaptation. (Brown et al., 2020; Groeneveld et al., 2024; Lambert et al., 2025). We ask: do modern LMS *rediscover the classical NLP pipeline* observed in early models, and how does model scale and architecture influence where and how linguistic structure is encoded?

To answer these questions we systematically probe 25 pretrained models ranging from `BERT Base` to `Llama-3.1 8B`, spanning multiple architectures, sizes, and training regimes. We train simple classifiers at each layer to predict eight linguistic tasks in English and evaluate where information emerges.

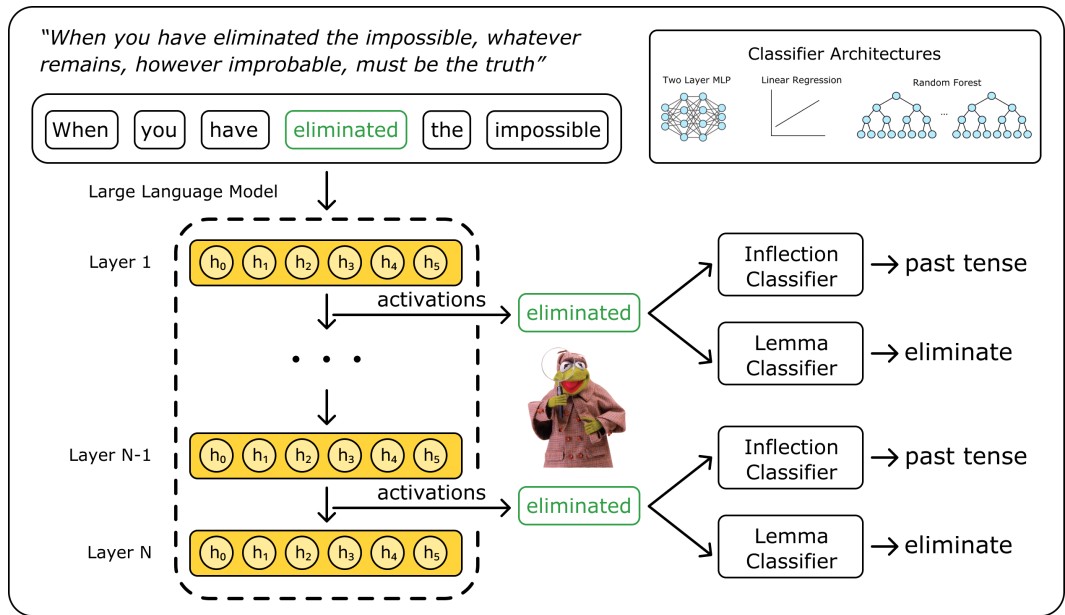

Figure 1: Overview of our probing methodology. We extract hidden state activations from each model layer for target words and train simple linear and shallow non-linear classifiers for token, span and pairwise edge predictions (POS, dependencies, constituents, NER, SRL, SPR, coreference, and relations), as well as word-level lemma and inflection prediction. We compute selectivity using control labels and summarize where performance emerges with expected layer and center of gravity.

Beyond this pipeline analysis, we perform a targeted case study on two linguistic properties: *lemma identity* and *inflectional features*. These properties help disentangle meaning from surface form - consider the words *walk*, *walked*, *jump*, and *jumped*. Do language models group words by shared meaning (*walk*, *walked*) or by shared grammar (*walked*, *jumped*)? More broadly, where and how do LMs encode a word's lemma and its inflectional features?

We examine six typologically diverse languages - English, Chinese, German, French, Russian, and Turkish - to test whether observed patterns generalize beyond English. We also test where lemma and inflectional information resides (attention heads vs. residual streams), track when these representations emerge during pretraining, and evaluate the impact of editing activations via steering vectors. We find that:

1. Modern LMs rediscover the classical NLP pipeline. Syntactic tasks peak earliest, semantic tasks peak in the middle, and discourse tasks peak latest. Larger models compress this pipeline towards shallower layers, suggesting they learn richer representations more quickly.

2. Lemma information is encoded prominently in early layers and becomes increasingly non-linear deeper in the network, whereas inflectional information remains linearly accessible across all layers and languages.

3. Lemma and inflectional information emerge early in pretraining and reside primarily in the residual stream; inflectional features occupy compact, steerable subspaces that enable effective interventions.

## 2  PROBE DESIGN AND METRICS

We investigate how language models encode linguistic information using simple classifiers (*probes*) trained on activations from individual layers. Following Tenney et al. (2019b), we consider three types of predictions: token-level tasks (*e.g.*, , POS), span-level tasks (constituency, named entity recognition, semantic role labeling, semantic proto-roles), and edge or pairwise tasks (dependency arcs and coreference links). For our case study we additionally train probes to predict each word's lemma and its inflectional features.

## 2.1 PROBE ARCHITECTURES

For each layer of a model we extract residual-stream representations for a target word, span or pair and train two simple classifiers: a linear regression probe and non-linear multi-layer perceptron (MLP) probe. The linear probe measures how well information is linearly separable in the representation space, while the non-linear probe tests whether a non-linear decision boundary yields better performance. Comparing these probes allows us to infer whether a property is encoded *linearly* or *non-linearly*. Architecture details and hyperparameters are provided in Appendix C.

## 2.2 REPRESENTATIONS AND TASKS

For token-level tasks we use the representation of the last subword token for the target word; for span-level tasks we mean-pool representations across subwords; for pairwise tasks we concatenate and element-wise combine representations following Tenney et al. (2019b).

We evaluate eight linguistic tasks introduced by Tenney et al. (2019a), covering the classical NLP pipeline from syntax to discourse. At the syntactic level, we consider part-of-speech tagging, constituency parsing (phrase structure), and dependency parsing (head–dependent relations); at the semantic level, named entity recognition (persons, organizations, locations), semantic role labeling (agent and patient roles), and semantic proto-role labeling (*e.g.*, , volition, sentience); and at the discourse level, coreference resolution and relation extraction (relations between entities). Formal task definitions are provided in Appendix D.

## 2.3 METRICS

We define several metrics for localizing where information emerges with depth and for quantifying nonlinearity: selectivity, the linear separability gap, and two depth statistics inspired by Tenney et al. (2019a), expected layer and center of gravity.

**Selectivity.** Probes may simply memorize training data rather than extracting true linguistic information from the representations. To account for this, we train identical probes on randomly permuted labels (control tasks) following Hewitt & Liang (2019). We define selectivity at layer $\ell$ as the difference between real and control accuracies:

$$\text{Sel}_\ell = \text{Acc}_\ell^{\text{real}} - \text{Acc}_\ell^{\text{control}} \tag{1}$$

Higher values mean the classifier is extracting true linguistic information rather than memorizing.

**Linear separability gap.** We quantify nonlinearity at a layer as the difference in accuracy between a non-linear and linear probe:

$$\text{Gap}_\ell = \text{Acc}_\ell^{\text{nonlin}} - \text{Acc}_\ell^{\text{linear}}, \tag{2}$$

where positive values indicate useful information is present but not linearly separable.

**Center of gravity and expected layer.** Let $a_\ell$ be the test accuracy using layer $\ell$ for $\ell = 0, \dots, L$, and let $b_\ell = \max_{j \le \ell} a_j$ be the cumulative (best-so-far) curve. We weight layers by their consolidation relative to the baseline and take an index-weighted average:

$$w_\ell = \frac{b_\ell - b_0}{\sum_{k=0}^{L}(b_k - b_0)}, \quad \text{CenterOfGravity} = \sum_{\ell=0}^{L} \ell \, w_\ell. \tag{3}$$

Then, to localize where marginal gains first occur, we use the nonnegative increments of the cumulative curve and take their weighted average:

$$\Delta_\ell = \max(b_\ell - b_{\ell-1}, 0) \,, \quad p_\ell = \frac{\Delta_\ell}{\sum_{j=1}^{L} \Delta_j}, \quad \text{ExpectedLayer} = \sum_{\ell=1}^{L} \ell \, p_\ell. \tag{4}$$

Unlike center of gravity (which weights consolidated performance), this emphasizes where useful information first becomes available, highlighting the specific layers at which the model begins to encode properties relevant to the task.

## 3 EXPERIMENTS

Using the methodology introduced in Section §2, we describe the components of our experimental setup: the datasets, model suite, and procedure for extracting token-level representations.

### 3.1 DATASETS

We use several annotated datasets for our eight classical NLP pipeline tasks: UD English-GUM (POS, dependencies, named entities, coreference, constituents) (Nivre et al., 2016; Zeldes, 2017), Universal Propositions English-EWT (SRL) (Jindal et al., 2022), SPR1 datasets (PropBank and UD-EWT sources; SPR), and SemEval-2010 Task 8 (relations). We use the same token/span/edge labeling schemes.

For our in-depth analysis of lemma identity and inflectional features, we use Universal Dependencies corpora across six languages - English, Chinese, German, French, Russian, Turkish (Nivre et al., 2016). We select GUM for English (Zeldes, 2017), GSD for Chinese/German/French (McDonald et al., 2013; Guillaume et al., 2019), SynTagRus for Russian (Droganova et al., 2018), and IMST for Turkish (Sulubacak et al., 2016). [1]

### 3.2 MODELS

We study a diverse set of pretrained transformer language models spanning different architectures, sizes, and training regimes. Table 1 lists all models used in this study (see Table 16 for the HuggingFace identifiers).

For English, we evaluate all models listed in Table 1 (excluding the non-English Goldfish models). For the five non-English languages (Chinese, German, French, Russian, Turkish), we focus on models that have explicit multilingual training: the Goldfish monolingual models trained specifically for each target language (Chang et al., 2024), multilingual Qwen2.5 variants that include these languages in their training data, and the multilingual mT5-base model (Xue et al., 2021). This ensures that we evaluate models on languages they were trained on while maintaining sufficient coverage.

Table 1: Overview of models used in this study.

| Model | Parameters | Pretraining Data | Layers |
|---|---|---|---|
| **Encoder-only** | | | |
| BERT Base | 110M | 12.6B tokens[1] | 12 |
| BERT Large | 340M | 12.6B tokens[1] | 24 |
| DeBERTa V3 Large | 418M | 32B tokens[1] | 24 |
| **Decoder-only** | | | |
| GPT-2-Small | 124M | 8B tokens[1] | 12 |
| GPT-2-Large | 708M | 8B tokens[1] | 36 |
| GPT-2-XL | 1.5B | 8B tokens[1] | 48 |
| Goldfish English 1000mb | 124M | 200M tokens | 12 |
| Goldfish Chinese 1000mb | 124M | 200M tokens | 12 |
| Goldfish German 1000mb | 124M | 200M tokens | 12 |
| Goldfish French 1000mb | 124M | 200M tokens | 12 |
| Goldfish Russian 1000mb | 124M | 200M tokens | 12 |
| Goldfish Turkish 1000mb | 124M | 200M tokens | 12 |
| Pythia-6.9B | 6900M | 300B tokens | 32 |
| Pythia-6.9B Tulu | 6900M | 300B tokens | 32 |
| OLMo-2-7B | 7300M | 4T tokens | 32 |
| OLMo-2-7B-Instruct | 7300M | 4T tokens | 32 |
| Gemma-2-2B | 2610M | 2T tokens | 26 |
| Gemma-2-2B-Instruct | 2610M | 2T tokens | 26 |
| Qwen2.5-1.5B | 1540M | 18T tokens | 28 |
| Qwen2.5-1.5B-Instruct | 1540M | 18T tokens | 28 |
| Qwen2.5-7B | 7620M | 18T tokens | 28 |
| Qwen2.5-7B-Instruct | 7620M | 18T tokens | 28 |
| Llama-3.1-8B | 8000M | 15T tokens | 32 |
| Llama-3.1-8B-Instruct | 8000M | 15T tokens | 32 |
| **Encoder-Decoder** | | | |
| mT5-base | 580M | 1T tokens | 12 |

[1] Converted from GB to tokens using the approximation that 1GB of data is approximately 200M tokens in English (Chang et al., 2024).

### 3.3 REPRESENTATION EXTRACTION

We tokenize inputs with model-specific tokenizers and run a forward pass to collect residual-stream activations from every layer. Token, span, and pair encodings follow Section §2. For words split into multiple subwords, we use the last subword's representation (Devlin et al., 2019). Models are used as-is (no fine-tuning), and we report results by layer using these activations.

---

[1]See Appendix §H for complete details including dataset statistics, tokenization information, and visualizations for all languages

# 4 THE CLASSICAL NLP PIPELINE

We probe 18 models across eight linguistic tasks to test whether modern language models rediscover the classical NLP pipeline. In this section, we present three representative models - encoder-only, decoder-only and instruction-tuned architectures - with full results for all models in Appendix §E.

## 4.1 LAYERWISE PATTERNS CLEANLY SEPARATE MODEL FAMILIES

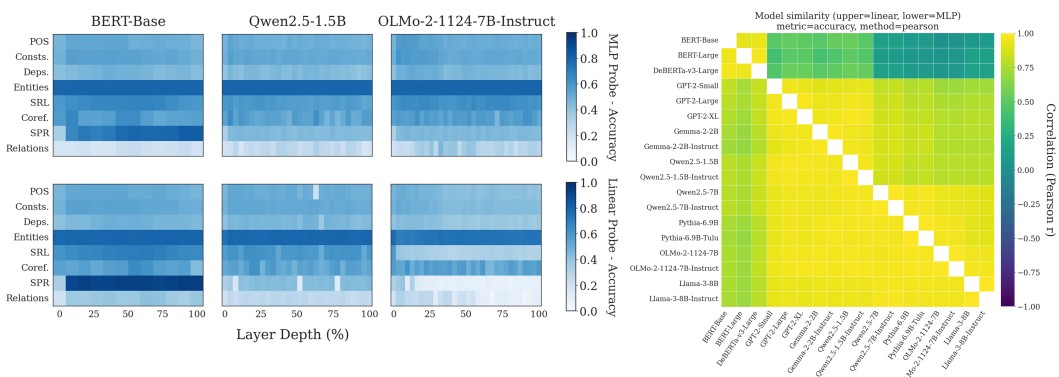

Figure 2: **Left:** Probe accuracy across layers for `BERT Base`, `Qwen2.5-1.5B`, and `OLMo-2-1124-7B-Instruct`. Top panels show MLP probes and bottom panel show linear probes. **Right:** Pearson correlations between all models, computed from flattening each model's task-by-layer accuracy grid and correlating across all pairs of models. Lower triangle: MLP correlations; upper triangle: linear correlations

**Probe performance.** Our results in Figure 2 (left) show that MLP probes consistently match or exceed linear probe accuracy across all tasks (see Figures 7 and 8 for complete results). The linear separability gap - the difference between MLP and linear performance - peaks for late-pipeline tasks, specifically SPR and Relations. This pattern holds across all 18 models (see Appendix §E).

**Model correlations.** The correlation matrix, Figure 2 (right), provides a global summary between all 18 models. A high correlation indicates that two models' layerwise accuracies across tasks are similar; low correlations indicate divergent accuracy patterns. We observe three distinct trends:

1. *Models cluster by architecture.* Encoder-only models (*e.g.*, `BERT` and `DeBERTa`) correlate strongly with each other while having low correlations with decoder models. The same is true for decoder-only architectures, such as `GPT-2`, `Pythia`, `Qwen2.5` and `Llama 3.1`, which form their own cluster with high internal similarity.

2. *Instruction tuning preserves base model latent structure.* Fine-tuned variants maintain high correlations with their base counterparts, indicating that post-training does not fundamentally reorganize linguistic representations.

3. *Model size forms a secondary clustering, but only for linear probes.* Models around one billion parameters group together separately from 7B+ models for linear probe accuracy. MLP probes don't show this size-based clustering, likely because their additional capacity masks any scale-dependent representation differences.

## 4.2 LARGER MODEL COMPRESS THE HIERARCHY

To pinpoint where linguistic properties emerge and consolidate, we compute compute expected layer and center of gravity as defined in Section 2. Intuitively, the *expected layer* captures marginal accuracy gains and highlights the depth at which information first emerges, while *center of gravity* weights each layer by cumulative best accuracy to locate performance ultimately consolidates most strongly.

**The hierarchy persists.** Figure 3 shows a shared relative ordering partially emerges across all models. Syntactic tasks (POS, Constituency, Dependencies) tend to emerge before semantic tasks

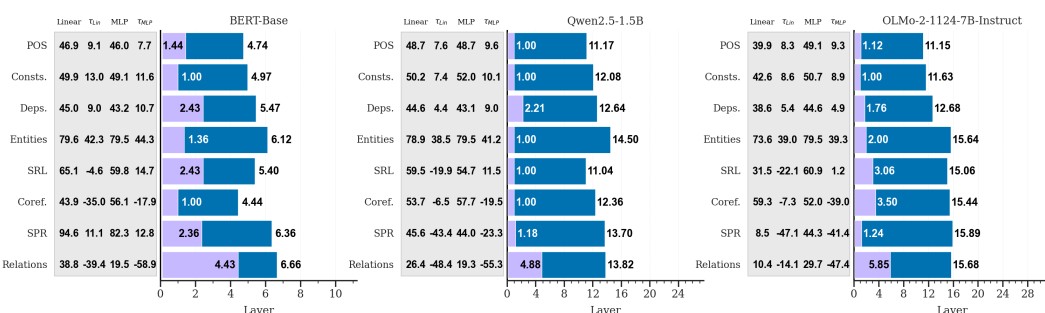

Figure 3: Expected layer (blue) and center of gravity (purple) for the same three models. The left four columns show accuracy and selectivity ($\tau$) for linear and MLP probes, averaged across layers. Selectivity measures how much of the accuracy is due to genuine signal rather than memorization by the probe.

(Entities, SRL, SPR), which emerge before discourse phenomena (Corefernce, Relations). However, this hierarchical progression is less distinct in modern models than in early ones, suggesting that the hierarchy exists but is compressed.

**Scale compresses depth.** Model capacity determines *where* and *whether* this hierarchy forms. For example, `BERT Base` (12 layers) places relation extraction around layer 8, while both `Qwen2.5-1.5B` (28 layers) and `OLMo-2-7B-Instruct` (32 layers) compress it to approximately one-fifth depth. Larger models seem to encode the complete linguistic hierarchy using fewer layers, suggesting that they build useful representations earlier.

**Selectivity reveals probe limitations.** MLP probes appear to achieve high accuracy, but have strong negative selectivity, meaning they memorize the task rather than extract meaningful information from the representations. Linear probes are better, showing positive selectivity for syntactic tasks. However, they drop to near zero selectivity for discourse tasks (Coreference, Relations), suggesting that while discourse information exists in representations, linear decoding struggles to extract it cleanly.

### 4.3 DISCUSSION

Our analysis establishes two key findings:

1. The hierarchical organization observed in early transformers survives in modern models but with less separation between levels. But this relative ordering is detectable across architectures (encoder, decoder, encoder–decoder), training regimes (causal and masked language modeling, instruction tuning), and scale (100M to 8B parameters), but boundaries blur as models compress the pipeline.

2. Modern models encode all linguistic levels at shallower depths. Where `BERT Base` clearly separated syntactic, semantic, and discourse processing across its layers, a 7B model (`OLMo-2-7B-Instruct`) compresses this entire hierarchy into its early layers. This compression is evidence that as models become more powerful, they need fewer layers to learn this hierarchical linguistic structure, perhaps because they have higher representational capacity per layer and benefit from more extensive training.

These results suggest that while the classical NLP pipeline represented how early transformers organized knowledge, modern models develop a more compressed, interleaved representation of linguistic structure.

## 5 LEMMA IDENTITY AND INFLECTIONAL FEATURES

We now examine two important token-level properties: lemma identity and inflectional features. Using the same probing framework from Section §4, we expand to six typologically diverse languages: English, Chinese, German, French, Russian, and Turkish. We investigate where these properties emerge in model representations and how they become linearly accessible across layers.

## 5.1 RESULTS

We report layer-wise accuracies for lemma and inflection prediction across classifier types and languages. We evaluate 19 English models and six multi/monolingual models across lemma and inflection prediction tasks. Detailed layer-wise accuracy and selectivity tables are provided in Appendix §G.

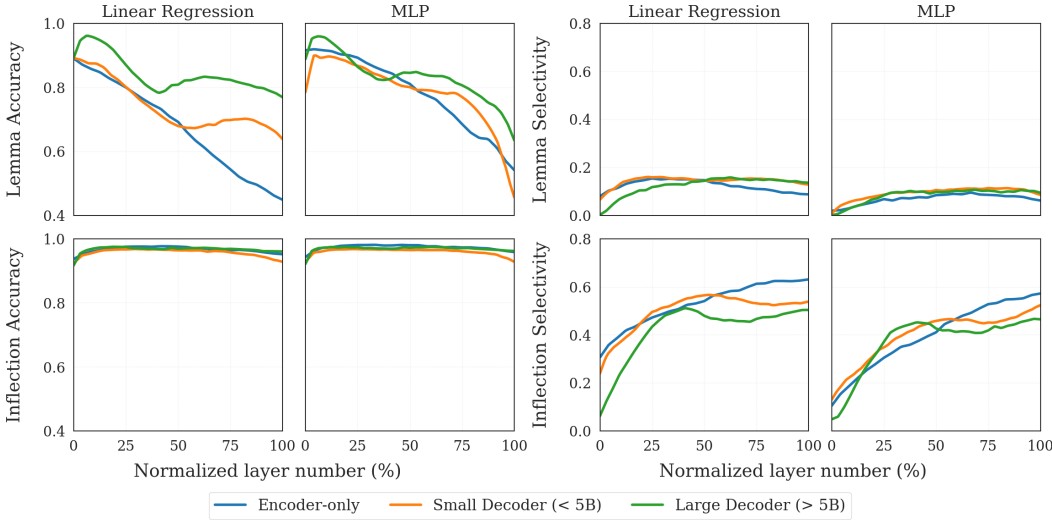

Figure 4: Lemma and inflection probing results for English, *averaged by model category*: encoder-only (BERT, DeBERTa), small decoder <5B (GPT-2, Gemma-2-2B (and instruct), Qwen2.5-1.5B (and instruct)), and large decoder >5B (Pythia-6.9B, OLMo-2-7B, Llama-3.1-8B and instruct versions). Columns show prediction accuracy (Linear vs. MLP probes) and selectivity scores (linguistic minus control accuracy). Note that for readability, the y-axis for accuracy starts at 0.4. Full (non-averaged) results for individual models are provided in Appendix §F.

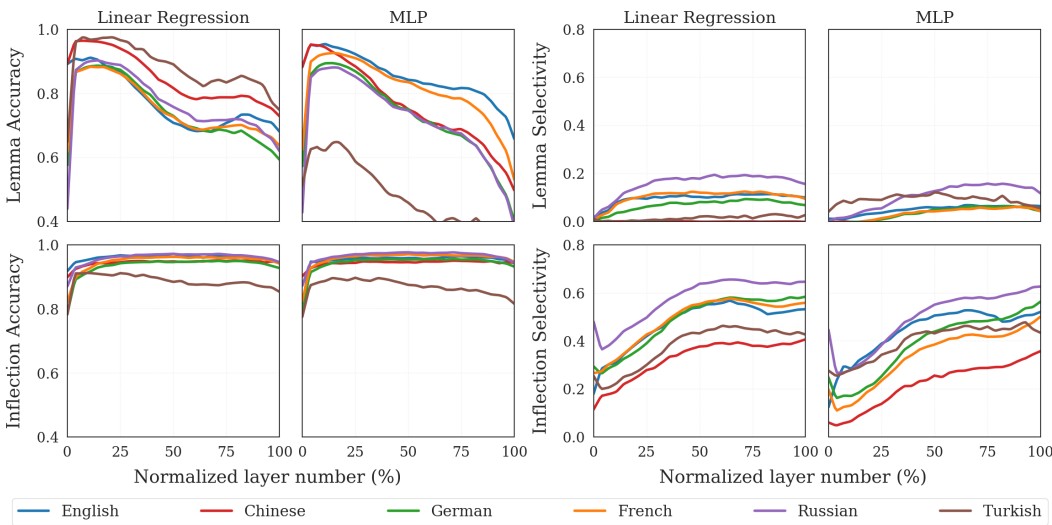

Figure 5: Cross-linguistic probing results *averaged across all models within each language*. Each language averages over multilingual models mT5-base, Qwen2.5-1.5B (and instruct), Qwen2.5-7B (and instruct) and its corresponding monolingual Goldfish <Language> 1000mb model (*e.g.*, Goldfish English 1000mb). Columns show lemma and inflection accuracy (Linear vs. MLP) followed by selectivity scores. Note that for readability, the y-axis for accuracy starts at 0.4. Full (non-averaged) results for individual models are provided in Appendix §F.

**Lemma.** Lemma accuracy under linear regression starts high (0.8–1.0) and decreases with depth in all English model families (Figure 4, top left). Encoder-only models show the strongest decrease, while small decoders decline more gradually and large decoders maintain higher accuracy in deeper layers. Across languages (Figure 5, top left), Turkish shows the largest drop (0.95 to 0.25), while Russian and Chinese retain 0.6–0.8 accuracy in later layers. MLP accuracy is similar but slightly higher than linear at most depths (middle column). Selectivity for lemma remains close to zero across depths and languages (right column), indicating that high lemma accuracy early in the network is mostly driven by surface correlations rather than strongly selective lexical structure.

**Inflection.** Inflectional features remain readable across all layers and architectures. For English, linear regression accuracy stays near 0.9–1.0 throughout the layers (Figure 4, bottom left). This pattern holds cross-linguistically (Figure 5, bottom left): English, German, French, and Russian exceed 0.9 accuracy at most depths, while Turkish is slightly lower, hovering around 0.8–0.9. MLP probes follow the same pattern (middle column). Selectivity scores for inflection remain positive (0.4–0.6) across models and languages (right column), with Russian and German at the upper end, supporting the view that inflectional features are encoded in stable, linearly accessible subspaces.

**Probe error analysis.** Frequency strongly correlates with probe accuracy for both tasks. Frequent lemmas and inflectional categories achieve higher accuracy, while rare words and rare inflections account for most errors. For inflection, comparative and superlative degrees and low-frequency verb forms are the most error-prone categories. Turkish shows the strongest sensitivity to frequency, likely due to its morphological complexity creating a long tail of rare forms. A detailed breakdown by part of speech and inflectional category is given in Appendix §L.

## 5.2 ANALYSIS

Our results show that lemma identity is encoded strongly in early layers but becomes less accessible in deeper layers, whereas inflectional features remain robustly decodable throughout the model. We analyze this further along several axes.

**Inflection is linearly separable; lemma shows limited nonlinearity.** We report the linear separability gap, defined in equation (2), which measures the accuracy difference between MLP and linear probes. Detailed plots for lemma and inflection appear in Appendix §I.3. For inflection, the gap stays close to zero across layers, architectures, and languages, typically within $\pm 0.02$ accuracy, consistent with the near-overlap of linear and MLP curves in Figures 4 and 5. This is evidence that inflectional features are encoded linearly in the representations. For lemma, gaps are modest but positive, especially in early and middle layers of encoder-only models and smaller decoders, where MLPs achieve slightly higher accuracy than linear probes before both degrade in deeper layers. This suggests that lemma information is present but less linearly separable than inflection.

**Some models show extreme mid-layer dimensionality compression; others gradually compress representations.** To characterize the representation geometry of these models, we estimate intrinsic dimensionality by counting the fraction of principal components required to reach fixed variance thresholds over our entire dataset of collected activations (full results appear in Appendix §I.1).Encoder-only models (BERT, DeBERTa) and several decoders (Gemma, Llama, OLMo-2) exhibit gradual compression: even at 90–99% variance, the relative number of components decreases only slowly as depth increases. In contrast, GPT-2, Qwen2.5, and Pythia enter a regime in their middle layers where very few components - often just a single dimension - account for most of the variance at these thresholds. Analysis of activation statistics (Appendix §I.2) reveals that this low intrinsic dimensionality is driven by outlier dimensions with large activation values: models like Qwen2.5-1.5B reach maximum absolute activations of 8000 in middle layers, while models like Llama-3-8B reach values of only 30-40 (Sun et al., 2024; Rudman et al., 2023).

**Residual streams retain more linguistic information than attention outputs.** Probing attention-head outputs and residual-stream activations for BERT and contemporary decoders (Figures 27 and 28) highlights different roles for these components. For both lemma and inflection, probes on attention outputs yield lower accuracy than probes on the residual stream at almost all depths. For lemma, attention-based accuracy falls to around 0.2–0.4 in middle layers, while residual streams remain closer to 0.6–0.9. For inflection, both components maintain high accuracy (0.7–1.0), but residual streams consistently outperform attention outputs, particularly in middle layers. Selectivity follows the same pattern: lemma selectivity is near zero for attention outputs and higher for residuals, while

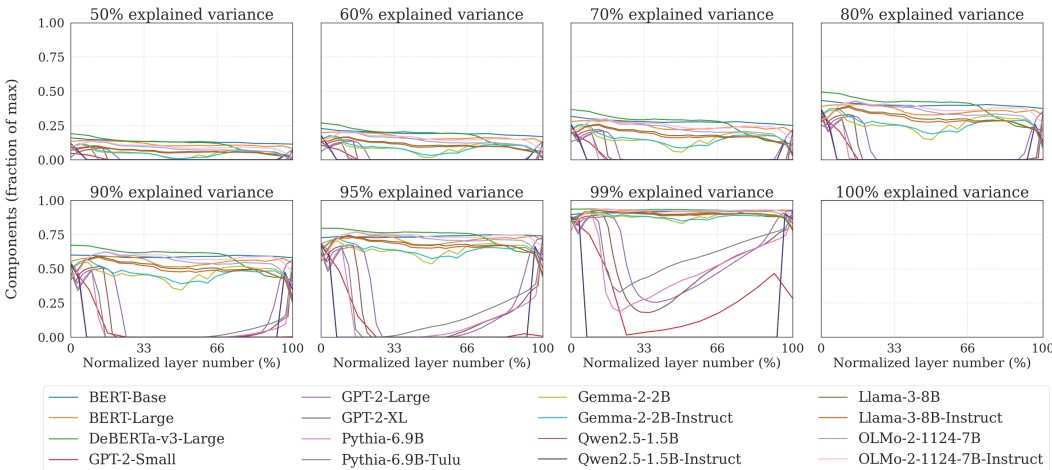

Figure 6: Intrinsic dimensionality across layers. Lines show fraction of PCA components needed to reach variance thresholds (50–100%). Models with strong mid-layer compression (few components for high variance) align with the inflection-is-linear, lemma-is-nonlinear split. Full curves per model appear in Figure 15.

inflection selectivity reaches 0.4–0.5 in both streams with residuals slightly higher. These results support a view in which attention primarily aggregates contextual relationships, whereas the residual stream/MLP layers preserve token-level lexical and morphological information that supports both lemma and inflection prediction.

**Inflection steering effectiveness tracks intrinsic dimensionality.** Steering experiments with inflection features (*e.g.*, , singular vs. plural) connect these representational properties to causal control. For each pair of categories, we compute a difference vector between mean hidden states and apply scaled interventions at each layer. Figures 29 and 30 show that, for most architectures and layers, even moderate steering scales (*e.g.*, , $\lambda = 5$) yield large changes in predicted inflection and high flip rates, indicating that a single direction in activation space can reliably control morphological representations. `Qwen2.5` variants demonstrate an interesting property: in their early–middle layers, steering is much less effective, with both probability change and flip rate reaching their minima. This region aligns with the layers where intrinsic dimensionality is lowest in Figures 6 and 15. Combined with the accuracy curves in Figure 4, this suggests that strongly compressed representations are more resistant to causal manipulation, even when inflection remains linearly decodable, whereas higher-dimensional layers permit more effective steering of inflectional morphology.

**Inflection stabilizes early in training; lemma continues to change.** Pretraining checkpoint analysis for `OLMo-2-7B` and `Pythia-6.9B` (Figures 25 and 26) shows that morphological analysis emerges very early, whereas lemma information continues to evolve with training. For both model families, inflection accuracy is already high at the earliest checkpoints and increases only slightly with additional updates; inflection selectivity grows quickly in the first few checkpoints and then remains near its final value. Lemma behaves differently. In `OLMo-2-7B`, lemma accuracy and selectivity increase gradually across checkpoints, with the largest gains in middle layers. In `Pythia-6.9B`, early checkpoints exhibit much lower lemma accuracy and near-zero lemma selectivity in deeper layers, and both quantities rise steadily as training progresses. These trends indicate that models identify and stabilize inflectional categories early in pretraining, while lemma representations remain more plastic and continue to be reshaped throughout training, especially in the later layers of decoder-only models.

## 5.3 Discussion

The previous analyses present a comprehensive picture of how lemma identity and inflectional features are organized inside transformer language models. Lemma information is strongly encoded in early layers but becomes less accessible as depth increases, particularly in models that undergo strong mid-layer compression. Inflectional features, in contrast, remain decodable across virtually all layers and models, with small linear separability gaps and high selectivity.

The linear separability results suggest that grammatical features are encoded in low-dimensional, approximately linear subspaces, while lemma identity relies more on higher-variance directions that are later deemphasized. Intrinsic dimensionality measurements, together with the steering experiments, tell us that aggressive compression in some decoder-only models limits the space in which such directions can be causally manipulated. Specifically, steering remains effective in higher-dimensional regions but degrades in layers whose variance is captured by very few components. The comparison between attention outputs and residual streams further implies that lexical information is preserved in the residual stream.

Taken together, these findings point to an organization in which inflection is a stable and linearly accessible component of the internal state, supporting both probing and controlled interventions, while lemma identity is encoded in a way that is useful for early processing but increasingly traded off against compact, context-oriented representations as models optimize for next-token prediction.

## 6 RELATED WORK

**Probing for linguistic information.** Probing studies typically use supervised classifiers to predict linguistic properties from model representations (Alain & Bengio, 2017; Adi et al., 2017). Extensive work has established that early transformer models (BERT, GPT-2) learn hierarchical linguistic structures, with different layers specializing in different information types: lower layers capture surface features and morphology, middle layers encode syntax, and upper layers represent semantics and context (Jawahar et al., 2019; Tenney et al., 2019a; Rogers et al., 2020). More relevant to our work, Vulić et al. (2020) found that lexical information concentrates in lower layers, while Ethayarajh (2019) showed that representations become increasingly context-specific in higher layers.

**Representation dynamics in modern LLMs.** Recent research has extended these analyses to modern, larger-scale generative models, examining how representational geometry evolves with scale. Cheng et al. (2025) identify a distinct high-dimensional abstraction phase in the early-to-middle layers of models like Llama and OLMo, suggesting that the transition from surface-level to abstract linguistic features occurs earlier than in previous architectures. Similarly, Skean et al. (2025) demonstrate that intermediate layers in modern LLMs often encode richer task-transferable representations than final layers, challenging the assumption that semantic capability monotonically increases with depth. These findings align with the pipeline compression we observe in Section §3.

**Activation steering.** Beyond probing, recent work has explored manipulating model behavior by intervening on internal representations. This includes steering vectors (Subramani et al., 2022), inference-time interventions (Li et al., 2023), representation editing (Meng et al., 2022), sparse autoencoders for feature discovery (Bricken et al., 2023), and causal mediation analysis (Vig et al., 2020). While these methods typically evaluate changes in model outputs, our steering experiments focus on measuring representational changes. See Appendix §B for more detailed discussion.

**Mechanistic interpretability and feature discovery.** Mechanistic interpretability approaches aim to reverse-engineer the algorithms learned by neural networks (Elhage et al., 2021), offering a more causal view of internal structure. Recent work uses sparse autoencoders to decompose dense representations into interpretable latent features (Cunningham et al., 2023; Bricken et al., 2023), providing clearer targets for interpretation than raw activations. While probing detects correlations between representations and linguistic concepts, these methods seek to identify the specific components and causal circuits that implement these behaviors.

## 7 CONCLUSION

In this work, we analyzed 25 transformer models and found that modern LMs show signs of rediscovering the classical NLP pipeline, progressing from syntax to semantics and discourse. However, we observe that larger models compress this hierarchy into earlier layers, suggesting that increased capacity allows useful representations to emerge sooner. Our case study further reveals that while lemma identity becomes increasingly non-linear with depth, inflectional features remain linearly accessible and steerable within the residual stream across languages. Collectively, these findings indicate that despite rapid advances in scale and training, transformers converge on robust, shared mechanisms for organizing linguistic information.

## 8 REPRODUCIBILITY STATEMENT

We will release a GitHub repository containing code to reproduce dataset construction, probing experiments, and all plots and analyses. The main paper specifies the probe design and metrics (Section §2), datasets and model suite (Sections §3 and Table 1), and evaluation summaries for the classical pipeline and for lemma identity and inflectional features (Sections §4 and §5). The appendix provides complete classifier and training details, dataset statistics, and full-resolution figure grids referenced in the text. Together, these materials are intended to enable end-to-end reproduction of our results.

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

# A  LIMITATIONS

**Representation Extraction for Decoder Models**  Our current approach for extracting word representations from decoder-only models uses the final subword token. This assumption is an intuitive and natural choice, but may not be optimal for all architectures and models. Future work could develop better extraction methods that account for subword tokenization effects and leverage attention patterns to create more accurate word-level representations.

**Form and Function in Inflection**  Some languages contain cases where different grammatical functions share the same surface form (*e.g.*, , infinitive and non-past verb forms in English). We do not explicitly examine these cases in our classification experiments, but these ambiguities create opportunities to better examine how models separate form from function across languages.

**Indirect Nature of Classifiers**  While our classifier methodology follows established best practices (Hewitt & Liang, 2019; Liu et al., 2019), we only detect correlations in hidden activations, not causal mechanisms.

**Scope of Steering Experiments**  Our steering vector experiments measure changes in classifier performance rather than downstream model outputs. Evaluating effects on actual model generation would require more complex experimental designs to control for confounding factors and ensure that observed changes result from the intended representational modifications rather than other influences.

# B  ADDITIONAL RELATED WORK

## B.1  ADVANCED PROBING METHODOLOGIES

Beyond standard linear probes, there are many sophisticated approaches to understand model representations. Amnesic probing (Elazar et al., 2021) removes specific information from representations to test whether it's necessary for downstream tasks. Minimum description length probes (Voita & Titov, 2020) balance probe complexity with performance to avoid overfitting. Causal abstraction (Geiger et al., 2021) aims to establish causal rather than merely correlational relationships between representations and linguistic properties. Recently, Subramani et al. (2025) find that decoding from activations directly using the Logit Lens can be used to learn confidence estimators for tool-calling agents (nostalgebraist, 2020).

## B.2  MODEL MANIPULATION AND STEERING

Steering vectors demonstrate that specific directions in activation space correspond to high-level behavioral changes (Subramani et al., 2022). Building on this, Panickssery et al. (2024) achieves behavioral control by adding activation differences between contrasting examples. Li et al. (2023) introduce inference-time intervention, a method that shifts model activations during inference across limited attention heads to control model behavior. While these methods operate in activation space, task vectors enable arithmetic operations on model capabilities by manipulating weight space (Ilharco et al., 2023).

Recent work has also examined how multilingual models like mT5 and ByT5 encode morphological information differently across languages (Dang et al., 2024), finding that tokenization strategies significantly impact morphological representation quality, particularly for morphologically rich languages.

# C  PROBE DETAILS

In this appendix we provide implementation details for the linear regression and two-layer multi-layer perceptron (MLP) probes used throughout our experiments. These classifiers are trained on frozen residual-stream activations from each layer to predict the labels of our linguistic tasks, lemma identity and inflectional features.

**Training details.** We stratify each dataset into train, validation, and test splits. Probes are trained on the training split, hyperparameters are selected using the validation split, and we report accuracy and macro F1 on the held-out test split. For the linear regression probe we apply ridge regularization with $\lambda = 0.01$ and solve equation (5) in closed form. For the MLP probe we use a hidden dimension of 64, a learning rate of 0.001, weight decay of 0.01, and train for up to 100 epochs with early stopping based on validation loss, optimizing cross-entropy with AdamW. Both probes share the same data splits to enable fair comparison.

## C.1  LINEAR REGRESSION CLASSIFIER

Consistent with best practices for probing (Hewitt & Liang, 2019; Liu et al., 2019), we use a ridge-regularized linear regression classifier. Given training representations $X_{\text{train}} \in \mathbb{R}^{m \times d}$ and one-hot encoded labels $Y_{\text{train}} \in \mathbb{R}^{m \times c}$, the optimal weight matrix $W \in \mathbb{R}^{d \times c}$ is obtained in closed form as

$$W = \left( X_{\text{train}}^{\top} X_{\text{train}} + \lambda I \right)^{-1} X_{\text{train}}^{\top} Y_{\text{train}}, \tag{5}$$

where $\lambda$ controls the strength of $\ell_2$ regularization and $I$ is the identity matrix. Predictions on test representations $X_{\text{test}}$ are then given by $\hat{Y}_{\text{test}} = X_{\text{test}} W$.

## C.2  MLP CLASSIFIER

To test for non-linear separability, we train a simple two-layer MLP with ReLU activation. The classifier computes

$$\hat{Y} = \text{softmax}\Big( \text{ReLU}(X W_1) W_2 \Big), \tag{6}$$

where $W_1 \in \mathbb{R}^{d \times h}$ and $W_2 \in \mathbb{R}^{h \times c}$ are learned weight matrices, $h$ is the hidden dimension (we use $h = 64$), and biases are omitted for brevity. Two-layer MLPs with ReLU activation are universal function approximators capable of representing any continuous function to arbitrary precision given sufficient width (Hornik et al., 1989). We train the MLP with cross-entropy loss using the same splits and optimization hyperparameters described above.

# D  LINGUISTIC TASK DEFINITIONS

We probe eight linguistic tasks originally introduced by Tenney et al. (2019a) that span the classical NLP pipeline. Here we provide formal definitions for each task:

**Part-of-Speech tagging (POS).** This task assigns each word a grammatical category such as noun, verb, adjective, or adverb, following the Universal Dependencies tagset (Petrov et al., 2012). POS tagging is fundamental to syntactic analysis and serves as input to many downstream NLP tasks.

**Constituency parsing.** This task identifies the hierarchical phrase structure of sentences by grouping words into nested constituents such as noun phrases, verb phrases, and sentences (Marcus et al., 1993). The output is typically represented as a parse tree showing how smaller units combine to form larger syntactic structures.

**Dependency parsing.** This task predicts syntactic head-dependent relations between words, such as subject-verb and modifier-head relationships, following Universal Dependencies guidelines (Nivre et al., 2016). Each word is linked to exactly one head (except the root), forming a directed tree structure that captures grammatical relations.

**Named Entity Recognition (NER).** This task identifies and classifies named entities such as persons, organizations, locations, and dates into predefined categories (Tjong Kim Sang & De Meulder, 2003). NER bridges syntactic and semantic analysis by identifying referential expressions that denote real-world entities.

**Semantic Role Labeling (SRL).** This task assigns semantic roles such as agent, patient, instrument, or location to arguments of predicates in a sentence (Gildea & Jurafsky, 2002). SRL captures the underlying semantic relationships between predicates and their arguments, abstracting away from surface syntactic variations.

**Semantic Proto-Roles (SPR).**    This task predicts fine-grained semantic properties of predicate arguments, such as whether an argument is sentient, undergoes a change of state, or is volitional (Reisinger et al., 2015). SPR provides a more nuanced characterization of semantic roles through scalar properties rather than categorical labels.

**Coreference resolution.**    This task determines which expressions in a text refer to the same real-world entity, linking pronouns and noun phrases to their antecedents (Pradhan et al., 2012). Coreference resolution is essential for understanding discourse coherence and tracking entities across sentences.

**Relation extraction.**    This task identifies semantic relationships between entity mentions, such as organization-location or person-affiliation relations, typically across sentence boundaries (Hendrickx et al., 2010). Relation extraction connects named entities through typed semantic links, enabling structured knowledge representation.

These tasks form the classical NLP pipeline described by (Tenney et al., 2019a), progressing from syntactic analysis (POS, constituency, dependencies) through semantic interpretation (NER, SRL, SPR) to discourse-level understanding (coreference, relations).

## E   FULL RESULTS FOR THE CLASSICAL NLP PIPELINE

The full heatmaps and summary statistics for pipeline analyses across all models are shown in Figures 7–11. These figures show model-by-layer accuracy/selectivity patterns and the expected layer/center-of-gravity summaries reported in the main text.

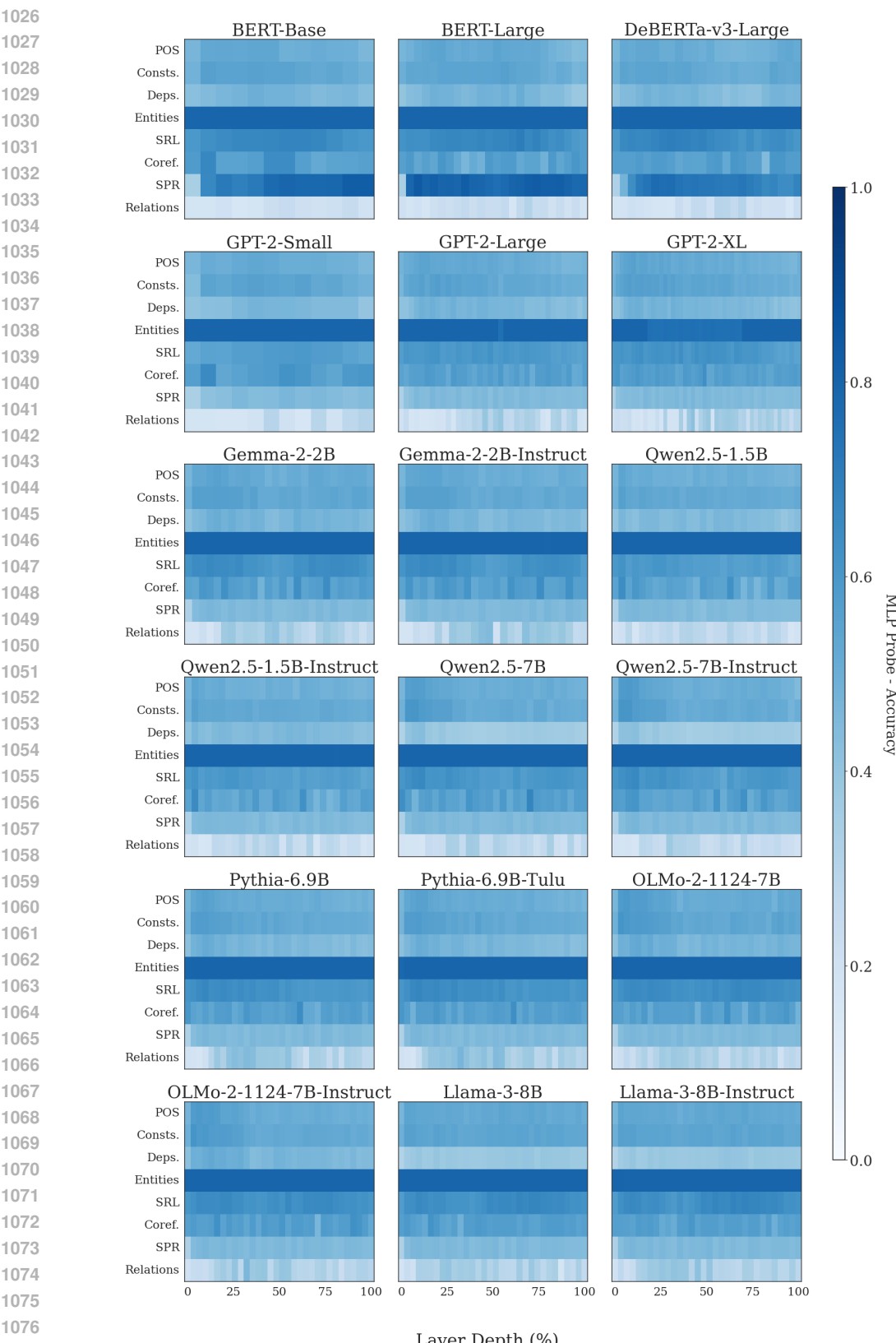

Figure 7: Full heatmaps for MLP probe accuracy across all tasks, models, and layers. Rows show tasks; columns show models; each plot shows accuracy by layer depth.

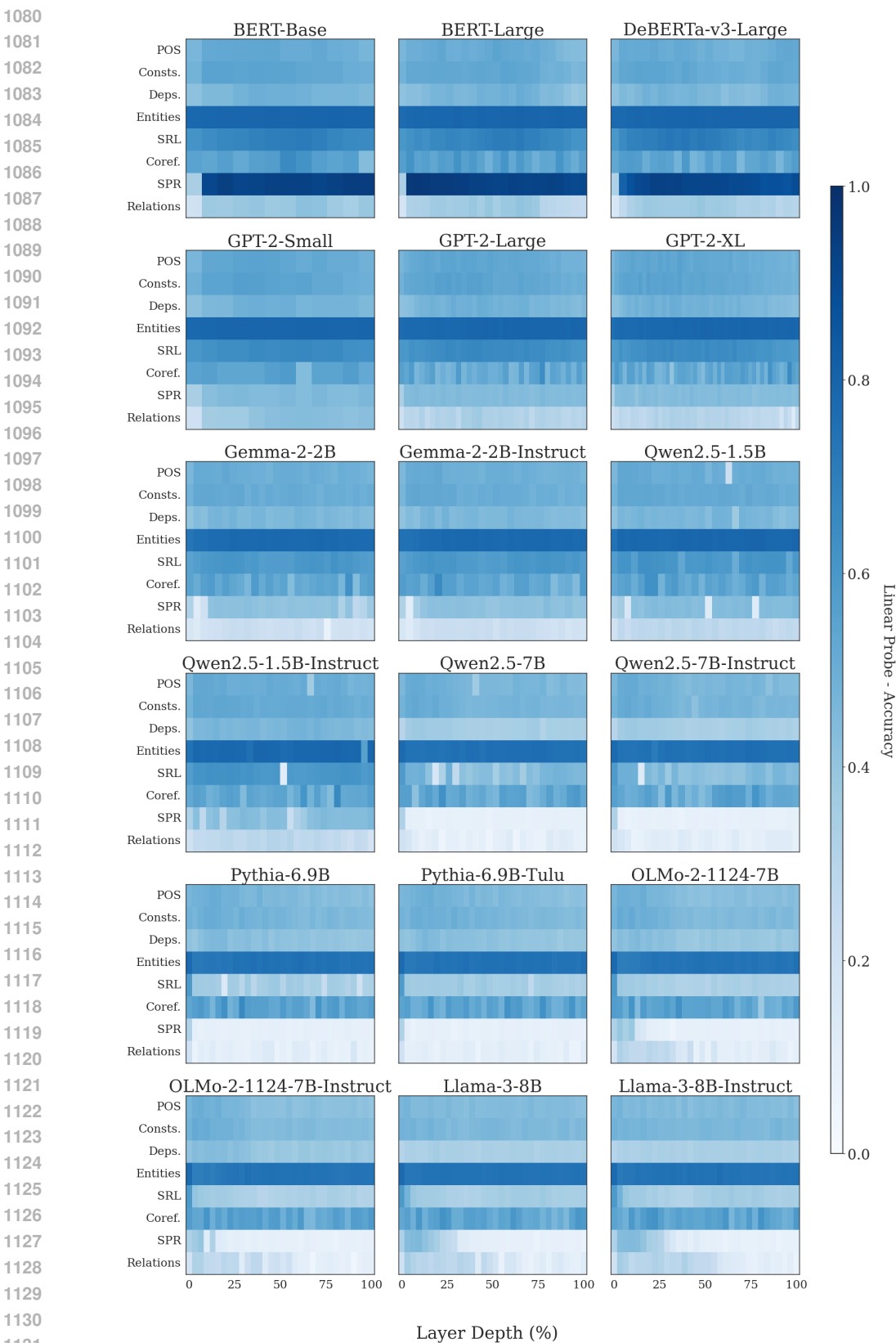

Figure 8: Full heatmaps for linear probe accuracy across all tasks, models, and layers. Trends mirror the MLP version but with stronger model-size effects in deeper layers.

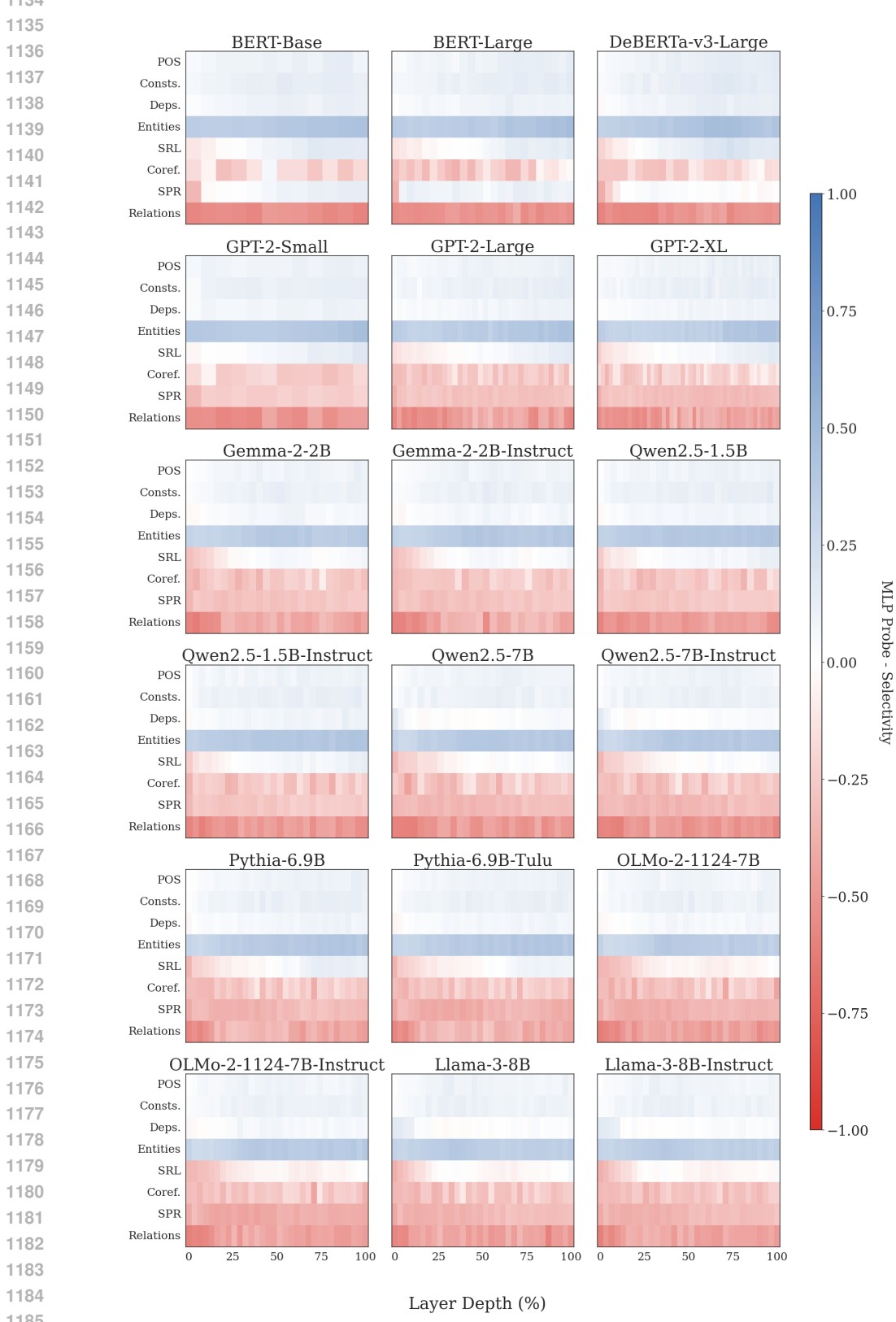

Figure 9: Full heatmaps for MLP probe selectivity (real vs. control task accuracy).

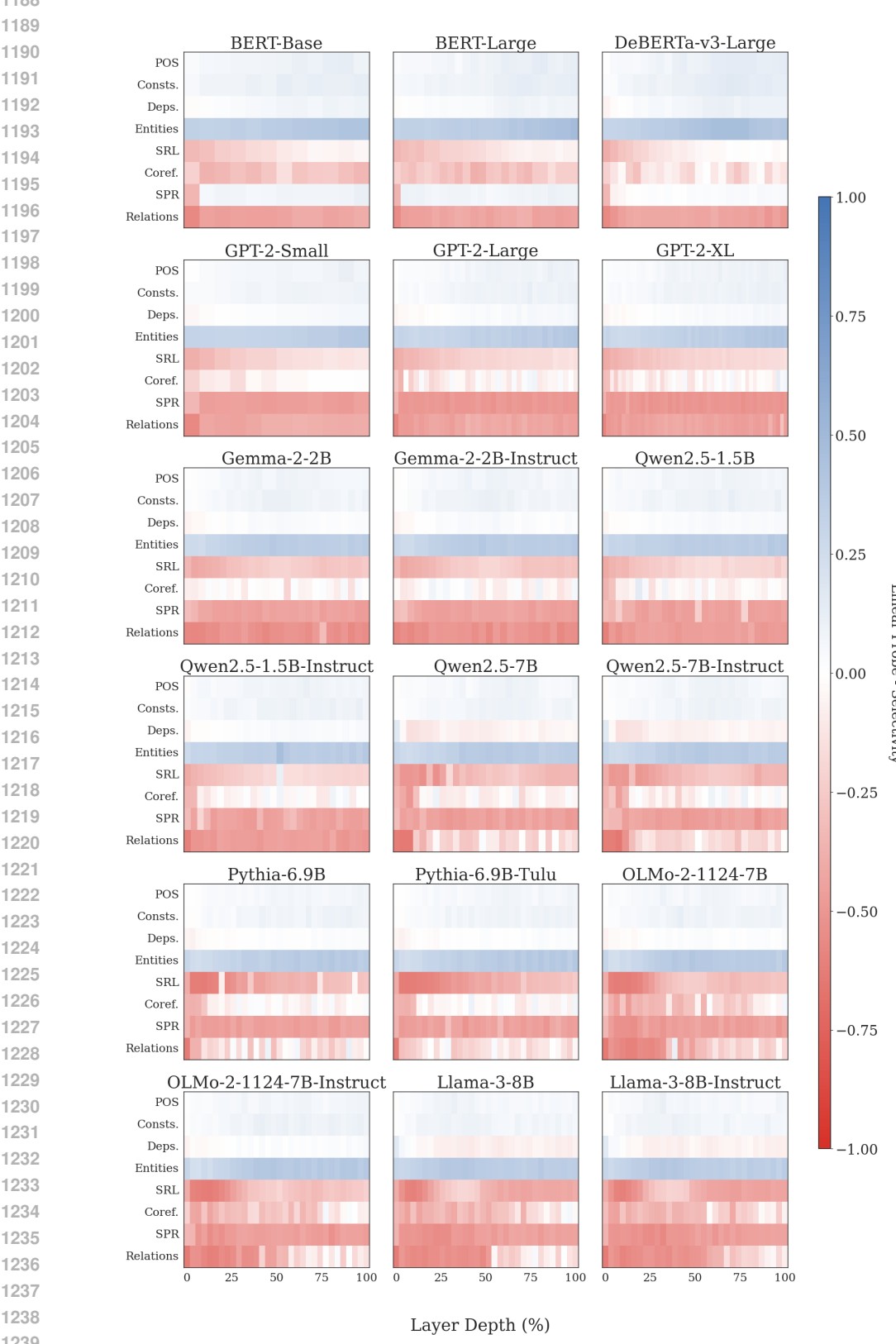

Figure 10: Full heatmaps for linear probe selectivity (real vs. control task accuracy).

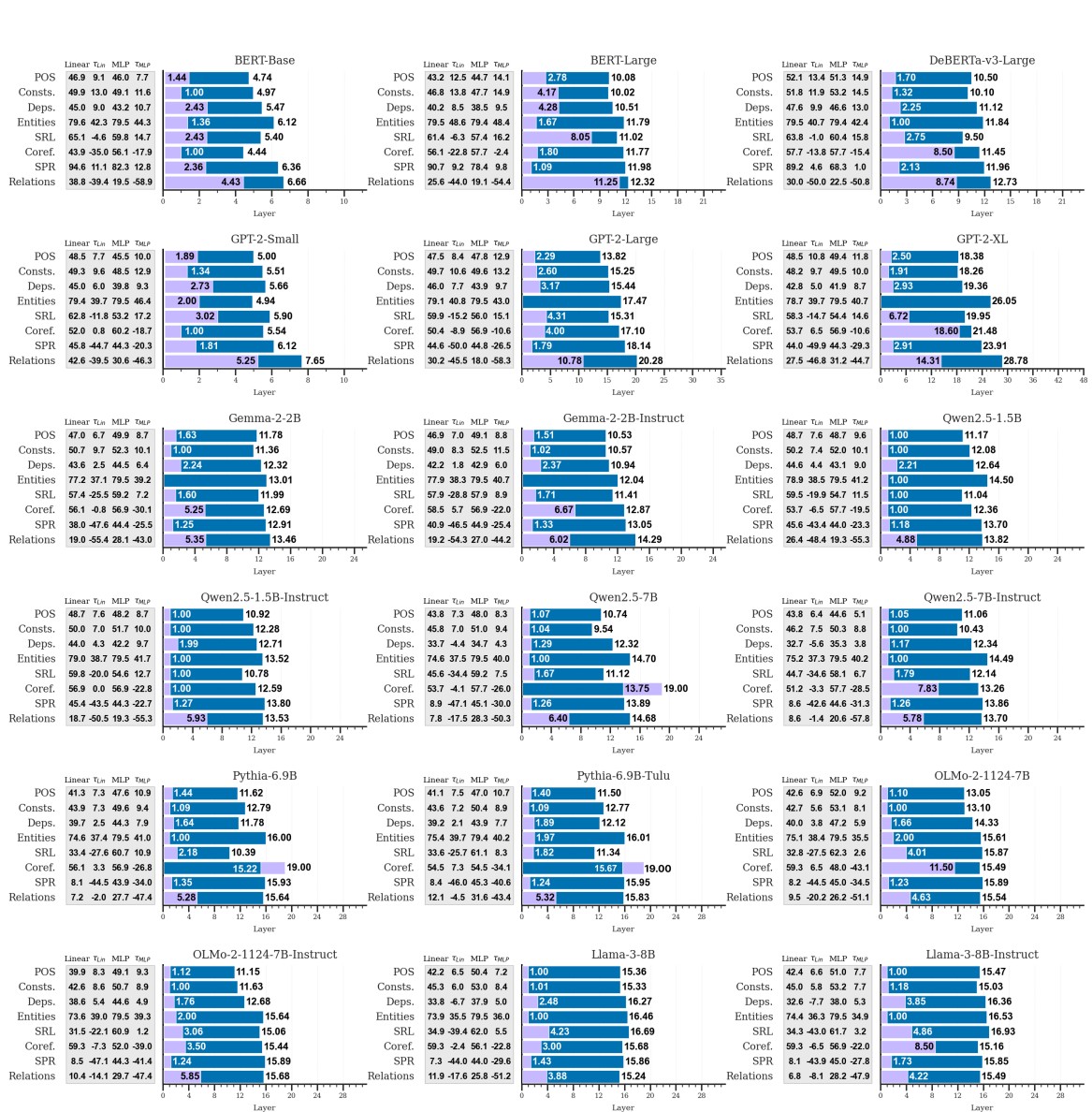

Figure 11: Expected Layer and Center of Gravity plots with $\tau_{\text{lin}}$ and $\tau_{\text{MLP}}$ selectivity scores for all models.

## F  FULL LEMMA AND INFLECTION PROBE RESULTS

We provide the full, non-averaged results for the linguistic probing tasks (lemma identity and inflectional features) for every individual model. Figure 12 shows the detailed breakdown for English models, and Figure 13 presents the results for all six languages.

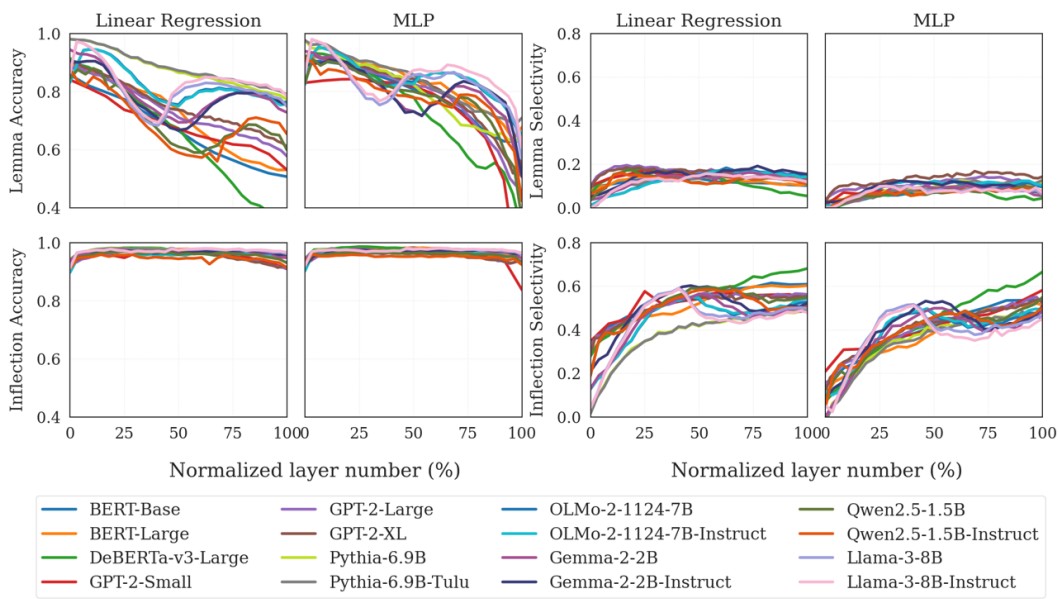

Figure 12: Full lemma and inflection probing results for English, showing individual curves for every model. Columns show prediction accuracy (Linear vs. MLP probes) and selectivity scores (linguistic minus control accuracy).

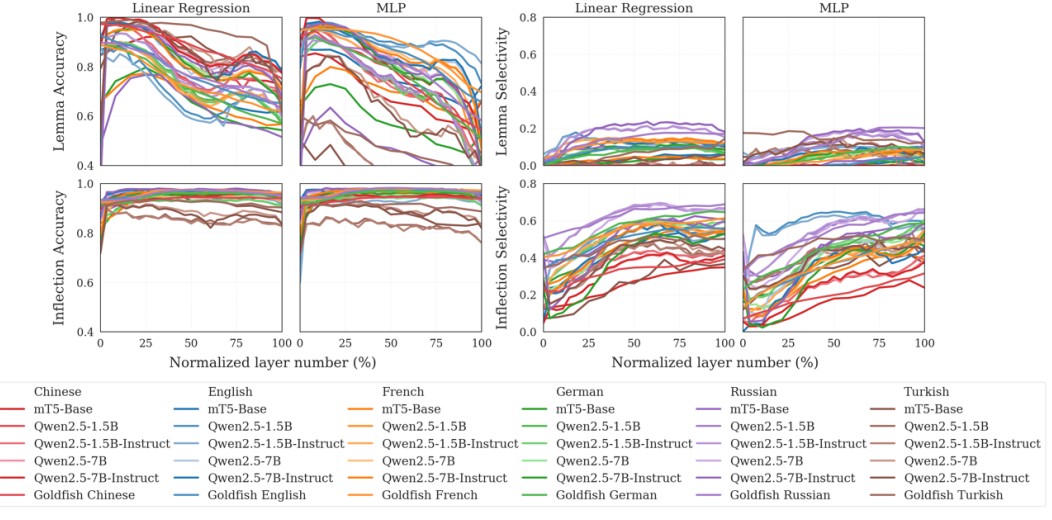

Figure 13: Full cross-linguistic probing results showing individual curves for every model within each language. Columns show lemma and inflection accuracy (Linear vs. MLP) followed by selectivity scores.

## G  DETAILED LAYER-WISE TABLES FOR LEMMA AND INFLECTION RESULTS

This section contains detailed tables for layer-wise accuracy and selectivity across all models and languages.

Table 2: English Accuracy (Linear Probes) Across Layer Depths

| Model | Task | 0% | 25% | 50% | 75% | 100% |
|-------|------|------|------|------|------|------|
| BERT-Base | Inflection | 0.934 | 0.971 | 0.977 | 0.966 | 0.951 |
| | Lexeme | 0.858 | 0.773 | 0.665 | 0.564 | 0.507 |
| BERT-Large | Inflection | 0.938 | 0.972 | 0.978 | 0.966 | 0.949 |
| | Lexeme | 0.896 | 0.820 | 0.737 | 0.585 | 0.531 |
| DeBERTa-v3-Large | Inflection | 0.939 | 0.982 | 0.972 | 0.960 | 0.955 |
| | Lexeme | 0.914 | 0.805 | 0.673 | 0.482 | 0.309 |
| GPT-2-Large | Inflection | 0.927 | 0.970 | 0.971 | 0.946 | 0.912 |
| | Lexeme | 0.874 | 0.818 | 0.711 | 0.665 | 0.577 |
| GPT-2-Small | Inflection | 0.939 | 0.948 | 0.971 | 0.956 | 0.909 |
| | Lexeme | 0.840 | 0.737 | 0.671 | 0.618 | 0.530 |
| GPT-2-XL | Inflection | 0.929 | 0.974 | 0.973 | 0.946 | 0.915 |
| | Lexeme | 0.906 | 0.827 | 0.737 | 0.690 | 0.609 |
| Gemma-2-2B | Inflection | 0.936 | 0.974 | 0.963 | 0.972 | 0.948 |
| | Lexeme | 0.944 | 0.817 | 0.694 | 0.792 | 0.728 |
| Gemma-2-2B-Instruct | Inflection | 0.917 | 0.971 | 0.960 | 0.972 | 0.960 |
| | Lexeme | 0.904 | 0.802 | 0.667 | 0.791 | 0.752 |
| Llama-3-8B | Inflection | 0.912 | 0.971 | 0.974 | 0.976 | 0.962 |
| | Lexeme | 0.864 | 0.794 | 0.796 | 0.816 | 0.749 |
| Llama-3-8B-Instruct | Inflection | 0.913 | 0.972 | 0.974 | 0.977 | 0.967 |
| | Lexeme | 0.864 | 0.803 | 0.812 | 0.839 | 0.783 |
| OLMo-2-1124-7B | Inflection | 0.897 | 0.970 | 0.959 | 0.961 | 0.965 |
| | Lexeme | 0.832 | 0.883 | 0.755 | 0.808 | 0.763 |
| OLMo-2-1124-7B-Instruct | Inflection | 0.897 | 0.970 | 0.958 | 0.962 | 0.961 |
| | Lexeme | 0.832 | 0.881 | 0.749 | 0.806 | 0.757 |
| Pythia-6.9B | Inflection | 0.942 | 0.982 | 0.974 | 0.966 | 0.953 |
| | Lexeme | 0.980 | 0.928 | 0.865 | 0.829 | 0.772 |
| Pythia-6.9B-Tulu | Inflection | 0.941 | 0.980 | 0.975 | 0.968 | 0.954 |
| | Lexeme | 0.980 | 0.928 | 0.872 | 0.841 | 0.789 |
| Qwen2.5-1.5B | Inflection | 0.913 | 0.969 | 0.959 | 0.961 | 0.930 |
| | Lexeme | 0.845 | 0.799 | 0.610 | 0.654 | 0.599 |
| Qwen2.5-1.5B-Instruct | Inflection | 0.910 | 0.957 | 0.944 | 0.949 | 0.914 |
| | Lexeme | 0.876 | 0.768 | 0.590 | 0.647 | 0.654 |

Table 3: English Selectivity (Linear Probes) Across Layer Depths

| Model | Task | 0% | 25% | 50% | 75% | 100% |
|---|---|---|---|---|---|---|
| BERT-Base | Inflection | 0.348 | 0.486 | 0.556 | 0.596 | 0.609 |
| | Lexeme | 0.094 | 0.132 | 0.134 | 0.114 | 0.101 |
| BERT-Large | Inflection | 0.292 | 0.459 | 0.520 | 0.598 | 0.603 |
| | Lexeme | 0.083 | 0.145 | 0.148 | 0.119 | 0.107 |
| DeBERTa-v3-Large | Inflection | 0.279 | 0.471 | 0.545 | 0.644 | 0.681 |
| | Lexeme | 0.062 | 0.183 | 0.158 | 0.097 | 0.054 |
| GPT-2-Large | Inflection | 0.324 | 0.489 | 0.563 | 0.550 | 0.562 |
| | Lexeme | 0.099 | 0.183 | 0.162 | 0.161 | 0.121 |
| GPT-2-Small | Inflection | 0.347 | 0.576 | 0.541 | 0.558 | 0.547 |
| | Lexeme | 0.100 | 0.171 | 0.152 | 0.134 | 0.121 |
| GPT-2-XL | Inflection | 0.289 | 0.488 | 0.549 | 0.547 | 0.553 |
| | Lexeme | 0.092 | 0.172 | 0.172 | 0.162 | 0.135 |
| Gemma-2-2B | Inflection | 0.131 | 0.466 | 0.565 | 0.464 | 0.512 |
| | Lexeme | 0.036 | 0.132 | 0.136 | 0.161 | 0.128 |
| Gemma-2-2B-Instruct | Inflection | 0.210 | 0.517 | 0.594 | 0.487 | 0.522 |
| | Lexeme | 0.052 | 0.157 | 0.144 | 0.184 | 0.154 |
| Llama-3-8B | Inflection | 0.041 | 0.516 | 0.482 | 0.474 | 0.497 |
| | Lexeme | -0.003 | 0.132 | 0.146 | 0.140 | 0.117 |
| Llama-3-8B-Instruct | Inflection | 0.042 | 0.508 | 0.473 | 0.453 | 0.478 |
| | Lexeme | -0.003 | 0.131 | 0.146 | 0.144 | 0.119 |
| OLMo-2-1124-7B | Inflection | 0.128 | 0.440 | 0.538 | 0.476 | 0.528 |
| | Lexeme | 0.011 | 0.104 | 0.152 | 0.170 | 0.144 |
| OLMo-2-1124-7B-Instruct | Inflection | 0.127 | 0.443 | 0.545 | 0.483 | 0.543 |
| | Lexeme | 0.011 | 0.104 | 0.149 | 0.164 | 0.144 |
| Pythia-6.9B | Inflection | 0.017 | 0.347 | 0.416 | 0.457 | 0.484 |
| | Lexeme | -0.002 | 0.117 | 0.134 | 0.140 | 0.141 |
| Pythia-6.9B-Tulu | Inflection | 0.016 | 0.348 | 0.419 | 0.454 | 0.492 |
| | Lexeme | -0.002 | 0.118 | 0.137 | 0.143 | 0.150 |
| Qwen2.5-1.5B | Inflection | 0.199 | 0.483 | 0.589 | 0.566 | 0.548 |
| | Lexeme | 0.034 | 0.146 | 0.117 | 0.129 | 0.108 |
| Qwen2.5-1.5B-Instruct | Inflection | 0.189 | 0.474 | 0.581 | 0.541 | 0.503 |
| | Lexeme | 0.061 | 0.143 | 0.113 | 0.129 | 0.112 |

Table 4: English Accuracy (MLP Probes) Across Layer Depths

| Model | Task | 0% | 25% | 50% | 75% | 100% |
|---|---|---|---|---|---|---|
| BERT-Base | Inflection | 0.941 | 0.978 | 0.983 | 0.975 | 0.960 |
| | Lexeme | 0.906 | 0.888 | 0.808 | 0.719 | 0.629 |
| BERT-Large | Inflection | 0.943 | 0.977 | 0.981 | 0.973 | 0.958 |
| | Lexeme | 0.920 | 0.904 | 0.843 | 0.762 | 0.677 |
| DeBERTa-v3-Large | Inflection | 0.944 | 0.986 | 0.976 | 0.971 | 0.957 |
| | Lexeme | 0.920 | 0.885 | 0.781 | 0.574 | 0.318 |
| GPT-2-Large | Inflection | 0.930 | 0.964 | 0.964 | 0.952 | 0.937 |
| | Lexeme | 0.892 | 0.878 | 0.825 | 0.719 | 0.552 |
| GPT-2-Small | Inflection | 0.943 | 0.963 | 0.965 | 0.957 | 0.837 |
| | Lexeme | 0.830 | 0.843 | 0.817 | 0.705 | 0.075 |
| GPT-2-XL | Inflection | 0.929 | 0.965 | 0.963 | 0.948 | 0.937 |
| | Lexeme | 0.906 | 0.884 | 0.844 | 0.734 | 0.571 |
| Gemma-2-2B | Inflection | 0.940 | 0.977 | 0.970 | 0.975 | 0.946 |
| | Lexeme | 0.939 | 0.868 | 0.732 | 0.812 | 0.506 |
| Gemma-2-2B-Instruct | Inflection | 0.919 | 0.973 | 0.965 | 0.973 | 0.960 |
| | Lexeme | 0.890 | 0.874 | 0.731 | 0.831 | 0.508 |
| Llama-3-8B | Inflection | 0.920 | 0.972 | 0.977 | 0.976 | 0.958 |
| | Lexeme | 0.863 | 0.808 | 0.857 | 0.840 | 0.568 |
| Llama-3-8B-Instruct | Inflection | 0.920 | 0.972 | 0.977 | 0.977 | 0.964 |
| | Lexeme | 0.863 | 0.821 | 0.873 | 0.870 | 0.605 |
| OLMo-2-1124-7B | Inflection | 0.903 | 0.974 | 0.970 | 0.975 | 0.964 |
| | Lexeme | 0.825 | 0.877 | 0.833 | 0.845 | 0.621 |
| OLMo-2-1124-7B-Instruct | Inflection | 0.903 | 0.975 | 0.968 | 0.973 | 0.964 |
| | Lexeme | 0.825 | 0.880 | 0.825 | 0.847 | 0.650 |
| Pythia-6.9B | Inflection | 0.940 | 0.973 | 0.971 | 0.959 | 0.959 |
| | Lexeme | 0.976 | 0.891 | 0.823 | 0.683 | 0.655 |
| Pythia-6.9B-Tulu | Inflection | 0.944 | 0.973 | 0.970 | 0.963 | 0.961 |
| | Lexeme | 0.976 | 0.904 | 0.858 | 0.752 | 0.709 |
| Qwen2.5-1.5B | Inflection | 0.919 | 0.967 | 0.963 | 0.962 | 0.929 |
| | Lexeme | 0.829 | 0.864 | 0.800 | 0.777 | 0.356 |
| Qwen2.5-1.5B-Instruct | Inflection | 0.935 | 0.957 | 0.952 | 0.952 | 0.923 |
| | Lexeme | 0.916 | 0.841 | 0.779 | 0.789 | 0.423 |

Table 5: English Selectivity (MLP Probes) Across Layer Depths

| Model | Task | 0% | 25% | 50% | 75% | 100% |
|---|---|---|---|---|---|---|
| BERT-Base | Inflection | 0.139 | 0.307 | 0.416 | 0.505 | 0.536 |
| | Lexeme | 0.025 | 0.057 | 0.069 | 0.084 | 0.068 |
| BERT-Large | Inflection | 0.104 | 0.294 | 0.386 | 0.496 | 0.514 |
| | Lexeme | 0.028 | 0.084 | 0.088 | 0.106 | 0.074 |
| DeBERTa-v3-Large | Inflection | 0.071 | 0.313 | 0.428 | 0.582 | 0.666 |
| | Lexeme | 0.005 | 0.061 | 0.096 | 0.065 | 0.043 |
| GPT-2-Large | Inflection | 0.157 | 0.323 | 0.446 | 0.487 | 0.521 |
| | Lexeme | 0.040 | 0.093 | 0.123 | 0.138 | 0.115 |
| GPT-2-Small | Inflection | 0.210 | 0.342 | 0.410 | 0.470 | 0.581 |
| | Lexeme | 0.012 | 0.077 | 0.094 | 0.107 | 0.061 |
| GPT-2-XL | Inflection | 0.118 | 0.334 | 0.433 | 0.491 | 0.527 |
| | Lexeme | 0.027 | 0.110 | 0.145 | 0.163 | 0.142 |
| Gemma-2-2B | Inflection | 0.002 | 0.360 | 0.499 | 0.395 | 0.483 |
| | Lexeme | -0.007 | 0.067 | 0.090 | 0.088 | 0.054 |
| Gemma-2-2B-Instruct | Inflection | 0.089 | 0.393 | 0.523 | 0.404 | 0.509 |
| | Lexeme | 0.026 | 0.102 | 0.111 | 0.100 | 0.058 |
| Llama-3-8B | Inflection | 0.050 | 0.454 | 0.414 | 0.402 | 0.463 |
| | Lexeme | -0.001 | 0.098 | 0.079 | 0.084 | 0.060 |
| Llama-3-8B-Instruct | Inflection | 0.050 | 0.442 | 0.400 | 0.364 | 0.451 |
| | Lexeme | -0.002 | 0.098 | 0.074 | 0.084 | 0.069 |
| OLMo-2-1124-7B | Inflection | 0.118 | 0.363 | 0.470 | 0.426 | 0.483 |
| | Lexeme | 0.007 | 0.088 | 0.121 | 0.109 | 0.095 |
| OLMo-2-1124-7B-Instruct | Inflection | 0.118 | 0.371 | 0.476 | 0.434 | 0.493 |
| | Lexeme | 0.007 | 0.094 | 0.122 | 0.119 | 0.103 |
| Pythia-6.9B | Inflection | -0.025 | 0.301 | 0.404 | 0.444 | 0.451 |
| | Lexeme | -0.005 | 0.064 | 0.096 | 0.110 | 0.113 |
| Pythia-6.9B-Tulu | Inflection | -0.021 | 0.294 | 0.394 | 0.438 | 0.445 |
| | Lexeme | -0.005 | 0.057 | 0.092 | 0.112 | 0.126 |
| Qwen2.5-1.5B | Inflection | 0.085 | 0.325 | 0.456 | 0.455 | 0.549 |
| | Lexeme | -0.007 | 0.053 | 0.072 | 0.082 | 0.062 |
| Qwen2.5-1.5B-Instruct | Inflection | 0.046 | 0.337 | 0.460 | 0.440 | 0.502 |
| | Lexeme | -0.012 | 0.064 | 0.082 | 0.082 | 0.058 |

## Table 6: Probing Results for Chinese

### (a) Accuracy (Linear Probes)

| Model | Task | 0% | 25% | 50% | 75% | 100% |
|---|---|---|---|---|---|---|
| Goldfish Chinese (Chinese) | Inflection | 0.911 | 0.928 | 0.944 | 0.942 | 0.941 |
| | Lexeme | 0.972 | 0.941 | 0.887 | 0.824 | 0.751 |
| Qwen2.5-1.5B (Chinese) | Inflection | 0.898 | 0.948 | 0.949 | 0.950 | 0.946 |
| | Lexeme | 0.883 | 0.905 | 0.735 | 0.743 | 0.667 |
| Qwen2.5-1.5B-Instruct (Chinese) | Inflection | 0.897 | 0.948 | 0.949 | 0.950 | 0.948 |
| | Lexeme | 0.883 | 0.907 | 0.729 | 0.748 | 0.678 |
| Qwen2.5-7B (Chinese) | Inflection | 0.893 | 0.957 | 0.951 | 0.956 | 0.950 |
| | Lexeme | 0.883 | 0.983 | 0.844 | 0.828 | 0.776 |
| Qwen2.5-7B-Instruct (Chinese) | Inflection | 0.893 | 0.957 | 0.950 | 0.956 | 0.949 |
| | Lexeme | 0.883 | 0.981 | 0.839 | 0.823 | 0.773 |
| mT5-Base (Chinese) | Inflection | 0.901 | 0.933 | 0.945 | 0.941 | 0.943 |
| | Lexeme | 0.846 | 0.919 | 0.863 | 0.757 | 0.727 |

### (b) Selectivity (Linear Probes)

| Model | Task | 0% | 25% | 50% | 75% | 100% |
|---|---|---|---|---|---|---|
| Goldfish Chinese (Chinese) | Inflection | 0.223 | 0.292 | 0.346 | 0.356 | 0.391 |
| | Lexeme | -0.000 | -0.001 | -0.002 | -0.001 | -0.003 |
| Qwen2.5-1.5B (Chinese) | Inflection | 0.122 | 0.345 | 0.429 | 0.412 | 0.441 |
| | Lexeme | -0.000 | -0.000 | -0.002 | -0.002 | -0.003 |
| Qwen2.5-1.5B-Instruct (Chinese) | Inflection | 0.122 | 0.345 | 0.430 | 0.412 | 0.437 |
| | Lexeme | -0.000 | -0.001 | -0.003 | -0.001 | -0.002 |
| Qwen2.5-7B (Chinese) | Inflection | 0.047 | 0.250 | 0.387 | 0.386 | 0.408 |
| | Lexeme | -0.000 | 0.000 | -0.001 | 0.000 | -0.000 |
| Qwen2.5-7B-Instruct (Chinese) | Inflection | 0.048 | 0.250 | 0.392 | 0.387 | 0.411 |
| | Lexeme | -0.000 | -0.000 | -0.001 | 0.000 | -0.001 |
| mT5-Base (Chinese) | Inflection | 0.123 | 0.186 | 0.274 | 0.321 | 0.348 |
| | Lexeme | 0.001 | 0.000 | -0.001 | 0.000 | -0.003 |

### (c) Accuracy (MLP Probes)

| Model | Task | 0% | 25% | 50% | 75% | 100% |
|---|---|---|---|---|---|---|
| Goldfish Chinese (Chinese) | Inflection | 0.913 | 0.930 | 0.946 | 0.944 | 0.939 |
| | Lexeme | 0.922 | 0.874 | 0.797 | 0.652 | 0.543 |
| Qwen2.5-1.5B (Chinese) | Inflection | 0.896 | 0.947 | 0.943 | 0.950 | 0.942 |
| | Lexeme | 0.882 | 0.869 | 0.738 | 0.695 | 0.449 |
| Qwen2.5-1.5B-Instruct (Chinese) | Inflection | 0.896 | 0.941 | 0.942 | 0.950 | 0.943 |
| | Lexeme | 0.883 | 0.864 | 0.719 | 0.691 | 0.383 |
| Qwen2.5-7B (Chinese) | Inflection | 0.899 | 0.952 | 0.947 | 0.955 | 0.946 |
| | Lexeme | 0.881 | 0.951 | 0.795 | 0.749 | 0.471 |
| Qwen2.5-7B-Instruct (Chinese) | Inflection | 0.900 | 0.952 | 0.948 | 0.955 | 0.943 |
| | Lexeme | 0.881 | 0.950 | 0.791 | 0.750 | 0.475 |
| mT5-Base (Chinese) | Inflection | 0.907 | 0.938 | 0.947 | 0.942 | 0.948 |
| | Lexeme | 0.841 | 0.796 | 0.658 | 0.587 | 0.661 |

### (d) Selectivity (MLP Probes)

| Model | Task | 0% | 25% | 50% | 75% | 100% |
|---|---|---|---|---|---|---|
| Goldfish Chinese (Chinese) | Inflection | 0.072 | 0.153 | 0.198 | 0.252 | 0.315 |
| | Lexeme | -0.001 | -0.039 | -0.049 | -0.064 | -0.047 |
| Qwen2.5-1.5B (Chinese) | Inflection | 0.058 | 0.168 | 0.280 | 0.305 | 0.411 |
| | Lexeme | 0.000 | -0.001 | -0.006 | -0.001 | 0.002 |
| Qwen2.5-1.5B-Instruct (Chinese) | Inflection | 0.059 | 0.164 | 0.290 | 0.301 | 0.411 |
| | Lexeme | 0.000 | 0.002 | -0.012 | -0.000 | -0.002 |
| Qwen2.5-7B (Chinese) | Inflection | 0.061 | 0.136 | 0.287 | 0.312 | 0.388 |
| | Lexeme | -0.001 | -0.001 | -0.007 | -0.008 | 0.000 |
| Qwen2.5-7B-Instruct (Chinese) | Inflection | 0.055 | 0.135 | 0.298 | 0.314 | 0.379 |
| | Lexeme | -0.001 | -0.003 | -0.006 | -0.004 | -0.000 |
| mT5-Base (Chinese) | Inflection | 0.055 | 0.070 | 0.179 | 0.238 | 0.238 |
| | Lexeme | 0.003 | -0.008 | -0.013 | -0.006 | -0.011 |

## Table 7: Probing Results for English

### (a) Accuracy (Linear Probes)

| Model | Task | 0% | 25% | 50% | 75% | 100% |
|---|---|---|---|---|---|---|
| Goldfish English (English) | Inflection | 0.934 | 0.956 | 0.972 | 0.964 | 0.949 |
| | Lexeme | 0.926 | 0.859 | 0.755 | 0.690 | 0.652 |
| Qwen2.5-1.5B (English) | Inflection | 0.913 | 0.969 | 0.959 | 0.961 | 0.930 |
| | Lexeme | 0.845 | 0.799 | 0.610 | 0.654 | 0.599 |
| Qwen2.5-1.5B-Instruct (English) | Inflection | 0.910 | 0.957 | 0.944 | 0.949 | 0.914 |
| | Lexeme | 0.876 | 0.768 | 0.590 | 0.647 | 0.654 |
| Qwen2.5-7B (English) | Inflection | 0.915 | 0.977 | 0.966 | 0.973 | 0.958 |
| | Lexeme | 0.916 | 0.952 | 0.769 | 0.808 | 0.781 |
| Qwen2.5-7B-Instruct (English) | Inflection | 0.915 | 0.977 | 0.964 | 0.973 | 0.957 |
| | Lexeme | 0.916 | 0.950 | 0.791 | 0.810 | 0.781 |
| mT5-Base (English) | Inflection | 0.920 | 0.966 | 0.969 | 0.966 | 0.958 |
| | Lexeme | 0.868 | 0.862 | 0.731 | 0.634 | 0.619 |

### (b) Selectivity (Linear Probes)

| Model | Task | 0% | 25% | 50% | 75% | 100% |
|---|---|---|---|---|---|---|
| Goldfish English (English) | Inflection | 0.341 | 0.437 | 0.522 | 0.549 | 0.556 |
| | Lexeme | 0.017 | 0.073 | 0.096 | 0.100 | 0.110 |
| Qwen2.5-1.5B (English) | Inflection | 0.199 | 0.483 | 0.589 | 0.566 | 0.548 |
| | Lexeme | 0.034 | 0.146 | 0.117 | 0.129 | 0.108 |
| Qwen2.5-1.5B-Instruct (English) | Inflection | 0.189 | 0.474 | 0.581 | 0.541 | 0.503 |
| | Lexeme | 0.061 | 0.143 | 0.113 | 0.129 | 0.112 |
| Qwen2.5-7B (English) | Inflection | 0.059 | 0.383 | 0.538 | 0.533 | 0.523 |
| | Lexeme | -0.006 | 0.098 | 0.121 | 0.131 | 0.099 |
| Qwen2.5-7B-Instruct (English) | Inflection | 0.059 | 0.392 | 0.542 | 0.531 | 0.528 |
| | Lexeme | -0.006 | 0.098 | 0.116 | 0.129 | 0.100 |
| mT5-Base (English) | Inflection | 0.229 | 0.348 | 0.462 | 0.533 | 0.530 |
| | Lexeme | 0.003 | 0.005 | 0.047 | 0.053 | 0.063 |

### (c) Accuracy (MLP Probes)

| Model | Task | 0% | 25% | 50% | 75% | 100% |
|---|---|---|---|---|---|---|
| Goldfish English (English) | Inflection | 0.937 | 0.962 | 0.976 | 0.971 | 0.964 |
| | Lexeme | 0.952 | 0.922 | 0.871 | 0.790 | 0.656 |
| Qwen2.5-1.5B (English) | Inflection | 0.666 | 0.956 | 0.956 | 0.961 | 0.929 |
| | Lexeme | 0.792 | 0.959 | 0.901 | 0.886 | 0.731 |
| Qwen2.5-1.5B-Instruct (English) | Inflection | 0.598 | 0.922 | 0.928 | 0.942 | 0.913 |
| | Lexeme | 0.852 | 0.939 | 0.880 | 0.900 | 0.812 |
| Qwen2.5-7B (English) | Inflection | 0.919 | 0.970 | 0.963 | 0.970 | 0.953 |
| | Lexeme | 0.913 | 0.935 | 0.831 | 0.818 | 0.506 |
| Qwen2.5-7B-Instruct (English) | Inflection | 0.930 | 0.976 | 0.970 | 0.976 | 0.951 |
| | Lexeme | 0.913 | 0.933 | 0.824 | 0.818 | 0.521 |
| mT5-Base (English) | Inflection | NaN | NaN | NaN | NaN | NaN |
| | Lexeme | 0.871 | 0.845 | 0.744 | 0.686 | 0.722 |

### (d) Selectivity (MLP Probes)

| Model | Task | 0% | 25% | 50% | 75% | 100% |
|---|---|---|---|---|---|---|
| Goldfish English (English) | Inflection | 0.131 | 0.272 | 0.371 | 0.438 | 0.493 |
| | Lexeme | 0.006 | -0.021 | 0.005 | 0.031 | 0.050 |
| Qwen2.5-1.5B (English) | Inflection | 0.248 | 0.588 | 0.647 | 0.625 | 0.595 |
| | Lexeme | 0.019 | 0.085 | 0.096 | 0.101 | 0.090 |
| Qwen2.5-1.5B-Instruct (English) | Inflection | 0.187 | 0.571 | 0.629 | 0.608 | 0.549 |
| | Lexeme | 0.069 | 0.090 | 0.097 | 0.100 | 0.087 |
| Qwen2.5-7B (English) | Inflection | 0.051 | 0.282 | 0.456 | 0.457 | 0.507 |
| | Lexeme | -0.008 | 0.039 | 0.070 | 0.073 | 0.061 |
| Qwen2.5-7B-Instruct (English) | Inflection | 0.001 | 0.229 | 0.424 | 0.416 | 0.457 |
| | Lexeme | -0.008 | 0.042 | 0.072 | 0.067 | 0.068 |
| mT5-Base (English) | Inflection | NaN | NaN | NaN | NaN | NaN |
| | Lexeme | -0.012 | -0.023 | 0.003 | 0.011 | 0.026 |

## Table 8: Probing Results for French

### (a) Accuracy (Linear Probes)

| Model | Task | 0% | 25% | 50% | 75% | 100% |
|---|---|---|---|---|---|---|
| Goldfish French (French) | Inflection | 0.924 | 0.959 | 0.976 | 0.970 | 0.963 |
| | Lexeme | 0.888 | 0.813 | 0.714 | 0.665 | 0.619 |
| Qwen2.5-1.5B (French) | Inflection | 0.792 | 0.947 | 0.954 | 0.952 | 0.928 |
| | Lexeme | 0.541 | 0.850 | 0.696 | 0.696 | 0.602 |
| Qwen2.5-1.5B-Instruct (French) | Inflection | 0.792 | 0.945 | 0.951 | 0.949 | 0.925 |
| | Lexeme | 0.541 | 0.845 | 0.687 | 0.690 | 0.611 |
| Qwen2.5-7B (French) | Inflection | 0.793 | 0.966 | 0.965 | 0.964 | 0.945 |
| | Lexeme | 0.541 | 0.943 | 0.801 | 0.769 | 0.714 |
| Qwen2.5-7B-Instruct (French) | Inflection | 0.793 | 0.963 | 0.962 | 0.961 | 0.941 |
| | Lexeme | 0.541 | 0.942 | 0.790 | 0.760 | 0.706 |
| mT5-Base (French) | Inflection | 0.840 | 0.943 | 0.967 | 0.961 | 0.944 |
| | Lexeme | 0.656 | 0.773 | 0.674 | 0.596 | 0.567 |

### (b) Selectivity (Linear Probes)

| Model | Task | 0% | 25% | 50% | 75% | 100% |
|---|---|---|---|---|---|---|
| Goldfish French (French) | Inflection | 0.403 | 0.497 | 0.581 | 0.585 | 0.613 |
| | Lexeme | 0.039 | 0.109 | 0.132 | 0.131 | 0.122 |
| Qwen2.5-1.5B (French) | Inflection | 0.244 | 0.463 | 0.576 | 0.559 | 0.561 |
| | Lexeme | 0.002 | 0.137 | 0.126 | 0.132 | 0.088 |
| Qwen2.5-1.5B-Instruct (French) | Inflection | 0.244 | 0.467 | 0.582 | 0.565 | 0.562 |
| | Lexeme | 0.002 | 0.137 | 0.126 | 0.132 | 0.092 |
| Qwen2.5-7B (French) | Inflection | 0.228 | 0.370 | 0.538 | 0.545 | 0.540 |
| | Lexeme | 0.003 | 0.116 | 0.145 | 0.139 | 0.112 |
| Qwen2.5-7B-Instruct (French) | Inflection | 0.228 | 0.375 | 0.545 | 0.552 | 0.544 |
| | Lexeme | 0.002 | 0.118 | 0.143 | 0.138 | 0.112 |
| mT5-Base (French) | Inflection | 0.248 | 0.386 | 0.495 | 0.548 | 0.530 |
| | Lexeme | -0.006 | 0.027 | 0.045 | 0.046 | 0.037 |

### (c) Accuracy (MLP Probes)

| Model | Task | 0% | 25% | 50% | 75% | 100% |
|---|---|---|---|---|---|---|
| Goldfish French (French) | Inflection | 0.932 | 0.972 | 0.980 | 0.979 | 0.971 |
| | Lexeme | 0.947 | 0.949 | 0.916 | 0.829 | 0.696 |
| Qwen2.5-1.5B (French) | Inflection | 0.789 | 0.956 | 0.965 | 0.960 | 0.929 |
| | Lexeme | 0.536 | 0.910 | 0.845 | 0.788 | 0.360 |
| Qwen2.5-1.5B-Instruct (French) | Inflection | 0.791 | 0.954 | 0.962 | 0.959 | 0.929 |
| | Lexeme | 0.537 | 0.911 | 0.835 | 0.798 | 0.535 |
| Qwen2.5-7B (French) | Inflection | 0.791 | 0.971 | 0.971 | 0.968 | 0.943 |
| | Lexeme | 0.533 | 0.953 | 0.862 | 0.830 | 0.461 |
| Qwen2.5-7B-Instruct (French) | Inflection | 0.791 | 0.968 | 0.969 | 0.965 | 0.939 |
| | Lexeme | 0.534 | 0.949 | 0.851 | 0.823 | 0.467 |
| mT5-Base (French) | Inflection | 0.851 | 0.962 | 0.975 | 0.967 | 0.969 |
| | Lexeme | 0.654 | 0.785 | 0.698 | 0.633 | 0.665 |

### (d) Selectivity (MLP Probes)

| Model | Task | 0% | 25% | 50% | 75% | 100% |
|---|---|---|---|---|---|---|
| Goldfish French (French) | Inflection | 0.131 | 0.289 | 0.372 | 0.443 | 0.486 |
| | Lexeme | -0.016 | 0.006 | 0.032 | 0.072 | 0.084 |
| Qwen2.5-1.5B (French) | Inflection | 0.236 | 0.260 | 0.402 | 0.405 | 0.548 |
| | Lexeme | 0.006 | 0.025 | 0.046 | 0.056 | 0.043 |
| Qwen2.5-1.5B-Instruct (French) | Inflection | 0.235 | 0.263 | 0.411 | 0.410 | 0.552 |
| | Lexeme | 0.002 | 0.030 | 0.048 | 0.055 | 0.064 |
| Qwen2.5-7B (French) | Inflection | 0.227 | 0.198 | 0.394 | 0.415 | 0.516 |
| | Lexeme | -0.001 | 0.025 | 0.065 | 0.070 | 0.027 |
| Qwen2.5-7B-Instruct (French) | Inflection | 0.229 | 0.205 | 0.407 | 0.426 | 0.514 |
| | Lexeme | 0.000 | 0.028 | 0.072 | 0.070 | 0.037 |
| mT5-Base (French) | Inflection | 0.127 | 0.190 | 0.321 | 0.404 | 0.388 |
| | Lexeme | -0.020 | -0.036 | -0.015 | -0.000 | -0.000 |

## Table 9: Probing Results for German

### (a) Accuracy (Linear Probes)

| Model | Task | 0% | 25% | 50% | 75% | 100% |
|---|---|---|---|---|---|---|
| Goldfish German (German) | Inflection | 0.911 | 0.955 | 0.961 | 0.961 | 0.952 |
| | Lexeme | 0.886 | 0.831 | 0.707 | 0.627 | 0.569 |
| Qwen2.5-1.5B (German) | Inflection | 0.744 | 0.929 | 0.931 | 0.930 | 0.911 |
| | Lexeme | 0.479 | 0.865 | 0.707 | 0.690 | 0.569 |
| Qwen2.5-1.5B-Instruct (German) | Inflection | 0.745 | 0.928 | 0.929 | 0.929 | 0.912 |
| | Lexeme | 0.479 | 0.862 | 0.694 | 0.686 | 0.582 |
| Qwen2.5-7B (German) | Inflection | 0.744 | 0.949 | 0.946 | 0.950 | 0.935 |
| | Lexeme | 0.480 | 0.942 | 0.811 | 0.764 | 0.651 |
| Qwen2.5-7B-Instruct (German) | Inflection | 0.760 | 0.958 | 0.954 | 0.958 | 0.938 |
| | Lexeme | 0.480 | 0.943 | 0.801 | 0.757 | 0.646 |
| mT5-Base (German) | Inflection | 0.811 | 0.942 | 0.954 | 0.956 | 0.916 |
| | Lexeme | 0.650 | 0.796 | 0.656 | 0.574 | 0.543 |

### (b) Selectivity (Linear Probes)

| Model | Task | 0% | 25% | 50% | 75% | 100% |
|---|---|---|---|---|---|---|
| Goldfish German (German) | Inflection | 0.418 | 0.503 | 0.593 | 0.622 | 0.645 |
| | Lexeme | -0.001 | 0.057 | 0.088 | 0.088 | 0.069 |
| Qwen2.5-1.5B (German) | Inflection | 0.289 | 0.446 | 0.577 | 0.582 | 0.599 |
| | Lexeme | 0.009 | 0.084 | 0.082 | 0.096 | 0.065 |
| Qwen2.5-1.5B-Instruct (German) | Inflection | 0.289 | 0.450 | 0.577 | 0.583 | 0.602 |
| | Lexeme | 0.009 | 0.082 | 0.082 | 0.098 | 0.069 |
| Qwen2.5-7B (German) | Inflection | 0.286 | 0.357 | 0.546 | 0.574 | 0.597 |
| | Lexeme | 0.009 | 0.065 | 0.104 | 0.113 | 0.085 |
| Qwen2.5-7B-Instruct (German) | Inflection | 0.228 | 0.210 | 0.468 | 0.509 | 0.530 |
| | Lexeme | 0.009 | 0.067 | 0.105 | 0.111 | 0.084 |
| mT5-Base (German) | Inflection | 0.251 | 0.371 | 0.494 | 0.560 | 0.529 |
| | Lexeme | 0.000 | 0.012 | 0.023 | 0.043 | 0.033 |

### (c) Accuracy (MLP Probes)

| Model | Task | 0% | 25% | 50% | 75% | 100% |
|---|---|---|---|---|---|---|
| Goldfish German (German) | Inflection | 0.923 | 0.955 | 0.969 | 0.969 | 0.960 |
| | Lexeme | 0.902 | 0.876 | 0.794 | 0.657 | 0.511 |
| Qwen2.5-1.5B (German) | Inflection | 0.741 | 0.943 | 0.944 | 0.942 | 0.910 |
| | Lexeme | 0.473 | 0.869 | 0.763 | 0.682 | 0.292 |
| Qwen2.5-1.5B-Instruct (German) | Inflection | 0.741 | 0.943 | 0.943 | 0.941 | 0.910 |
| | Lexeme | 0.474 | 0.870 | 0.753 | 0.688 | 0.300 |
| Qwen2.5-7B (German) | Inflection | 0.740 | 0.956 | 0.954 | 0.956 | 0.935 |
| | Lexeme | 0.471 | 0.943 | 0.820 | 0.746 | 0.383 |
| Qwen2.5-7B-Instruct (German) | Inflection | 0.758 | 0.962 | 0.959 | 0.961 | 0.935 |
| | Lexeme | 0.471 | 0.943 | 0.810 | 0.749 | 0.397 |
| mT5-Base (German) | Inflection | 0.820 | 0.956 | 0.954 | 0.952 | 0.939 |
| | Lexeme | 0.641 | 0.710 | 0.563 | 0.486 | 0.530 |

### (d) Selectivity (MLP Probes)

| Model | Task | 0% | 25% | 50% | 75% | 100% |
|---|---|---|---|---|---|---|
| Goldfish German (German) | Inflection | 0.229 | 0.350 | 0.457 | 0.523 | 0.585 |
| | Lexeme | -0.021 | 0.017 | 0.051 | 0.084 | 0.082 |
| Qwen2.5-1.5B (German) | Inflection | 0.292 | 0.292 | 0.457 | 0.492 | 0.604 |
| | Lexeme | 0.008 | 0.022 | 0.054 | 0.060 | 0.035 |
| Qwen2.5-1.5B-Instruct (German) | Inflection | 0.291 | 0.295 | 0.472 | 0.488 | 0.600 |
| | Lexeme | 0.009 | 0.023 | 0.056 | 0.063 | 0.032 |
| Qwen2.5-7B (German) | Inflection | 0.288 | 0.214 | 0.455 | 0.493 | 0.596 |
| | Lexeme | 0.007 | 0.011 | 0.066 | 0.070 | 0.046 |
| Qwen2.5-7B-Instruct (German) | Inflection | 0.227 | 0.111 | 0.382 | 0.424 | 0.532 |
| | Lexeme | 0.008 | 0.015 | 0.073 | 0.077 | 0.049 |
| mT5-Base (German) | Inflection | 0.160 | 0.253 | 0.405 | 0.475 | 0.462 |
| | Lexeme | -0.012 | -0.037 | 0.000 | 0.023 | 0.012 |

## Table 10: Probing Results for Russian

### (a) Accuracy (Linear Probes)

| Model | Task | 0% | 25% | 50% | 75% | 100% |
|---|---|---|---|---|---|---|
| Goldfish Russian (Russian) | Inflection | 0.932 | 0.952 | 0.975 | 0.968 | 0.950 |
| | Lexeme | 0.896 | 0.854 | 0.758 | 0.710 | 0.631 |
| Qwen2.5-1.5B (Russian) | Inflection | 0.850 | 0.966 | 0.966 | 0.962 | 0.933 |
| | Lexeme | 0.315 | 0.896 | 0.739 | 0.720 | 0.598 |
| Qwen2.5-1.5B-Instruct (Russian) | Inflection | 0.850 | 0.965 | 0.964 | 0.960 | 0.932 |
| | Lexeme | 0.315 | 0.893 | 0.725 | 0.714 | 0.600 |
| Qwen2.5-7B (Russian) | Inflection | 0.850 | 0.977 | 0.976 | 0.974 | 0.954 |
| | Lexeme | 0.315 | 0.960 | 0.834 | 0.798 | 0.696 |
| Qwen2.5-7B-Instruct (Russian) | Inflection | 0.858 | 0.977 | 0.974 | 0.980 | 0.953 |
| | Lexeme | 0.315 | 0.959 | 0.821 | 0.785 | 0.680 |
| mT5-Base (Russian) | Inflection | 0.882 | 0.944 | 0.974 | 0.971 | 0.952 |
| | Lexeme | 0.480 | 0.766 | 0.666 | 0.570 | 0.515 |

### (b) Selectivity (Linear Probes)

| Model | Task | 0% | 25% | 50% | 75% | 100% |
|---|---|---|---|---|---|---|
| Goldfish Russian (Russian) | Inflection | 0.505 | 0.568 | 0.663 | 0.675 | 0.688 |
| | Lexeme | 0.024 | 0.115 | 0.157 | 0.175 | 0.170 |
| Qwen2.5-1.5B (Russian) | Inflection | 0.518 | 0.576 | 0.679 | 0.661 | 0.665 |
| | Lexeme | 0.012 | 0.190 | 0.187 | 0.202 | 0.153 |
| Qwen2.5-1.5B-Instruct (Russian) | Inflection | 0.518 | 0.582 | 0.685 | 0.664 | 0.666 |
| | Lexeme | 0.012 | 0.191 | 0.190 | 0.205 | 0.152 |
| Qwen2.5-7B (Russian) | Inflection | 0.517 | 0.504 | 0.670 | 0.671 | 0.658 |
| | Lexeme | 0.011 | 0.165 | 0.218 | 0.222 | 0.183 |
| Qwen2.5-7B-Instruct (Russian) | Inflection | 0.431 | 0.332 | 0.581 | 0.594 | 0.593 |
| | Lexeme | 0.011 | 0.167 | 0.221 | 0.222 | 0.181 |
| mT5-Base (Russian) | Inflection | 0.388 | 0.418 | 0.548 | 0.595 | 0.605 |
| | Lexeme | 0.004 | 0.088 | 0.092 | 0.099 | 0.092 |

### (c) Accuracy (MLP Probes)

| Model | Task | 0% | 25% | 50% | 75% | 100% |
|---|---|---|---|---|---|---|
| Goldfish Russian (Russian) | Inflection | 0.944 | 0.966 | 0.982 | 0.977 | 0.959 |
| | Lexeme | 0.896 | 0.878 | 0.814 | 0.732 | 0.582 |
| Qwen2.5-1.5B (Russian) | Inflection | 0.848 | 0.972 | 0.971 | 0.969 | 0.924 |
| | Lexeme | 0.302 | 0.874 | 0.768 | 0.690 | 0.281 |
| Qwen2.5-1.5B-Instruct (Russian) | Inflection | 0.848 | 0.971 | 0.970 | 0.967 | 0.927 |
| | Lexeme | 0.302 | 0.870 | 0.760 | 0.692 | 0.282 |
| Qwen2.5-7B (Russian) | Inflection | 0.850 | 0.981 | 0.981 | 0.977 | 0.940 |
| | Lexeme | 0.312 | 0.960 | 0.838 | 0.770 | 0.361 |
| Qwen2.5-7B-Instruct (Russian) | Inflection | 0.857 | 0.976 | 0.976 | 0.978 | 0.935 |
| | Lexeme | 0.308 | 0.958 | 0.828 | 0.765 | 0.371 |
| mT5-Base (Russian) | Inflection | 0.883 | 0.956 | 0.978 | 0.974 | 0.971 |
| | Lexeme | 0.448 | 0.571 | 0.470 | 0.397 | 0.437 |

### (d) Selectivity (MLP Probes)

| Model | Task | 0% | 25% | 50% | 75% | 100% |
|---|---|---|---|---|---|---|
| Goldfish Russian (Russian) | Inflection | 0.322 | 0.466 | 0.557 | 0.604 | 0.642 |
| | Lexeme | -0.007 | 0.049 | 0.120 | 0.203 | 0.201 |
| Qwen2.5-1.5B (Russian) | Inflection | 0.524 | 0.451 | 0.575 | 0.594 | 0.662 |
| | Lexeme | 0.004 | 0.080 | 0.122 | 0.162 | 0.108 |
| Qwen2.5-1.5B-Instruct (Russian) | Inflection | 0.523 | 0.457 | 0.589 | 0.594 | 0.660 |
| | Lexeme | 0.004 | 0.085 | 0.133 | 0.166 | 0.101 |
| Qwen2.5-7B (Russian) | Inflection | 0.526 | 0.376 | 0.591 | 0.604 | 0.652 |
| | Lexeme | 0.011 | 0.062 | 0.175 | 0.183 | 0.123 |
| Qwen2.5-7B-Instruct (Russian) | Inflection | 0.432 | 0.211 | 0.493 | 0.512 | 0.583 |
| | Lexeme | 0.008 | 0.065 | 0.184 | 0.193 | 0.125 |
| mT5-Base (Russian) | Inflection | 0.351 | 0.335 | 0.497 | 0.547 | 0.557 |
| | Lexeme | -0.016 | -0.016 | 0.023 | 0.035 | 0.042 |

## Table 11: Probing Results for Turkish

### (a) Accuracy (Linear Probes)

| Model | Task | 0% | 25% | 50% | 75% | 100% |
|---|---|---|---|---|---|---|
| Goldfish Turkish (Turkish) | Inflection | 0.907 | 0.930 | 0.925 | 0.913 | 0.903 |
| | Lexeme | 0.978 | 0.973 | 0.968 | 0.921 | 0.614 |
| Qwen2.5-1.5B (Turkish) | Inflection | 0.719 | 0.869 | 0.847 | 0.849 | 0.831 |
| | Lexeme | 0.530 | 0.959 | 0.868 | 0.815 | 0.796 |
| Qwen2.5-1.5B-Instruct (Turkish) | Inflection | 0.719 | 0.869 | 0.838 | 0.846 | 0.827 |
| | Lexeme | 0.530 | 0.961 | 0.852 | 0.804 | 0.786 |
| Qwen2.5-7B (Turkish) | Inflection | 0.718 | 0.917 | 0.889 | 0.879 | 0.839 |
| | Lexeme | 0.531 | 0.974 | 0.878 | 0.839 | 0.777 |
| Qwen2.5-7B-Instruct (Turkish) | Inflection | 0.718 | 0.911 | 0.875 | 0.874 | 0.836 |
| | Lexeme | 0.531 | 0.975 | 0.854 | 0.803 | 0.731 |
| mT5-Base (Turkish) | Inflection | 0.913 | 0.972 | 0.931 | 0.908 | 0.884 |
| | Lexeme | 0.792 | 0.952 | 0.922 | 0.819 | 0.785 |

### (b) Selectivity (Linear Probes)

| Model | Task | 0% | 25% | 50% | 75% | 100% |
|---|---|---|---|---|---|---|
| Goldfish Turkish (Turkish) | Inflection | 0.345 | 0.407 | 0.491 | 0.492 | 0.500 |
| | Lexeme | -0.002 | 0.008 | 0.074 | 0.089 | 0.144 |
| Qwen2.5-1.5B (Turkish) | Inflection | 0.277 | 0.323 | 0.452 | 0.455 | 0.411 |
| | Lexeme | 0.013 | 0.001 | 0.000 | -0.014 | 0.008 |
| Qwen2.5-1.5B-Instruct (Turkish) | Inflection | 0.277 | 0.319 | 0.447 | 0.452 | 0.416 |
| | Lexeme | 0.013 | 0.003 | 0.004 | -0.009 | 0.009 |
| Qwen2.5-7B (Turkish) | Inflection | 0.275 | 0.293 | 0.472 | 0.462 | 0.417 |
| | Lexeme | 0.014 | -0.002 | 0.003 | -0.010 | -0.021 |
| Qwen2.5-7B-Instruct (Turkish) | Inflection | 0.276 | 0.291 | 0.463 | 0.497 | 0.445 |
| | Lexeme | 0.014 | 0.003 | -0.003 | -0.017 | -0.037 |
| mT5-Base (Turkish) | Inflection | 0.071 | 0.165 | 0.261 | 0.331 | 0.366 |
| | Lexeme | 0.008 | 0.013 | 0.042 | 0.035 | 0.057 |

### (c) Accuracy (MLP Probes)

| Model | Task | 0% | 25% | 50% | 75% | 100% |
|---|---|---|---|---|---|---|
| Goldfish Turkish (Turkish) | Inflection | 0.911 | 0.918 | 0.916 | 0.899 | 0.887 |
| | Lexeme | 0.601 | 0.536 | 0.459 | 0.418 | 0.360 |
| Qwen2.5-1.5B (Turkish) | Inflection | 0.713 | 0.854 | 0.823 | 0.829 | 0.760 |
| | Lexeme | 0.459 | 0.500 | 0.369 | 0.325 | 0.232 |
| Qwen2.5-1.5B-Instruct (Turkish) | Inflection | 0.712 | 0.857 | 0.831 | 0.817 | 0.760 |
| | Lexeme | 0.462 | 0.491 | 0.367 | 0.333 | 0.233 |
| Qwen2.5-7B (Turkish) | Inflection | 0.713 | 0.923 | 0.902 | 0.900 | 0.828 |
| | Lexeme | 0.519 | 0.805 | 0.639 | 0.525 | 0.441 |
| Qwen2.5-7B-Instruct (Turkish) | Inflection | 0.717 | 0.914 | 0.900 | 0.895 | 0.820 |
| | Lexeme | 0.521 | 0.797 | 0.615 | 0.523 | 0.383 |
| mT5-Base (Turkish) | Inflection | 0.884 | 0.915 | 0.875 | 0.836 | 0.839 |
| | Lexeme | 0.503 | 0.395 | 0.300 | 0.257 | 0.312 |

### (d) Selectivity (MLP Probes)

| Model | Task | 0% | 25% | 50% | 75% | 100% |
|---|---|---|---|---|---|---|
| Goldfish Turkish (Turkish) | Inflection | 0.333 | 0.449 | 0.499 | 0.488 | 0.496 |
| | Lexeme | 0.175 | 0.185 | 0.142 | 0.138 | 0.125 |
| Qwen2.5-1.5B (Turkish) | Inflection | 0.299 | 0.355 | 0.438 | 0.500 | 0.432 |
| | Lexeme | -0.001 | 0.045 | 0.093 | 0.049 | 0.045 |
| Qwen2.5-1.5B-Instruct (Turkish) | Inflection | 0.303 | 0.350 | 0.447 | 0.447 | 0.432 |
| | Lexeme | 0.001 | 0.042 | 0.104 | 0.065 | -0.078 |
| Qwen2.5-7B (Turkish) | Inflection | 0.297 | 0.299 | 0.422 | 0.473 | 0.442 |
| | Lexeme | 0.007 | 0.084 | 0.128 | 0.105 | 0.065 |
| Qwen2.5-7B-Instruct (Turkish) | Inflection | 0.303 | 0.297 | 0.437 | 0.492 | 0.432 |
| | Lexeme | 0.012 | 0.087 | 0.147 | 0.103 | 0.061 |
| mT5-Base (Turkish) | Inflection | 0.120 | 0.301 | 0.343 | 0.355 | 0.368 |
| | Lexeme | 0.047 | 0.094 | 0.109 | 0.097 | 0.106 |

# H  DATASET STATISTICS

This section provides statistics and visualizations for the datasets and models used in our experiments across all six languages. Only words containing alphabetic characters and apostrophes were considered.

| Language | Total Words | Unique Lemmas | Unique Forms | Inflection Types | Sentences | Avg. Length |
|---|---|---|---|---|---|---|
| English | 54,816 | 7,848 | 11,720 | 8 | 8,415 | 6.5 |
| Chinese | 44,166 | 11,184 | 11,237 | 4 | 7,892 | 5.8 |
| German | 84,710 | 24,140 | 31,890 | 9 | 9,234 | 7.3 |
| French | 115,847 | 13,804 | 24,485 | 6 | 8,765 | 6.6 |
| Russian | 193,320 | 20,943 | 59,830 | 8 | 10,234 | 7.1 |
| Turkish | 20,881 | 3,776 | 11,680 | 7 | 6,789 | 6.4 |

Table 12: Dataset statistics across all six languages. Russian has the largest dataset and the highest number of unique forms, reflecting its rich inflectional morphology. Turkish has the fewest total words and lemmas, while Chinese has the fewest inflection types.

## H.1  ENGLISH DATASET DETAILS

For the English GUM corpus specifically, the data covers three main syntactic categories: nouns (49.5%), verbs (31.2%), and adjectives (19.4%).

Table 13a shows the distribution of word categories in the English dataset, and Table 13b presents the distribution of inflection categories.

| Category | Count | % |
|---|---|---|
| Noun | 27111 | 49.5 |
| Verb | 17093 | 31.2 |
| Adjective | 10612 | 19.4 |

(a) Word categories

| Inflection | Count | % |
|---|---|---|
| Singular | 19830 | 36.2 |
| Base | 10076 | 18.4 |
| Positive | 9926 | 18.1 |
| Plural | 7281 | 13.3 |
| Past | 5604 | 10.2 |
| 3rd Person | 1413 | 2.6 |
| Comparative | 403 | 0.7 |
| Superlative | 283 | 0.5 |

(b) Inflection categories

| Metric | Value |
|---|---|
| Avg. Words | 6.5 |
| Median Words | 5 |
| Min. Words | 1 |
| Max. Words | 40 |

(c) Sentence length stats

Table 13: Distribution statistics for the English dataset. Table (a) shows syntactic categories, (b) details inflection types, and (c) provides sentence length heuristics.

## H.2  TOKENIZATION STATISTICS

| Model | Tokenizer Type |
|---|---|
| BERT Base/Large | WordPiece |
| DeBERTa V3 Large | SentencePiece |
| GPT-2 variants | BPE |
| Pythia variants | BPE |
| OLMo 2 variants | BPE (tiktoken) |
| Gemma 2 variants | SentencePiece |
| Qwen 2.5 variants | Byte-level BPE |
| Llama 3.1 variants | BPE (tiktoken) |

Table 14: Tokenization strategies used by different model families. BPE means byte-pair encoding.

An important consideration for our analysis is how different models tokenize the words in our dataset. Table 15 shows tokenization statistics across the models we analyze. Encoder-only models

like `BERT` and `DeBERTa` tend to split words into more tokens than decoder-only models like `GPT-2` and `Qwen2`, which may affect how information is encoded across layers.

| Model | Avg. tokens per word | Med. tokens per word | Max tokens per word | Percent multitoken |
|---|---|---|---|---|
| `BERT variants` | 1.11 | 1.0 | 6.0 | 6.95 |
| `DeBERTa-v3-large` | 1.03 | 1.0 | 4.0 | 2.2 |
| `GPT-2 variants` | 1.52 | 1.0 | 5.0 | 42.25 |
| `Pythia-6.9B variants` | 1.48 | 1.0 | 5.0 | 39.1 |
| `OLMo2-7B variants` | 1.43 | 1.0 | 4.0 | 35.9 |
| `Gemma2-2B variants` | 1.19 | 1.0 | 4.0 | 16.55 |
| `Qwen2.5-1.5B variants` | 1.43 | 1.0 | 4.0 | 35.9 |
| `Llama-3.1-8B variants` | 1.43 | 1.0 | 4.0 | 35.85 |

Table 15: Tokenization statistics across different models (English only). Most models have an average of 1.0-1.5 tokens per word and a median of 1, indicating that most words are tokenized as a single unit. However, there is variation in the proportion of words split into multiple tokens. Decoder-only models (*e.g.*, , `GPT-2`, `Pythia`, `Qwen2`, `LLaMA`) split 35-42% of words, while `BERT` and `DeBERTa` variants split fewer words (2-7%). Maximum tokens per word range from 4 to 6 across all models.

## H.3 EFFECTS OF TOKENIZATION

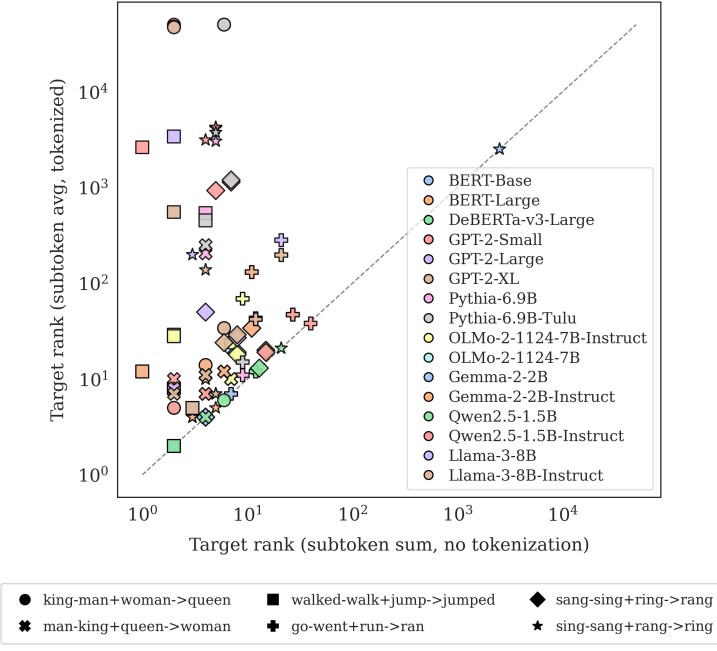

Figure 14: Effect of tokenization strategy on analogy completion rank. Each point corresponds to a model (color) and analogy (shape). The x-axis is the rank using whole-word representations. The y-axis is the rank using tokenized representations. Here, rank means the position of the expected word when all vocabulary words are sorted by similarity to the resulting embedding from vector arithmetic; lower is better. Points above the gray $y=x$ line mean tokenization hurts performance.

Tokenization is an essential component of language modeling. To test how tokenization influences our findings, we use analogy completion tasks in English (*e.g.*, *man:king::woman:?*) and compare two approaches: averaging subtoken embeddings after standard tokenization and summing embeddings from whole-word tokens.

For each approach, we perform vector arithmetic on word representations (*e.g.*, *king - man + woman*). We measure performance by ranking all vocabulary words by cosine similarity to the resulting

representation, and observe how highly the expected word (*e.g.*, *queen*) ranks, with a lower rank indicating better performance.

Whole-word representations markedly outperform averaged subtokens across all models (Figure 14), implying that linguistic regularities are primarily stored in whole-word embeddings rather than compositionally across subtokens. Despite tokenization effects, our classifier results show consistent patterns across models using different tokenizers (see Table 14), indicating robust encoding of lexical and morphological information.

| Model | HuggingFace ID |
|---|---|
| BERT-Base | bert-base-uncased |
| BERT-Large | bert-large-uncased |
| DeBERTa-v3-Large | microsoft/deberta-v3-large |
| mT5-base | google/mt5-base |
| GPT-2-Small | openai-community/gpt2 |
| GPT-2-Large | openai-community/gpt2-large |
| GPT-2-XL | openai-community/gpt2-xl |
| Pythia-6.9B | EleutherAI/pythia-6.9b |
| Pythia-6.9B-Tulu | allenai/open-instruct-pythia-6.9b-tulu |
| OLMo-2-1124-7B | allenai/OLMo-2-1124-7B |
| OLMo-2-1124-7B-Instruct | allenai/OLMo-2-1124-7B-Instruct |
| Gemma-2-2B | google/gemma-2-2b |
| Gemma-2-2B-Instruct | google/gemma-2-2b-it |
| Qwen2.5-1.5B | Qwen/Qwen2.5-1.5B |
| Qwen2.5-1.5B-Instruct | Qwen/Qwen2.5-1.5B-Instruct |
| Qwen2.5-7B | Qwen/Qwen2.5-7B |
| Qwen2.5-7B-Instruct | Qwen/Qwen2.5-7B-Instruct |
| Llama-3.1-8B | meta-llama/Llama-3.1-8B |
| Llama-3.1-8B-Instruct | meta-llama/Llama-3.1-8B-Instruct |
| Goldfish English | goldfish-models/goldfish_eng_latn_1000mb |
| Goldfish Chinese | goldfish-models/goldfish_zho_hans_1000mb |
| Goldfish German | goldfish-models/goldfish_deu_latn_1000mb |
| Goldfish French | goldfish-models/goldfish_fra_latn_1000mb |
| Goldfish Russian | goldfish-models/goldfish_rus_cyrl_1000mb |
| Goldfish Turkish | goldfish-models/goldfish_tur_latn_1000mb |

Table 16: Canonical HuggingFace model IDs used to load models in our study.

# I  ADDITIONAL ANALYSIS

## I.1  INTRINSIC DIMENSIONALITY RESULTS

Intrinsic dimensionality analyses are shown in Figure 15 and Table 17. These illustrate how compression varies across layers and between models.

| Model | $d_{\text{model}}$ | $\text{ID}_{50}$ | | | $\text{ID}_{70}$ | | | $\text{ID}_{90}$ | | |
|---|---|---|---|---|---|---|---|---|---|---|
| | | First | Mid | Final | First | Mid | Final | First | Mid | Final |
| BERT-Base | 768 | 123 | 100 | 88 | 244 | 212 | 192 | 461 | 451 | 446 |
| BERT-Large | 1024 | 138 | 105 | 85 | 286 | 226 | 208 | 567 | 527 | 554 |
| DeBERTa-v3-Large | 1024 | 196 | 133 | 29 | 377 | 299 | 113 | 688 | 635 | 423 |
| GPT-2-Small | 768 | 37 | 1 | 1 | 152 | 1 | 1 | 402 | 1 | 3 |
| GPT-2-Large | 1280 | 24 | 1 | 95 | 172 | 1 | 284 | 583 | 1 | 726 |
| GPT-2-XL | 1600 | 113 | 1 | 118 | 340 | 1 | 356 | 838 | 1 | 914 |
| Pythia-6.9B | 4096 | 391 | 1 | 96 | 865 | 1 | 517 | 1952 | 1 | 1925 |
| Pythia-6.9B-Tulu | 4096 | 390 | 1 | 244 | 862 | 1 | 832 | 1949 | 1 | 2292 |
| OLMo-2-7B | 4096 | 404 | 310 | 41 | 833 | 896 | 299 | 1772 | 2279 | 1550 |
| OLMo-2-7B-Instruct | 4096 | 404 | 358 | 111 | 833 | 974 | 567 | 1772 | 2361 | 1964 |
| Gemma-2-2B | 2304 | 216 | 8 | 11 | 505 | 130 | 70 | 1129 | 794 | 611 |
| Gemma-2-2B-Instruct | 2304 | 222 | 22 | 8 | 520 | 198 | 57 | 1153 | 899 | 572 |
| Qwen-2.5-1.5B | 1536 | 184 | 1 | 9 | 399 | 1 | 50 | 835 | 1 | 452 |
| Qwen-2.5-1.5B-Instruct | 1536 | 184 | 1 | 11 | 394 | 1 | 70 | 820 | 1 | 533 |
| Llama-3.1-8B | 4096 | 373 | 240 | 35 | 789 | 727 | 187 | 1722 | 2051 | 1119 |
| Llama-3.1-8B-Instruct | 4096 | 372 | 215 | 31 | 788 | 664 | 181 | 1722 | 1957 | 1093 |

Table 17: Number of principal-component axes required to reach 50% ($\text{ID}_{50}$), 70% ($\text{ID}_{70}$) and 90% ($\text{ID}_{90}$) explained variance in the first, middle and last layers of each model.

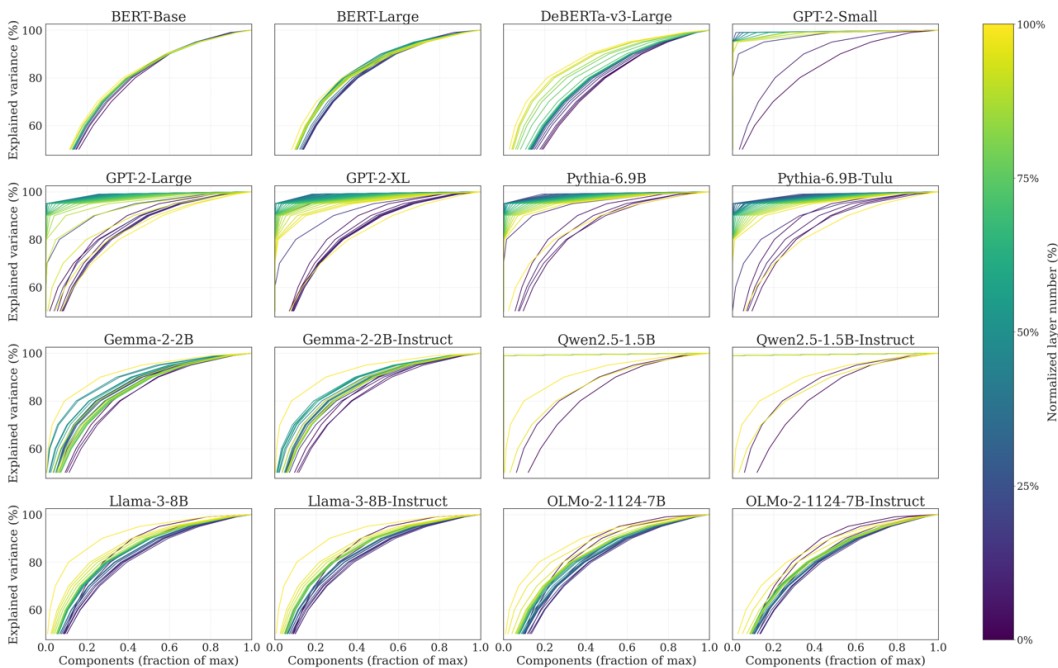

Figure 15: Intrinsic dimensionality curves for all models for English. Each subplot shows the relationship between the percentage of maximum PCA components (x-axis) and the percentage of explained variance (y-axis) across different layers. The color gradient from purple (early layers, 0%) to yellow (late layers, 100%) indicates the relative layer depth within each model. Models like BERT, Gemma, and Llama show similar compression patterns, while GPT-2 variants, Qwen and Pythia exhibit opposite trends in their middle layers.

## I.2  MASSIVE ACTIVATIONS AND OUTLIER DIMENSIONS

We computed the maximum absolute activation, maximum mean (absolute value) per dimension, and maximum standard deviation per dimension across all layers for representative models to understand the low intrinsic dimensionality observed in Table Table 17.

Figures Figures 16–22 show the results. Models like `Qwen2.5-1.5B` and `GPT-2` variants show large maximum activation values. For example, `Qwen2.5-1.5B` reaches maximum absolute activations around 8000, while models like `Llama-3-8B` and `OLMo-1124-7B` show gradual increases across layers, with maximum values only reaching 30-40 in final layers.

This corresponds with the intrinsic dimensionality measurements in Table Table 17. Models with large activations in middle layers correspond to those requiring only 1-2 components to reach 50-90% explained variance at those depths. Models with gradual activation increases correspond to those requiring hundreds of components at all depths. The presence of outlier dimensions with large activations makes the representation anisotropic, with variance concentrated along a small number of directions.

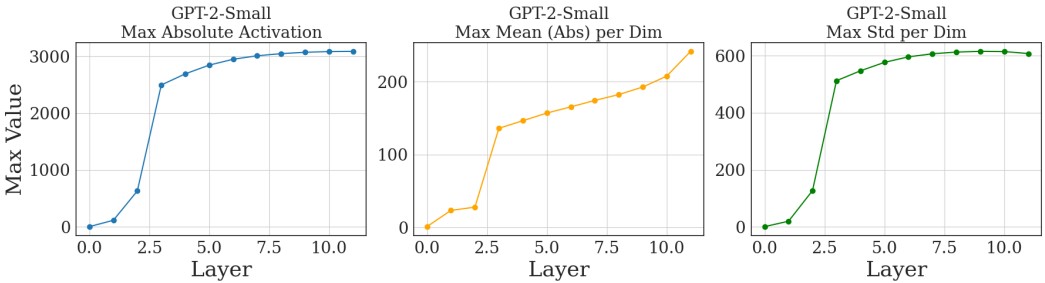

Figure 16: Activation statistics across layers for `GPT-2-Small`.

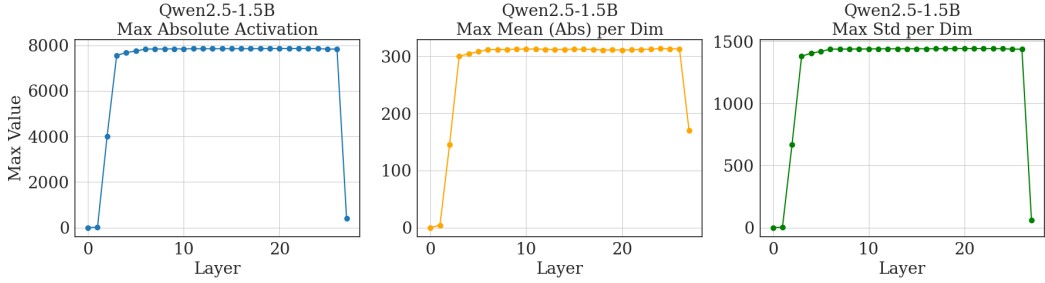

Figure 17: Activation statistics across layers for `Qwen2.5-1.5B`.

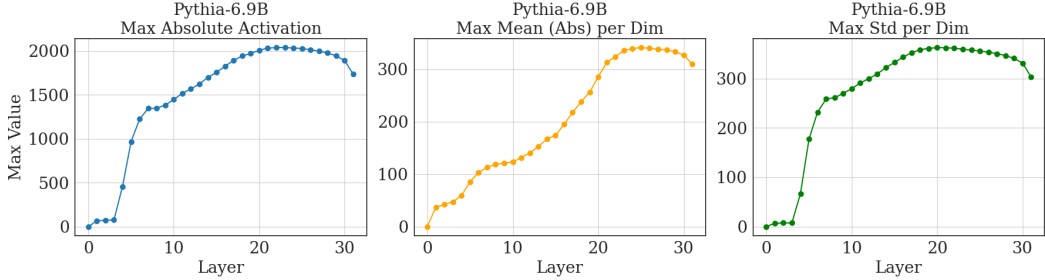

Figure 18: Activation statistics across layers for `Pythia-6.9B`.

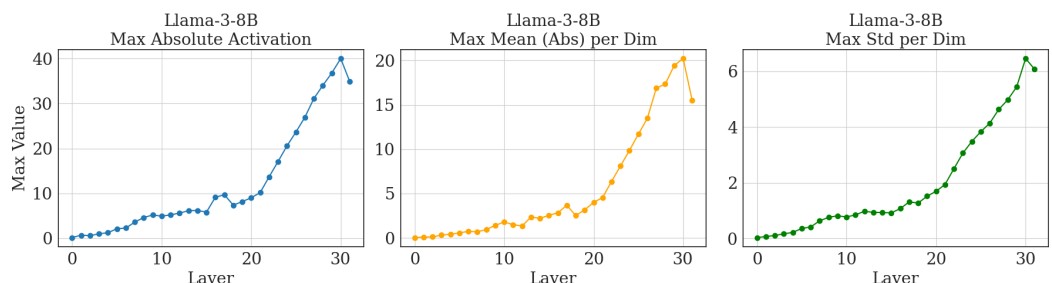

Figure 19: Activation statistics across layers for `Llama-3-8B`.

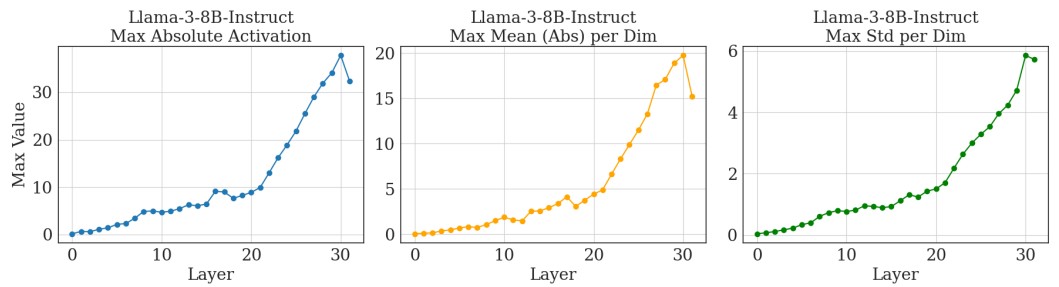

Figure 20: Activation statistics across layers for `Llama-3-8B-Instruct`.

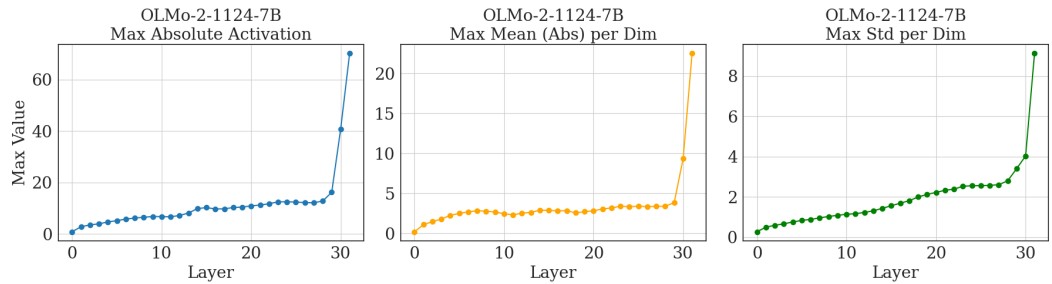

Figure 21: Activation statistics across layers for `OLMo-2-1124-7B`.

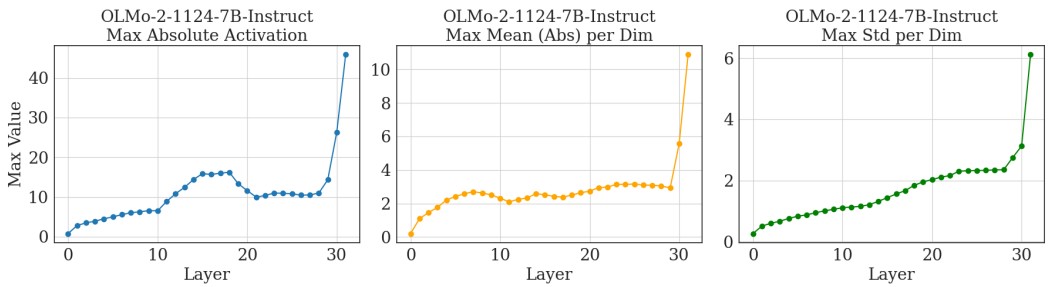

Figure 22: Activation statistics across layers for `OLMo-2-1124-7B-Instruct`.

## I.3   LINEAR SEPARABILITY GAP

Figures 23 and 24 show the linear separability gap for lemma and inflection prediction across models and layers.

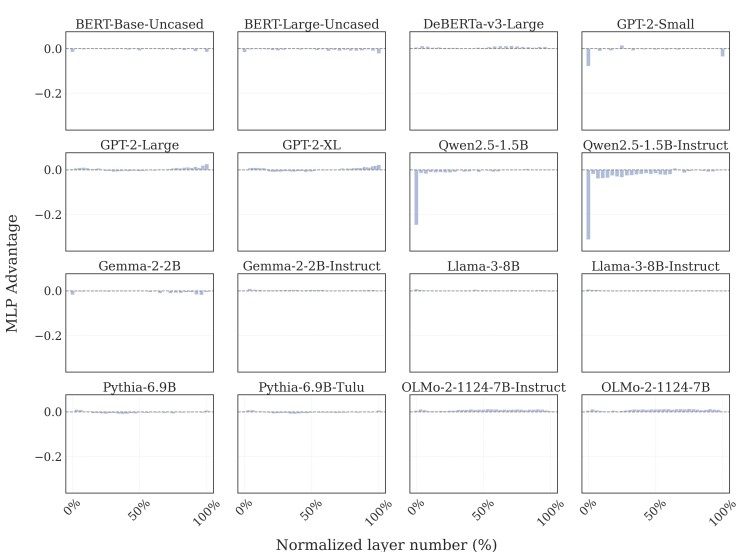

**(a)** Linear separability gap for inflection prediction

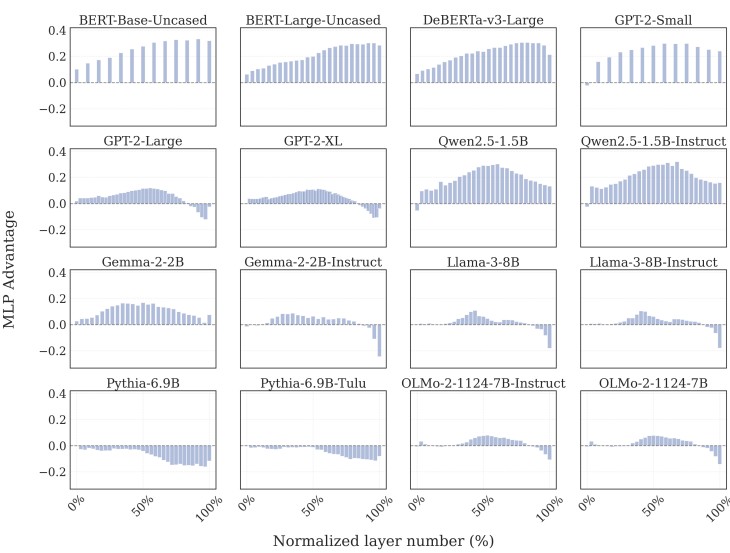

**(b)** Linear separability gap for lemma prediction

Figure 23: Performance advantage of MLP classifiers over linear classifiers (in percentage points) across model layers for English. The linear separability gap measures how much a non-linear transformation improves classifier performance compared to a simple linear mapping. For inflection prediction, the gap is consistently minimal (mostly within ±0.02 percentage points) and sometimes negative, indicating that inflectional features are primarily encoded in a linear fashion throughout the network. By contrast, the linear separability gap for lemma prediction is relatively large (0.1–0.3 percentage points) and positive across most models

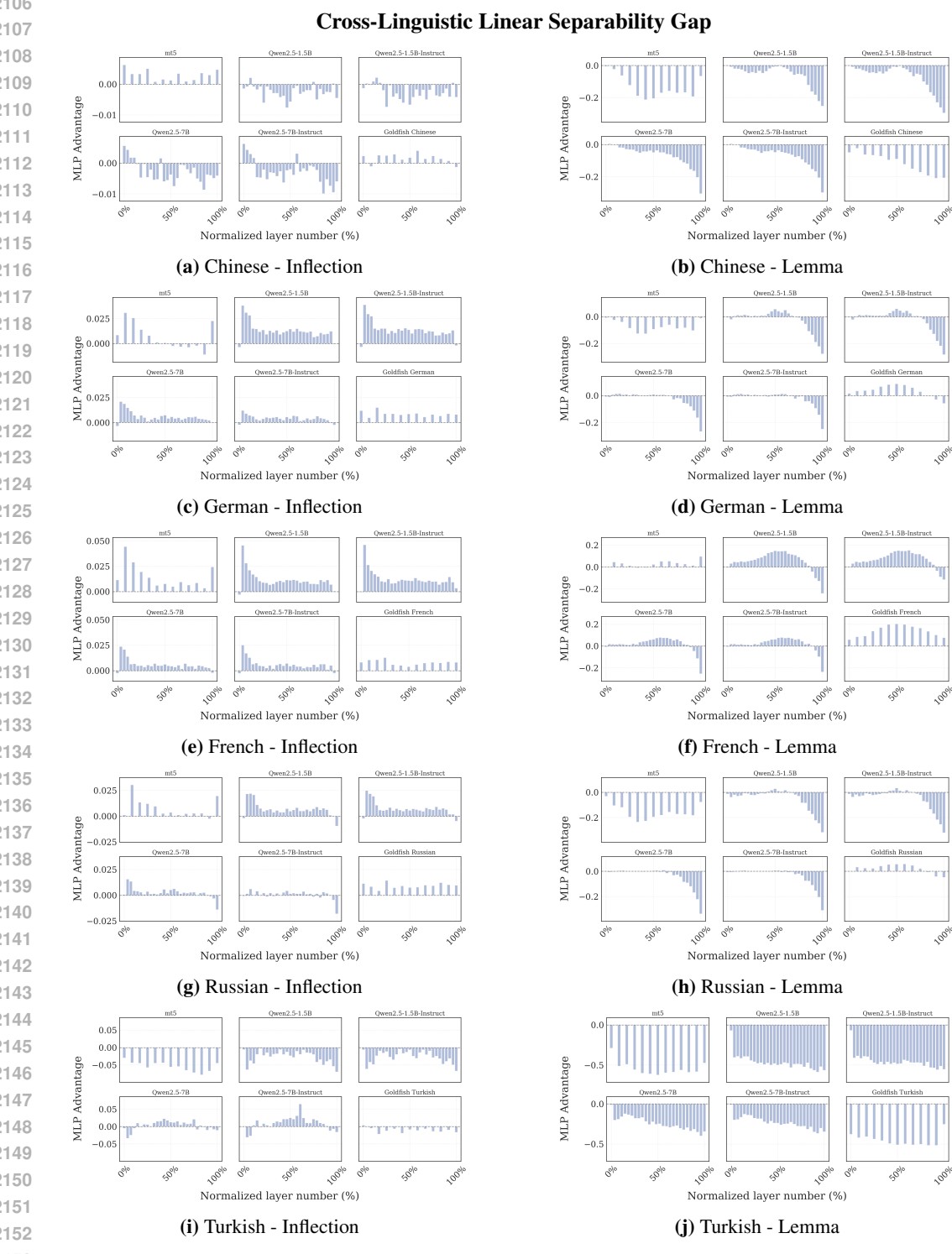

Figure 24: Cross-linguistic linear separability gap showing performance advantage of MLP classifiers over linear classifiers across model layers for five additional languages. For inflectional features, mT5 and Goldfish models show slight positive gaps (indicating modest benefits from non-linear classification), while Qwen2.5 variants show slight negative gaps (indicating linear classifiers are sufficient or superior). For lexical features, all models show negative gaps that are most pronounced in early layers, suggesting that linear regression with regularization consistently outperforms MLPs for lexical classification across all model families and languages.

## I.4  TRAINING DYNAMICS

See Figures 25 and 26 for probing accuracy and selectivity across pretraining checkpoints for `OLMo-2-7B` and `Pythia-6.9B`.

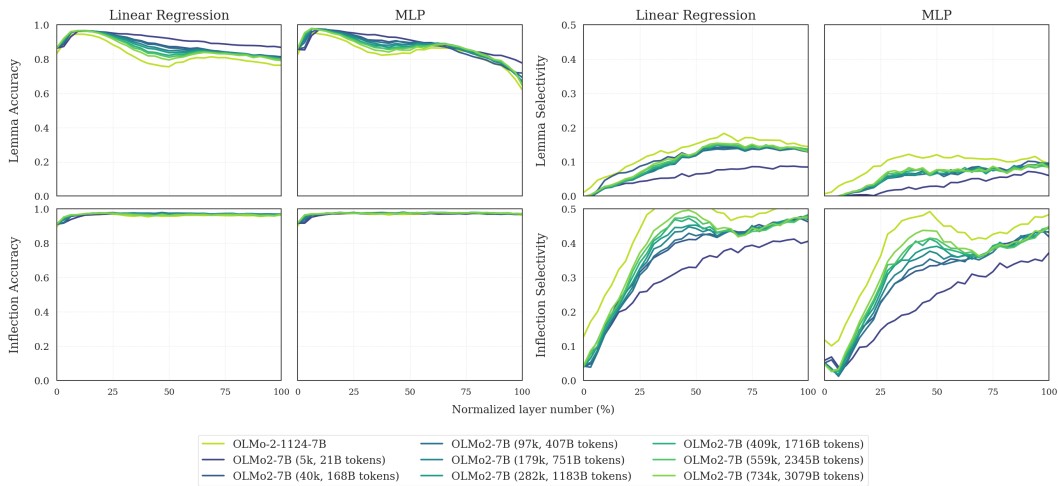

Figure 25: `OLMo-2-7B` **Training Dynamics.** Performance across pretraining checkpoints (5k–734k steps) for English. The full model is 928k steps. Checkpoints are colored from brightest (earliest) to darkest (latest). **Left:** Prediction accuracy for Lemma (top) and Inflection (bottom). Early checkpoints exhibit higher lemma accuracy than later ones, while inflectional accuracy remains flat. **Right:** Selectivity scores for the same tasks. Selectivity generally increases with model depth and training steps, particularly for inflection.

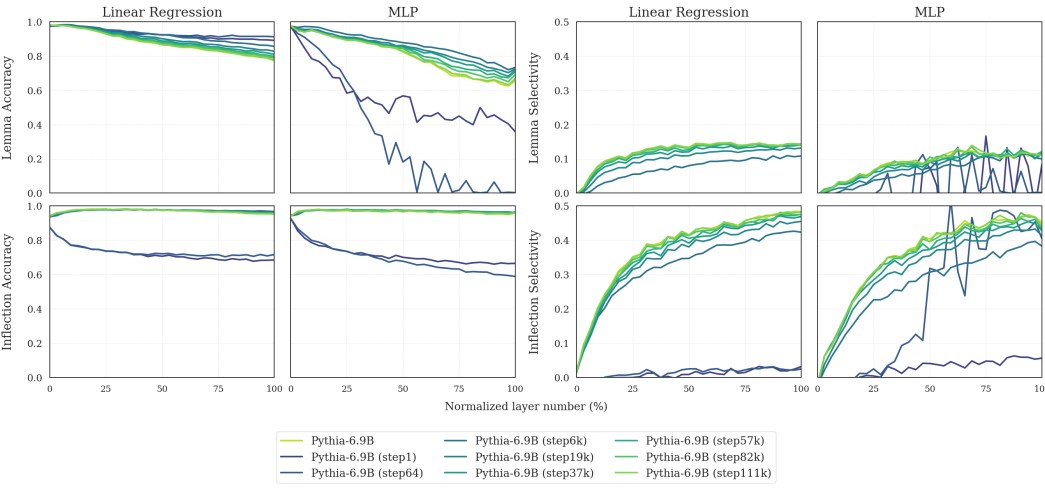

Figure 26: `Pythia-6.9B` **Training Dynamics.** Performance across pretraining checkpoints (step 1–111k) for English. The full model is 143k steps. Checkpoints are colored from brightest (earliest) to darkest (latest). **Left:** Prediction accuracy for Lemma (top) and Inflection (bottom). Lemma accuracy declines both with deeper layers and with more training, whereas inflectional accuracy stays uniformly high. **Right:** Selectivity scores for the same tasks, showing distinct separation between early and late checkpoints in the inflection task.

## J ATTENTION HEAD ANALYSIS

We conducted additional experiments analyzing attention head outputs alongside residual stream representations to understand how different components of transformer models contribute to linguistic encoding.

### J.1 METHODOLOGY

We averaged activations across all attention heads at each layer for `Qwen2.5-1.5B` and `Qwen2.5-1.5B-Instruct` models using the English dataset. We then trained linear regression and MLP classifiers on both attention head outputs and residual stream representations to compare their encoding patterns.

### J.2 RESULTS

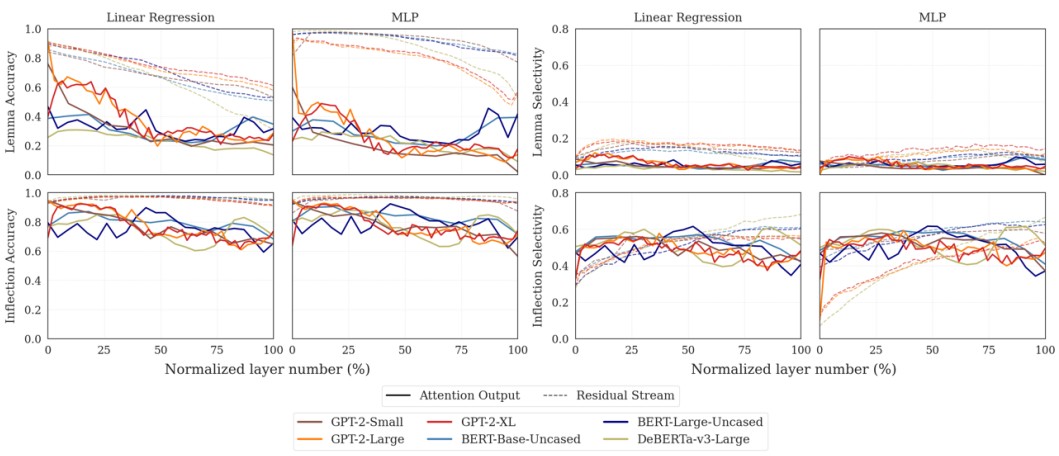

Figure 27: Combined analysis of linguistic task accuracy (left two columns) and classifier selectivity (right two columns) for attention head outputs (solid lines) versus residual stream representations (dashed lines) across BERT and GPT-2 model families. The top row corresponds to Lemma tasks, and the bottom row to Inflection tasks.

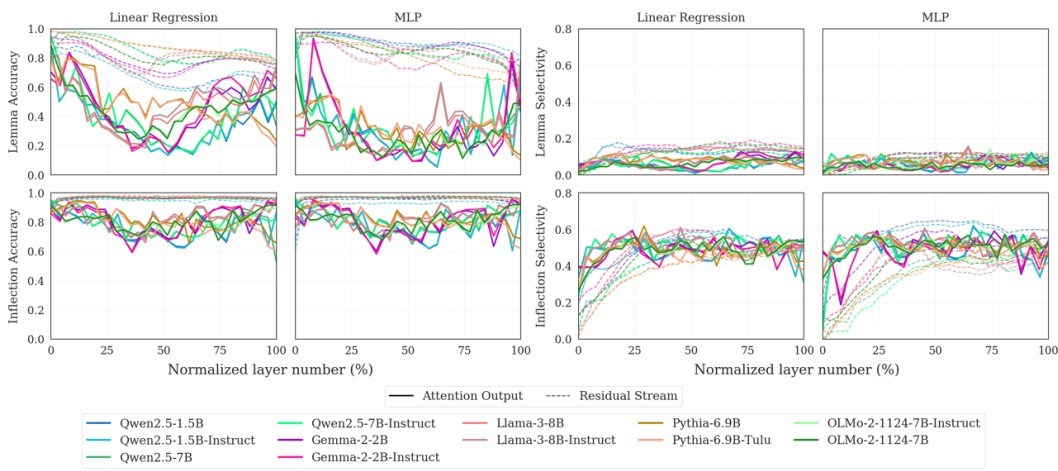

Figure 28: Combined analysis of linguistic task accuracy (left two columns) and classifier selectivity (right two columns) for attention head outputs (solid lines) versus residual stream representations (dashed lines) across contemporary model families. The top row corresponds to Lemma tasks, and the bottom row to Inflection tasks.

## K  STEERING VECTOR ANALYSIS

We conducted steering vector experiments to test whether inflectional representations can be functionally manipulated and to understand model sensitivity to activation interventions.

### K.1  METHODOLOGY

For each inflectional category, we computed steering vectors as:

$$\mathbf{s}_i = \mu_i - \lambda \cdot \frac{1}{|C| - 1} \sum_{j \in C, j \neq i} \mu_j \tag{7}$$

We tested multiple values of $\lambda$ (1, 5, 10, 20, 100) and measured the impact on MLP classifier performance when adding these steering vectors to existing activations for 1000 test words. We evaluated both mean probability change and prediction flip rate across all models.

### K.2  RESULTS

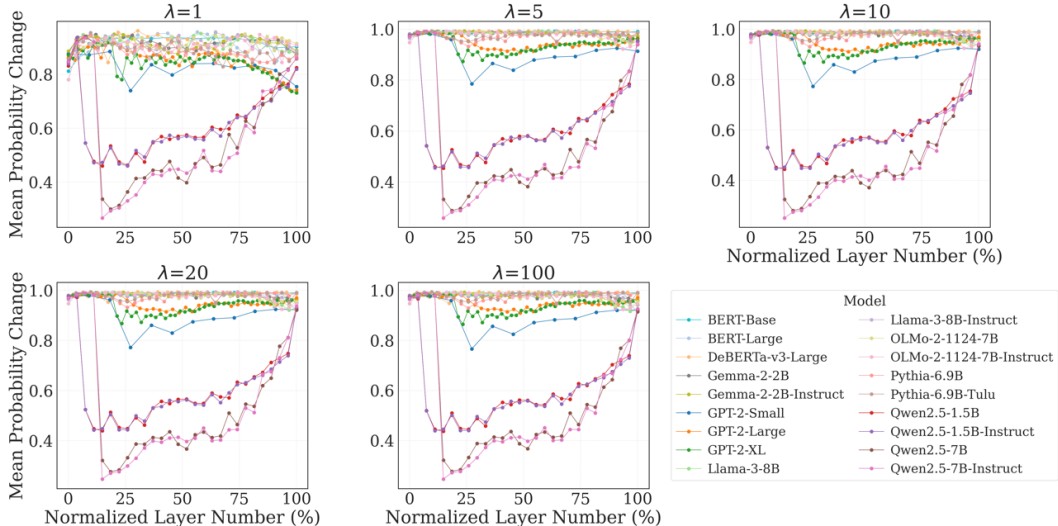

Figure 29: Mean probability change for inflection prediction when applying steering vectors across different $\lambda$ values. Five panels show results for $\lambda \in \{1, 5, 10, 20, 100\}$. All models start with high effectiveness ($\approx$0.9-1.0) at layer 0. Most models maintain stable performance, but Qwen2.5 variants show pronounced sensitivity dips around 10% layer depth before recovering. Higher $\lambda$ values increase steering effectiveness while preserving the overall pattern.

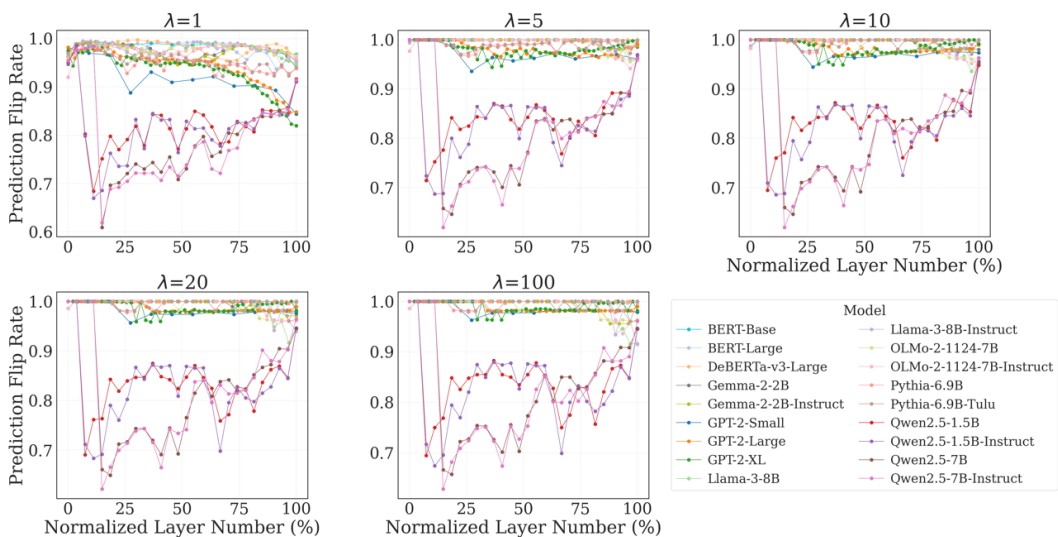

Figure 30: Prediction flip rate when applying steering vectors across different $\lambda$ values. The flip rate patterns mirror the probability change results, with most models maintaining high rates (0.98-1.00) throughout all layers. Qwen2.5 variants show characteristic V-shaped dips to $\approx$0.60-0.70 around 10% layer depth. The consistency across $\lambda$ values suggests that steering effectiveness depends more on model architecture than intervention strength.

## L CLASSIFIER ERROR ANALYSIS

We conducted a detailed error analysis of our classifiers to better understand their performance across different morphological features and languages. See Table 18 through Table 36 for the full results.

| Model | 3rd person (n=249) | Base (n=1,833) | Comparative (n=76) | Past (n=1,003) | Plural (n=1,247) | Positive (n=1,785) | Singular (n=3,587) | Superlative (n=52) |
|---|---|---|---|---|---|---|---|---|
| BERT-Base | 0.960 | 0.965 | 0.817 | 0.967 | 0.983 | 0.946 | 0.971 | 0.759 |
| BERT-Large | 0.956 | 0.964 | 0.861 | 0.968 | 0.982 | 0.950 | 0.971 | 0.768 |
| DeBERTa-v3-Large | 0.938 | 0.974 | 0.831 | 0.961 | 0.986 | 0.954 | 0.977 | 0.706 |
| GPT-2-Small | 0.828 | 0.958 | 0.840 | 0.956 | 0.974 | 0.941 | 0.964 | 0.754 |
| GPT-2-Large | 0.812 | 0.958 | 0.826 | 0.951 | 0.975 | 0.936 | 0.967 | 0.792 |
| GPT-2-XL | 0.817 | 0.959 | 0.813 | 0.948 | 0.977 | 0.940 | 0.968 | 0.788 |
| Pythia-6.9B | 0.886 | 0.972 | 0.904 | 0.964 | 0.989 | 0.957 | 0.977 | 0.907 |
| Pythia-6.9B-Tulu | 0.899 | 0.973 | 0.909 | 0.967 | 0.989 | 0.956 | 0.976 | 0.910 |
| OLMo-2-1124-7B | 0.938 | 0.968 | 0.902 | 0.972 | 0.981 | 0.923 | 0.966 | 0.888 |
| OLMo-2-1124-7B-Instruct | 0.927 | 0.967 | 0.896 | 0.971 | 0.981 | 0.923 | 0.965 | 0.872 |
| Gemma-2-2B | 0.901 | 0.968 | 0.797 | 0.969 | 0.986 | 0.947 | 0.974 | 0.833 |
| Gemma-2-2B-Instruct | 0.913 | 0.966 | 0.863 | 0.973 | 0.988 | 0.938 | 0.972 | 0.872 |
| Qwen2.5-1.5B | 0.856 | 0.950 | 0.802 | 0.942 | 0.972 | 0.919 | 0.957 | 0.688 |
| Qwen2.5-1.5B-Instruct | 0.774 | 0.954 | 0.647 | 0.945 | 0.972 | 0.921 | 0.965 | 0.630 |

Table 18: Breakdown of inflection classification accuracy by morphological feature for each model using linear regression classifiers (English). Inflections are grouped by their morphological features (*e.g.*, Past, Plural, Comparative). For each group, the reported accuracy is the average of accuracies from classifiers trained at each model layer. All accuracy values are on a 0–1 scale. Comparative and superlative forms consistently show the lowest accuracy across all models, reflecting the challenges of these less frequent morphological categories.

| Model | 3rd person (n=249) | Base (n=1,833) | Comparative (n=76) | Past (n=1,003) | Plural (n=1,247) | Positive (n=1,785) | Singular (n=3,587) | Superlative (n=52) |
|---|---|---|---|---|---|---|---|---|
| BERT-Base | 0.973 | 0.969 | 0.910 | 0.972 | 0.989 | 0.959 | 0.974 | 0.939 |
| BERT-Large | 0.967 | 0.970 | 0.910 | 0.973 | 0.988 | 0.961 | 0.975 | 0.931 |
| DeBERTa-v3-Large | 0.954 | 0.976 | 0.925 | 0.966 | 0.989 | 0.962 | 0.979 | 0.867 |
| GPT-2-Small | 0.921 | 0.963 | 0.928 | 0.952 | 0.972 | 0.930 | 0.963 | 0.870 |
| GPT-2-Large | 0.857 | 0.962 | 0.872 | 0.955 | 0.976 | 0.942 | 0.967 | 0.854 |
| GPT-2-XL | 0.921 | 0.963 | 0.928 | 0.952 | 0.972 | 0.930 | 0.963 | 0.870 |
| Pythia-6.9B | 0.932 | 0.972 | 0.921 | 0.961 | 0.982 | 0.949 | 0.971 | 0.886 |
| Pythia-6.9B-Tulu | 0.948 | 0.974 | 0.932 | 0.964 | 0.983 | 0.949 | 0.971 | 0.897 |
| OLMo-2-1124-7B | 0.957 | 0.968 | 0.926 | 0.966 | 0.989 | 0.949 | 0.973 | 0.905 |
| OLMo-2-1124-7B-Instruct | 0.939 | 0.967 | 0.903 | 0.967 | 0.988 | 0.949 | 0.973 | 0.873 |
| Gemma-2-2B | 0.913 | 0.967 | 0.863 | 0.968 | 0.990 | 0.950 | 0.976 | 0.907 |
| Gemma-2-2B-Instruct | 0.930 | 0.970 | 0.878 | 0.975 | 0.989 | 0.946 | 0.974 | 0.906 |
| Qwen2.5-1.5B | 0.882 | 0.948 | 0.822 | 0.943 | 0.974 | 0.927 | 0.957 | 0.736 |
| Qwen2.5-1.5B-Instruct | 0.808 | 0.953 | 0.697 | 0.947 | 0.974 | 0.930 | 0.965 | 0.682 |

Table 19: Breakdown of inflection classification accuracy by morphological feature for each model using Multi-Layer Perceptron (MLP) classifiers (English). Inflections are grouped by their morphological features (*e.g.*, Past, Plural, Comparative). For each group, the reported accuracy is the average of accuracies from classifiers trained at each model layer. All accuracy values are on a 0–1 scale. MLP classifiers provide modest improvements over linear regression, particularly for comparative and superlative forms, though the relative ordering across morphological features remains consistent.

| Model | Noun (n=1,739) | Verb (n=641) | Adjective (n=641) | Adverb (n=23) | Pronoun (n=1) | Preposition (n=1) | Conjunction (n=1) | Interjection (n=1) | Other (n=9) |
|---|---|---|---|---|---|---|---|---|---|
| BERT-Base | 0.636 | 0.737 | 0.609 | 0.805 | 0.292 | 0.000 | 0.585 | 0.000 | 0.902 |
| BERT-Large | 0.684 | 0.777 | 0.653 | 0.826 | 0.580 | 0.154 | 0.662 | 0.065 | 0.897 |
| DeBERTa-v3-Large | 0.592 | 0.737 | 0.585 | 0.723 | 0.440 | 0.077 | 0.438 | 0.081 | 0.866 |
| GPT-2-Small | 0.631 | 0.789 | 0.612 | 0.813 | 0.542 | 0.000 | 0.415 | 0.033 | 0.896 |
| GPT-2-Large | 0.691 | 0.810 | 0.688 | 0.847 | 0.853 | 0.174 | 0.267 | 0.115 | 0.912 |
| GPT-2-XL | 0.713 | 0.827 | 0.708 | 0.847 | 0.724 | 0.222 | 0.311 | 0.241 | 0.899 |
| Pythia-6.9B | 0.856 | 0.926 | 0.836 | 0.926 | 0.938 | 0.443 | 0.566 | 0.488 | 0.934 |
| Pythia-6.9B-Tulu | 0.864 | 0.930 | 0.843 | 0.930 | 0.923 | 0.514 | 0.651 | 0.476 | 0.936 |
| OLMo-2-1124-7B | 0.798 | 0.875 | 0.794 | 0.913 | 0.697 | 0.339 | 0.363 | 0.495 | 0.913 |
| OLMo-2-1124-7B-Instruct | 0.798 | 0.868 | 0.792 | 0.902 | 0.606 | 0.339 | 0.331 | 0.495 | 0.910 |
| Gemma-2-2B | 0.757 | 0.869 | 0.736 | 0.876 | 0.667 | 0.179 | 0.205 | 0.288 | 0.891 |
| Gemma-2-2B-Instruct | 0.749 | 0.844 | 0.742 | 0.872 | 0.620 | 0.137 | 0.152 | 0.247 | 0.912 |
| Qwen2.5-1.5B | 0.652 | 0.801 | 0.650 | 0.828 | 0.526 | 0.082 | 0.223 | 0.068 | 0.867 |
| Qwen2.5-1.5B-Instruct | 0.642 | 0.800 | 0.632 | 0.831 | 0.544 | 0.082 | 0.245 | 0.068 | 0.877 |
| Llama-3.1-8B | 0.776 | 0.882 | 0.771 | 0.887 | 0.831 | 0.286 | 0.396 | 0.321 | 0.911 |
| Llama-3.1-8B-Instruct | 0.796 | 0.892 | 0.788 | 0.896 | 0.908 | 0.300 | 0.443 | 0.357 | 0.917 |

Table 20: Breakdown of lemma classification accuracy by Part of Speech (POS) for each model using linear regression classifiers (English). Lemmas are grouped by their POS tags (*e.g.*, Noun, Verb, Adjective). For each group, the reported accuracy is the average of accuracies from classifiers trained at each model layer. All accuracy values are on a 0–1 scale. Performance varies significantly with frequency: frequent categories like nouns and verbs achieve higher accuracy, while infrequent categories like pronouns and prepositions show lower performance due to limited training examples.

| Model | Noun (n=1,739) | Verb (n=641) | Adjective (n=641) | Adverb (n=23) | Pronoun (n=1) | Preposition (n=1) | Conjunction (n=1) | Interjection (n=1) | Other (n=9) |
|---|---|---|---|---|---|---|---|---|---|
| BERT-Base | 0.775 | 0.831 | 0.748 | 0.873 | 0.458 | 0.125 | 0.756 | 0.267 | 0.898 |
| BERT-Large | 0.813 | 0.863 | 0.785 | 0.884 | 0.540 | 0.231 | 0.725 | 0.323 | 0.897 |
| DeBERTa-v3-Large | 0.689 | 0.803 | 0.682 | 0.802 | 0.700 | 0.115 | 0.662 | 0.242 | 0.861 |
| GPT-2-Small | 0.678 | 0.792 | 0.665 | 0.765 | 0.042 | 0.000 | 0.610 | 0.000 | 0.830 |
| GPT-2-Large | 0.754 | 0.837 | 0.755 | 0.827 | 0.347 | 0.188 | 0.596 | 0.385 | 0.871 |
| GPT-2-XL | 0.774 | 0.844 | 0.771 | 0.827 | 0.561 | 0.232 | 0.561 | 0.431 | 0.860 |
| Pythia-6.9B | 0.774 | 0.856 | 0.768 | 0.862 | 0.554 | 0.229 | 0.528 | 0.310 | 0.868 |
| Pythia-6.9B-Tulu | 0.818 | 0.880 | 0.803 | 0.887 | 0.554 | 0.343 | 0.613 | 0.381 | 0.889 |
| OLMo-2-1124-7B | 0.818 | 0.877 | 0.828 | 0.896 | 0.727 | 0.290 | 0.734 | 0.505 | 0.885 |
| OLMo-2-1124-7B-Instruct | 0.822 | 0.874 | 0.829 | 0.897 | 0.667 | 0.306 | 0.750 | 0.473 | 0.886 |
| Gemma-2-2B | 0.763 | 0.860 | 0.763 | 0.881 | 0.574 | 0.125 | 0.443 | 0.182 | 0.880 |
| Gemma-2-2B-Instruct | 0.777 | 0.846 | 0.785 | 0.882 | 0.580 | 0.137 | 0.400 | 0.299 | 0.875 |
| Qwen2.5-1.5B | 0.747 | 0.838 | 0.742 | 0.811 | 0.228 | 0.131 | 0.628 | 0.164 | 0.857 |
| Qwen2.5-1.5B-Instruct | 0.749 | 0.840 | 0.738 | 0.818 | 0.211 | 0.098 | 0.564 | 0.123 | 0.860 |
| Llama-3.1-8B | 0.798 | 0.879 | 0.807 | 0.886 | 0.800 | 0.214 | 0.679 | 0.393 | 0.882 |
| Llama-3.1-8B-Instruct | 0.824 | 0.893 | 0.826 | 0.895 | 0.831 | 0.257 | 0.689 | 0.429 | 0.887 |

Table 21: Breakdown of lemma classification accuracy by Part of Speech (POS) for each model using Multi-Layer Perceptron (MLP) classifiers (English). Lemmas are grouped by their POS tags (*e.g.*, Noun, Verb, Adjective). For each group, the reported accuracy is the average of accuracies from classifiers trained at each model layer. All accuracy values are on a 0–1 scale. MLP classifiers provide consistent improvements over linear regression across all POS categories, though the frequency-dependent performance patterns persist.

| Model | Linear Regression | | | | MLP | | | |
|---|---|---|---|---|---|---|---|---|
| | Positive (n=300) | Base (n=2,074) | Plural (n=3) | Singular (n=3,947) | Positive (n=300) | Base (n=2,074) | Plural (n=3) | Singular (n=3,947) |
| mT5-Base | 0.739 | 0.913 | 0.436 | 0.962 | 0.783 | 0.919 | 0.231 | 0.961 |
| Qwen2.5-1.5B | 0.785 | 0.929 | 0.034 | 0.969 | 0.801 | 0.924 | 0.092 | 0.967 |
| Qwen2.5-1.5B-Instruct | 0.779 | 0.925 | 0.034 | 0.964 | 0.803 | 0.923 | 0.057 | 0.967 |
| Qwen2.5-7B | 0.824 | 0.937 | 0.310 | 0.970 | 0.828 | 0.929 | 0.310 | 0.969 |
| Qwen2.5-7B-Instruct | 0.819 | 0.936 | 0.299 | 0.970 | 0.823 | 0.928 | 0.276 | 0.969 |
| Goldfish Chinese | 0.793 | 0.912 | 0.000 | 0.958 | 0.816 | 0.915 | 0.000 | 0.957 |

Table 22: Breakdown of inflection classification accuracy for each model by inflection type using Linear Regression and Multi-Layer Perceptron (MLP) classifiers (Chinese). Accuracies are calculated over all examples for a given group across all layers. Counts (n) are derived from a single representative layer for each group. All accuracy values are on a 0–1 scale.

| Model | Noun (n=1,179) | Verb (n=564) | Adjective (n=108) | Adverb (n=22) | Preposition (n=20) | Other (n=50) |
|---|---|---|---|---|---|---|
| mT5-Base | 0.838 | 0.828 | 0.786 | 0.762 | 0.920 | 0.726 |
| Qwen2.5-1.5B | 0.810 | 0.797 | 0.746 | 0.715 | 0.872 | 0.699 |
| Qwen2.5-1.5B-Instruct | 0.813 | 0.799 | 0.748 | 0.713 | 0.873 | 0.700 |
| Qwen2.5-7B | 0.887 | 0.882 | 0.846 | 0.847 | 0.915 | 0.817 |
| Qwen2.5-7B-Instruct | 0.886 | 0.877 | 0.843 | 0.835 | 0.913 | 0.811 |
| Goldfish Chinese | 0.883 | 0.878 | 0.845 | 0.875 | 0.954 | 0.858 |

Table 23: Breakdown of lemma classification accuracy by Part of Speech (POS) for each model, using Linear Regression classifiers (Chinese). Lemmas are grouped by their POS tags (*e.g.*, , Noun, Verb, Adjective). Accuracies are calculated over all examples for a given group across all layers. Counts (n) are derived from a single representative layer for each group. All accuracy values are on a 0–1 scale.

| Model | Noun (n=1,179) | Verb (n=564) | Adjective (n=108) | Adverb (n=22) | Preposition (n=20) | Other (n=50) |
|---|---|---|---|---|---|---|
| mT5-Base | 0.698 | 0.712 | 0.564 | 0.571 | 0.884 | 0.569 |
| Qwen2.5-1.5B | 0.748 | 0.761 | 0.658 | 0.668 | 0.826 | 0.669 |
| Qwen2.5-1.5B-Instruct | 0.735 | 0.745 | 0.643 | 0.643 | 0.814 | 0.655 |
| Qwen2.5-7B | 0.815 | 0.826 | 0.749 | 0.745 | 0.848 | 0.750 |
| Qwen2.5-7B-Instruct | 0.815 | 0.822 | 0.747 | 0.734 | 0.845 | 0.744 |
| Goldfish Chinese | 0.766 | 0.771 | 0.647 | 0.621 | 0.912 | 0.682 |

Table 24: Breakdown of lemma classification accuracy by Part of Speech (POS) for each model, using Multi-Layer Perceptron (MLP) classifiers (Chinese). Lemmas are grouped by their POS tags (*e.g.*, , Noun, Verb, Adjective). Accuracies are calculated over all examples for a given group across all layers. Counts (n) are derived from a single representative layer for each group. All accuracy values are on a 0–1 scale.

| Model | Base (n=417) | 3rd person (n=517) | Positive (n=1,720) | Past (n=839) | Plural (n=1,076) | Superlative (n=52) | Singular (n=3,197) | Comparative (n=141) |
|---|---|---|---|---|---|---|---|---|
| mT5-Base | 0.908 | 0.941 | 0.940 | 0.960 | 0.882 | 0.572 | 0.962 | 0.636 |
| Qwen2.5-1.5B | 0.849 | 0.889 | 0.922 | 0.914 | 0.888 | 0.657 | 0.953 | 0.796 |
| Qwen2.5-1.5B-Instruct | 0.844 | 0.887 | 0.922 | 0.910 | 0.889 | 0.659 | 0.952 | 0.795 |
| Qwen2.5-7B | 0.892 | 0.922 | 0.939 | 0.947 | 0.909 | 0.826 | 0.962 | 0.878 |
| Qwen2.5-7B-Instruct | 0.915 | 0.934 | 0.945 | 0.962 | 0.924 | 0.866 | 0.968 | 0.909 |
| Goldfish German | 0.938 | 0.941 | 0.955 | 0.979 | 0.916 | 0.542 | 0.968 | 0.708 |

Table 25: Breakdown of inflection classification accuracy for each model by inflection type using Linear Regression classifiers (German). Accuracies are calculated over all examples for a given group across all layers. Counts (n) are derived from a single representative layer for each group. All accuracy values are on a 0–1 scale.

| Model | Base (n=417) | 3rd person (n=517) | Positive (n=1,720) | Past (n=839) | Plural (n=1,076) | Superlative (n=52) | Singular (n=3,197) | Comparative (n=141) |
|---|---|---|---|---|---|---|---|---|
| mT5-Base | 0.921 | 0.945 | 0.948 | 0.959 | 0.884 | 0.723 | 0.967 | 0.770 |
| Qwen2.5-1.5B | 0.890 | 0.915 | 0.930 | 0.940 | 0.897 | 0.831 | 0.958 | 0.892 |
| Qwen2.5-1.5B-Instruct | 0.888 | 0.914 | 0.930 | 0.938 | 0.898 | 0.825 | 0.957 | 0.897 |
| Qwen2.5-7B | 0.912 | 0.932 | 0.944 | 0.956 | 0.913 | 0.868 | 0.964 | 0.924 |
| Qwen2.5-7B-Instruct | 0.925 | 0.941 | 0.950 | 0.966 | 0.928 | 0.901 | 0.970 | 0.936 |
| Goldfish German | 0.947 | 0.957 | 0.964 | 0.978 | 0.923 | 0.817 | 0.970 | 0.896 |

Table 26: Breakdown of inflection classification accuracy for each model by inflection type using Multi-Layer Perceptron (MLP) classifiers (German). Accuracies are calculated over all examples for a given group across all layers. Counts (n) are derived from a single representative layer for each group. All accuracy values are on a 0–1 scale.

| Model | Linear Regression | | | | MLP | | | |
|---|---|---|---|---|---|---|---|---|
| | Noun (n=1,262) | Verb (n=395) | Adjective (n=406) | Other (n=12) | Noun (n=1,262) | Verb (n=395) | Adjective (n=406) | Other (n=12) |
| mT5-Base | 0.685 | 0.662 | 0.568 | 0.750 | 0.611 | 0.602 | 0.486 | 0.723 |
| Qwen2.5-1.5B | 0.743 | 0.725 | 0.715 | 0.775 | 0.721 | 0.700 | 0.687 | 0.711 |
| Qwen2.5-1.5B-Instruct | 0.740 | 0.722 | 0.715 | 0.766 | 0.722 | 0.698 | 0.687 | 0.704 |
| Qwen2.5-7B | 0.821 | 0.809 | 0.808 | 0.829 | 0.795 | 0.786 | 0.783 | 0.814 |
| Qwen2.5-7B-Instruct | 0.815 | 0.803 | 0.803 | 0.821 | 0.795 | 0.785 | 0.782 | 0.813 |
| Goldfish German | 0.720 | 0.747 | 0.701 | 0.769 | 0.758 | 0.772 | 0.742 | 0.769 |

Table 27: Breakdown of lemma classification accuracy by Part of Speech (POS) for each model, using Linear Regression and Multi-Layer Perceptron (MLP) classifiers (German). Lemmas are grouped by their POS tags (*e.g.*, , Noun, Verb, Adjective). Accuracies are calculated over all examples for a given group across all layers. Counts (n) are derived from a single representative layer for each group. All accuracy values are on a 0–1 scale.

| Model | Base (n=688) | 3rd person (n=776) | Positive (n=1,833) | Past (n=857) | Plural (n=1,457) | Singular (n=5,169) |
|---|---|---|---|---|---|---|
| mT5-Base | 0.934 | 0.912 | 0.879 | 0.908 | 0.954 | 0.970 |
| Qwen2.5-1.5B | 0.933 | 0.858 | 0.896 | 0.903 | 0.958 | 0.967 |
| Qwen2.5-1.5B-Instruct | 0.930 | 0.852 | 0.893 | 0.898 | 0.958 | 0.966 |
| Qwen2.5-7B | 0.955 | 0.918 | 0.918 | 0.931 | 0.965 | 0.975 |
| Qwen2.5-7B-Instruct | 0.951 | 0.913 | 0.915 | 0.928 | 0.964 | 0.974 |
| Goldfish French | 0.942 | 0.955 | 0.937 | 0.930 | 0.968 | 0.976 |

Table 28: Breakdown of inflection classification accuracy for each model by inflection type using Linear Regression classifiers (French). Accuracies are calculated over all examples for a given group across all layers. Counts (n) are derived from a single representative layer for each group. All accuracy values are on a 0–1 scale.

| Model | Base (n=688) | 3rd person (n=776) | Positive (n=1,833) | Past (n=857) | Plural (n=1,457) | Singular (n=5,169) |
|---|---|---|---|---|---|---|
| mT5-Base | 0.957 | 0.937 | 0.911 | 0.935 | 0.957 | 0.977 |
| Qwen2.5-1.5B | 0.954 | 0.905 | 0.914 | 0.925 | 0.965 | 0.968 |
| Qwen2.5-1.5B-Instruct | 0.954 | 0.902 | 0.911 | 0.924 | 0.965 | 0.968 |
| Qwen2.5-7B | 0.966 | 0.936 | 0.930 | 0.937 | 0.970 | 0.976 |
| Qwen2.5-7B-Instruct | 0.962 | 0.931 | 0.926 | 0.934 | 0.970 | 0.975 |
| Goldfish French | 0.974 | 0.967 | 0.945 | 0.942 | 0.973 | 0.979 |

Table 29: Breakdown of inflection classification accuracy for each model by inflection type using Multi-Layer Perceptron (MLP) classifiers (French). Accuracies are calculated over all examples for a given group across all layers. Counts (n) are derived from a single representative layer for each group. All accuracy values are on a 0–1 scale.

| Model | Linear Regression | | | | MLP | | | |
|---|---|---|---|---|---|---|---|---|
| | Noun (n=1,496) | Verb (n=406) | Adjective (n=358) | Other (n=15) | Noun (n=1,496) | Verb (n=406) | Adjective (n=358) | Other (n=15) |
| mT5-Base | 0.708 | 0.577 | 0.605 | 0.799 | 0.755 | 0.560 | 0.636 | 0.820 |
| Qwen2.5-1.5B | 0.754 | 0.725 | 0.673 | 0.824 | 0.807 | 0.765 | 0.751 | 0.853 |
| Qwen2.5-1.5B-Instruct | 0.750 | 0.718 | 0.671 | 0.820 | 0.824 | 0.776 | 0.768 | 0.869 |
| Qwen2.5-7B | 0.840 | 0.814 | 0.764 | 0.869 | 0.856 | 0.825 | 0.794 | 0.884 |
| Qwen2.5-7B-Instruct | 0.833 | 0.805 | 0.758 | 0.860 | 0.851 | 0.818 | 0.792 | 0.883 |
| Goldfish French | 0.749 | 0.758 | 0.661 | 0.811 | 0.894 | 0.869 | 0.813 | 0.888 |

Table 30: Breakdown of lemma classification accuracy by Part of Speech (POS) for each model, using Linear Regression and Multi-Layer Perceptron (MLP) classifiers (French). Lemmas are grouped by their POS tags (*e.g.*, , Noun, Verb, Adjective). Accuracies are calculated over all examples for a given group across all layers. Counts (n) are derived from a single representative layer for each group. All accuracy values are on a 0–1 scale.

| Model | Base (n=690) | 3rd person (n=456) | Positive (n=1,192) | Past (n=455) | Plural (n=1,333) | Superlative (n=3) | Singular (n=3,316) | Comparative (n=23) |
|---|---|---|---|---|---|---|---|---|
| mT5-Base | 0.930 | 0.978 | 0.975 | 0.957 | 0.877 | 0.000 | 0.977 | 0.799 |
| Qwen2.5-1.5B | 0.925 | 0.946 | 0.974 | 0.938 | 0.923 | 0.015 | 0.966 | 0.835 |
| Qwen2.5-1.5B-Instruct | 0.924 | 0.943 | 0.974 | 0.934 | 0.921 | 0.015 | 0.966 | 0.817 |
| Qwen2.5-7B | 0.949 | 0.966 | 0.979 | 0.958 | 0.948 | 0.094 | 0.977 | 0.872 |
| Qwen2.5-7B-Instruct | 0.951 | 0.974 | 0.980 | 0.970 | 0.948 | 0.080 | 0.980 | 0.918 |
| Goldfish Russian | 0.940 | 0.950 | 0.976 | 0.931 | 0.921 | 0.000 | 0.976 | 0.867 |

Table 31: Breakdown of inflection classification accuracy for each model by inflection type using Linear Regression classifiers (Russian). Accuracies are calculated over all examples for a given group across all layers. Counts (n) are derived from a single representative layer for each group. All accuracy values are on a 0–1 scale.

| Model | Base (n=690) | 3rd person (n=456) | Positive (n=1,192) | Past (n=455) | Plural (n=1,333) | Superlative (n=3) | Singular (n=3,316) | Comparative (n=23) |
|---|---|---|---|---|---|---|---|---|
| mT5-Base | 0.959 | 0.978 | 0.969 | 0.966 | 0.904 | 0.000 | 0.978 | 0.849 |
| Qwen2.5-1.5B | 0.952 | 0.955 | 0.972 | 0.948 | 0.933 | 0.089 | 0.970 | 0.899 |
| Qwen2.5-1.5B-Instruct | 0.950 | 0.954 | 0.973 | 0.947 | 0.933 | 0.089 | 0.969 | 0.911 |
| Qwen2.5-7B | 0.963 | 0.964 | 0.978 | 0.960 | 0.951 | 0.246 | 0.979 | 0.910 |
| Qwen2.5-7B-Instruct | 0.961 | 0.970 | 0.978 | 0.966 | 0.949 | 0.126 | 0.980 | 0.924 |
| Goldfish Russian | 0.965 | 0.972 | 0.978 | 0.948 | 0.943 | 0.000 | 0.977 | 0.934 |

Table 32: Breakdown of inflection classification accuracy for each model by inflection type using Multi-Layer Perceptron (MLP) classifiers (Russian). Accuracies are calculated over all examples for a given group across all layers. Counts (n) are derived from a single representative layer for each group. All accuracy values are on a 0–1 scale.

| Model | Linear Regression | | | | MLP | | | |
|---|---|---|---|---|---|---|---|---|
| | Noun (n=982) | Verb (n=333) | Adjective (n=275) | Other (n=4) | Noun (n=982) | Verb (n=333) | Adjective (n=275) | Other (n=4) |
| mT5-Base | 0.660 | 0.614 | 0.542 | 0.648 | 0.492 | 0.484 | 0.387 | 0.426 |
| Qwen2.5-1.5B | 0.777 | 0.712 | 0.759 | 0.720 | 0.712 | 0.696 | 0.716 | 0.647 |
| Qwen2.5-1.5B-Instruct | 0.772 | 0.704 | 0.756 | 0.720 | 0.710 | 0.689 | 0.717 | 0.643 |
| Qwen2.5-7B | 0.854 | 0.790 | 0.843 | 0.812 | 0.798 | 0.794 | 0.813 | 0.749 |
| Qwen2.5-7B-Instruct | 0.845 | 0.778 | 0.835 | 0.807 | 0.794 | 0.785 | 0.809 | 0.744 |
| Goldfish Russian | 0.795 | 0.723 | 0.764 | 0.676 | 0.810 | 0.776 | 0.759 | 0.657 |

Table 33: Breakdown of lemma classification accuracy by Part of Speech (POS) for each model, using Linear Regression and Multi-Layer Perceptron (MLP) classifiers (Russian). Lemmas are grouped by their POS tags (*e.g.*, , Noun, Verb, Adjective). Accuracies are calculated over all examples for a given group across all layers. Counts (n) are derived from a single representative layer for each group. All accuracy values are on a 0–1 scale.

| Model | Base (n=154) | 3rd person (n=51) | Positive (n=401) | Past (n=168) | Plural (n=33) | Singular (n=632) |
|---|---|---|---|---|---|---|
| mT5-Base | 0.860 | 0.911 | 0.928 | 0.966 | 0.837 | 0.952 |
| Qwen2.5-1.5B | 0.808 | 0.802 | 0.721 | 0.928 | 0.861 | 0.892 |
| Qwen2.5-1.5B-Instruct | 0.809 | 0.817 | 0.720 | 0.941 | 0.878 | 0.899 |
| Qwen2.5-7B | 0.865 | 0.879 | 0.810 | 0.966 | 0.903 | 0.909 |
| Qwen2.5-7B-Instruct | 0.850 | 0.874 | 0.796 | 0.960 | 0.886 | 0.900 |
| Goldfish Turkish | 0.847 | 0.915 | 0.880 | 0.964 | 0.872 | 0.963 |

Table 34: Breakdown of inflection classification accuracy for each model by inflection type using Linear Regression classifiers (Turkish). Accuracies are calculated over all examples for a given group across all layers. Counts (n) are derived from a single representative layer for each group. All accuracy values are on a 0–1 scale.

| Model | Base (n=154) | 3rd person (n=51) | Positive (n=401) | Past (n=168) | Plural (n=33) | Singular (n=632) |
|---|---|---|---|---|---|---|
| mT5-Base | 0.755 | 0.760 | 0.848 | 0.922 | 0.515 | 0.949 |
| Qwen2.5-1.5B | 0.770 | 0.767 | 0.667 | 0.919 | 0.765 | 0.914 |
| Qwen2.5-1.5B-Instruct | 0.762 | 0.757 | 0.662 | 0.917 | 0.766 | 0.913 |
| Qwen2.5-7B | 0.853 | 0.845 | 0.791 | 0.956 | 0.875 | 0.937 |
| Qwen2.5-7B-Instruct | 0.845 | 0.844 | 0.786 | 0.956 | 0.875 | 0.932 |
| Goldfish Turkish | 0.832 | 0.879 | 0.870 | 0.957 | 0.834 | 0.957 |

Table 35: Breakdown of inflection classification accuracy for each model by inflection type using Multi-Layer Perceptron (MLP) classifiers (Turkish). Accuracies are calculated over all examples for a given group across all layers. Counts (n) are derived from a single representative layer for each group. All accuracy values are on a 0–1 scale.

| Model | Linear Regression | | | | MLP | | | |
|---|---|---|---|---|---|---|---|---|
| | Noun (n=221) | Verb (n=53) | Adjective (n=104) | Other (n=13) | Noun (n=221) | Verb (n=53) | Adjective (n=104) | Other (n=13) |
| mT5-Base | 0.866 | 0.823 | 0.921 | 0.955 | 0.215 | 0.421 | 0.374 | 0.637 |
| Qwen2.5-1.5B | 0.834 | 0.805 | 0.866 | 0.877 | 0.307 | 0.439 | 0.449 | 0.693 |
| Qwen2.5-1.5B-Instruct | 0.816 | 0.791 | 0.860 | 0.874 | 0.305 | 0.439 | 0.448 | 0.691 |
| Qwen2.5-7B | 0.871 | 0.850 | 0.900 | 0.904 | 0.595 | 0.625 | 0.695 | 0.809 |
| Qwen2.5-7B-Instruct | 0.850 | 0.823 | 0.883 | 0.885 | 0.579 | 0.613 | 0.678 | 0.800 |
| Goldfish Turkish | 0.929 | 0.904 | 0.940 | 0.969 | 0.386 | 0.550 | 0.477 | 0.808 |

Table 36: Breakdown of lemma classification accuracy by Part of Speech (POS) for each model, using Linear Regression and Multi-Layer Perceptron (MLP) classifiers (Turkish). Lemmas are grouped by their POS tags (*e.g.*, , Noun, Verb, Adjective). Accuracies are calculated over all examples for a given group across all layers. Counts (n) are derived from a single representative layer for each group. All accuracy values are on a 0–1 scale.

