# OpenReview forum: "Echoes of BERT: Do Modern Language Models Rediscover the Classical NLP Pipeline?"
_ICLR.cc/2026/Conference — ICLR 2026 Conference Withdrawn Submission_

### Official Review · Reviewer_HQt2 · 2025-10-25

**Soundness:** 2
**Presentation:** 1
**Contribution:** 2
**Rating:** 2
**Confidence:** 4

**Summary:**

The study employs both linear and non-linear probing techniques to investigate linguistic properties encoded into the intermediate representations across a wide range of mono- and multilingual models. The analysis reveals that modern decoder-only architectures reproduce several processing patterns observed in less-recent encoder-only models. Specifically, earlier layers predominantly capture syntactic features, middle layers encode semantic information, and later layers relate more closely to discourse-level phenomena. Furthermore, the study demonstrates that lexical information is more effectively extracted using non-linear probes, whereas inflectional morphology can be linearly identified across all layers. Overall, the findings suggest that the token prediction objective inherently drives transformer-based architectures to learn shared structural and linguistic properties of language.

**Strengths:**

1. The investigation takes into account large set of models
2. The set of metrics employed is large and meaningful

**Weaknesses:**

1. Short and outdated "Related work" section. There are more recent studies concerning the analysis of linguistic capabilities at different levels in modern LLMs (e.g., Cheng et al.: Emergence of a High-Dimensional Abstraction Phase in Language Transformers, 2025; Skean et al.: Layer by Layer: Uncovering Hidden Representations in Language Models, 2025)
2. lack of strong evidence for some of the highlighted results (see questions)
3. Many analyses dubbed as relevant and reported even in the abstract are discussed too quickly and superficially. For instance, Section 5.2 many potentially interesting properties and observations are just mentioned and then relegated to the appendix. I think the multilingual results are interesting, but they would deserve a discussion on their own. I'd suggest to prioritize some results and dwell on them in a proper manner.

**Questions:**

1. I think the paper would benefit from an explanation, even if very brief and in Appendix, of the tasks that are used throughout the paper (NER, SRC, SPR, ...), together with a paragraph telling which one of them relate to syntax/semantic/discourse properties.
2. conclusions drawn in section 4.1 are not so clear to me by looking at Figures 2, 6-9. For instance, the authors state "Mid-level semantic tasks such as SRL and SPR peak in middle layers" but it often look pretty flat (for linear probes, Fig. 8,9) if not peaking at early/late layers (see LLama, olmo, qwen in Fig 6,7)
3. In Fig 3 the grey columns (Accuracy(?)/Selectivity for Regression/MLP) show values averaged across the layers? This is not clear from the caption or the text. Furthermore, a large selectivity means the extraction of a linguistic information, while a low one implies memorization. How do the authors comment on the fact that only "Entities" has a positive large values, while all the others are close to zero if not significantly negative?

---

> ### Author Response · Authors · 2025-11-25
> **Response to Reviewer HQt2**
>
> Thank you for your detailed review! We address your questions and concerns below.
>
> **Concern 1**
>
> We agree that these are relevant studies. We have added a thorough discussion of more recent works to Section 6 (see revised manuscript). We discuss Cheng et al. (2025) regarding the high-dimensional abstraction phase in early-to-middle layers, and Skean et al. (2025) regarding the task-transferable nature of intermediate representations. We connect these works to our own findings regarding pipeline compression in Section 4.
>
> **Concern 2**
>
> See our response to Questions 2 and 3.
>
> **Concern 3**
>
> Thank you for bringing this up! We have made revisions to the organization of the paper with the intention of providing more depth to our key findings while maintaining a clear narrative. See our general response for a comprehensive picture of the changes made.
>
> **Question 1**
>
> We agree, and have included a discussion of the different linguistic tasks and their organization in the classical NLP pipeline (see Section 2.2 and Appendix D in the revised manuscript). To summarize, the eight tasks are:
>
> - POS (Part-of-Speech tagging): Identifying the grammatical category of each word (noun, verb, adjective, etc.)
> - Constituents (Constituency parsing): Identifying hierarchical phrase structure, determining which words group together to form constituents like noun phrases and verb phrases
> - Dependencies (Dependency parsing): Identifying grammatical relationships between words, such as subject-verb or modifier-head relations
> - Entities (Named Entity Recognition): Identifying and classifying named entities such as persons, organizations, and locations
> - SRL (Semantic Role Labeling): Identifying the semantic roles that entities play in events (e.g., agent, patient, instrument)
> - SPR (Semantic Proto-Roles): Predicting fine-grained semantic properties of arguments (e.g., whether an argument changes state or is sentient)
> - Coreference: Determining which expressions in a text refer to the same real-world entity
> - Relations (Relation Extraction): Identifying semantic relationships between entity mentions across sentences
>
> Following Tenney et al., we group POS, Constituents, and Dependencies as syntactic tasks; Entities, SRL, and SPR as semantic tasks; and Coreference and Relations as discourse-level tasks.
>
> **Question 2**
>
> > conclusions drawn in section 4.1 are not so clear to me
>
> Thank you for highlighting this. We have reorganized Section 4.1 in the revised manuscript to address this issue.
>
> We think that identifying peaks directly from heatmaps (Figures 6-7) is difficult because layer-wise accuracy differences are obscured by the color gradient. Therefore, we adapt the expected layer and center of gravity metrics from Tenney et al. to quantitatively capture where tasks peak across models. The original Section 4 was poorly organized, and we have made made several changes here. See our general response and revised manuscript for details.
>
> We hope this reorganization clarifies the relationship between our figures and conclusions.
>
> **Question 3**
>
> > In Fig 3 the grey columns (Accuracy(?)/Selectivity for Regression/MLP) show values averaged across the layers? This is not clear from the caption or the text.
>
> Thank you for pointing this out! To clarify, the four columns are: linear probe accuracy, linear probe selectivity, MLP probe accuracy and MLP probe selectivity. These values are averaged across layers. We have updated the figure caption to explain this (see the revised manuscript).
>
> > How do the authors comment on the fact that only "Entities" has a positive large values, while all the others are close to zero if not significantly negative?
>
> This is a good question! Regarding the negative selectivity values, these are slightly deceptive because we are averaging across all layers, making them coarse summary statistics. When we look at the full selectivity plots in Figures 8 and 9, we see that it varies substantially across layers for each model and task. As expected, MLP probe selectivity is generally more negative than linear probe selectivity due to increased model capacity.
>
> These negative average values do not invalidate our findings. This is because we avoid over-indexing on the probe accuracy numbers through our depth metrics. Specifically, these are the expected layer and center of gravity metrics we introduce in Section 4.2. Since these metrics compare relative differences in accuracy across layers, they are somewhat robust to absolute accuracy inflation from probe capacity.
>
> We agree this deserves clearer explanation in the main text and have included a discussion of this in Section 4.2 (labeled "Selectivity reveals probe limitations") on the interpretation of the average selectivity values.
>
> Thank you again for taking the time to review this paper and giving valuable feedback! Hopefully the above discussion and revised manuscript address your concerns. Please let us know if you have any questions or clarifications!

---

> > ### Comment · Reviewer_HQt2 · 2025-11-27
> >
> > I thank the authors for the though response and the important updates given to the manuscript. The added details greatly helped improving readability, reproducibility and understanding of their work.
> > Overall, I find that my concerns were properly addressed, however, I still find the paper a bit compressed, some interesting experiments are insufficiently explored and would deserve a more detailed explanation and testing.
> > I will reconsider my score accordingly.
> >
> > However, while re-reading the paper I found another potential issue: looking at fig 11, I noticed that in a lot of cases, the Expected Layer = 1.00. I find it curious, as , by looking at eq 4, this means that $\Delta_l$ is then always systematically 0, meaning that the accuracy is systematically smaller in all the next layers. I wonder whether this could simply be due to the rather obvious jump from the pure embeddings at layer 0 (where the context is still completely absent) to layer 1. How would the exclusion of Layer 0 affect the computation of Center of gravity and Expected Layer?
> >
> > More than a question, it's a sanity check: I suggest the authors to inspect the representations for which the PC1 explains basically all the variance. This is might be due to the presence of outliers dimensions (i.e. dimensions that display a mean, median and standard deviation much larger than the others) or massive activations (i.e. extremely large values for a dimension for some specific entries). These phenomenon can make the distribution very anysotropic and trick the PCA in finding a single relevant dimension.

---

> > > ### Author Response · Authors · 2025-11-29
> > > **Thanks + follow ups**
> > >
> > > Thank you for your response!
> > >
> > > > How would the exclusion of Layer 0 affect the computation of Center of gravity and Expected Layer?
> > >
> > > You are right about the performance jump. We investigated and found that the cases where Expected Layer (EL) = 1.00 are indeed driven by a big performance jump from the embedding layer to the first transformer layer. For example, the BERT Large accuracies for SPR look like: [0.34, 0.79, 0.84, 0.80, ...]. For these tasks and models, the LM is compressing most of the pipeline into this first transformer block as opposed to spreading them out over layers.
> > >
> > > However, after excluding Layer 0, we observe that in most cases, EL remains 1.00. Since EL is designed to weight layers by their marginal gains (the Delta terms in Eq. 4), if the best accuracy doesn't improve after Layer 1, then all subsequent delta values are zero, which means all the weight concentrates on Layer 1 and EL stays at 1.00. We find that this is the case most of the time where EL was already 1.00, so the best accuracy does not improve after Layer 1, and $\Delta_l$ is still systematically 0 even after removing Layer 0. An example of this is Gemma 2 2B for the Constituents task, where there is a jump from Layer 0, but the best accuracy also never changes after Layer 1:
> > >
> > > [0.49041193, 0.56818182, 0.5649858, 0.56569602, 0.56640625, 0.56676136, 0.56569602, 0.55965909, 0.53799716, 0.55575284, 0.51669034, 0.51740057, 0.51917614, 0.53870739, 0.52805398, 0.53764205, 0.51100852, 0.52379261, 0.53231534, 0.53267045, 0.54225852, 0.53125, 0.52272727, 0.54971591, 0.54296875, 0.5234375]
> > >
> > > > … I suggest the authors to inspect the representations for which the PC1 explains basically all the variance.
> > >
> > > This is a great suggestion, and thank you for raising it! We inspected the activation statistics for the cases where only 1-2 components are needed to reach 50% explained variance (as shown in Table 17), and your hypothesis about massive activations/outlier dimensions appears to be correct.
> > >
> > > We looked at the maximum absolute activation, maximum mean (absolute value) per dimension, and maximum standard deviation per dimension across layers for all models (see Appendix I.2, Figures 16-22). Models like Qwen2.5-1.5B and GPT-2 - where PC1 explains all the variance - show much larger activation values across their layers than models like Llama-3-8B and OLMo-1124-7B. For example, Qwen2.5-1.5B reaches a maximum absolute activation of ~8000, while Llama-3-8B reaches a value of only 30, and only in the last few layers.
> > >
> > > Like you pointed out, this connects to work on massive activations and outlier dimensions in LLMs. Sun et al. (2024) [1] showed that a small number of activations in models can be orders of magnitude larger than others and function as bias terms, while Rudman et al. (2023) [2] found that outlier dimensions with high variance can encode task-specific knowledge. Our results suggest that the low intrinsic dimensionality we observe in GPT-2 and Qwen2.5-1.5B is driven by these outlier dimensions, where PC1 captures the dominant direction. We have added a discussion of this in the revised manuscript, see Section 5.2 (L414-424) and Appendix I.2
> > >
> > > **References**
> > >
> > > [1] [Massive Activations in Large Language Models](https://openreview.net/forum?id=F7aAhfitX6) (Sun et al., COLM 2024)
> > >
> > > [2] [Outlier Dimensions Encode Task Specific Knowledge](https://aclanthology.org/2023.emnlp-main.901/) (Rudman et al., EMNLP 2023)

---

### Official Review · Reviewer_EjL9 · 2025-10-26

**Soundness:** 2
**Presentation:** 2
**Contribution:** 2
**Rating:** 4
**Confidence:** 4

**Summary:**

This work aims to answer this question *"Do current decoder LLMs keep the classical NLP pipeline?"* To this end, the authors analyze 25 language models, including both encoder and decoder language models through the probing tests of eight linguistic tasks. The findings indicate that hierarchical organization persists in modern models: early layers capture syntax, middle layers handle semantics and entity-level information, and later layers encode discourse phenomena. After, they study two properties, lexical identity and inflectional morphology, and find that  lexical information concentrates linearly in early layers but becomes increasingly nonlinear deeper in the network, while inflectional information remains linearly accessible throughout all layers.

**Strengths:**

* This work conducts extensive studies across 25 LLMs and 8 tasks
* It is interesting to see how LLMs handle lexical identity and inflectional morphology, which is worth studying
* The steering experiments provide a potential application about how to leverage their findings

**Weaknesses:**

* Some main claims are not well supported by their experimental results. For example, the authors claim that LLMs exhibit a hierarchical NLP pattern in Section 4. However, it is not clear to see the performance change of different tasks across layers in Figure 2. Likewise, Figure 3 is hard to understand and it is tricky to see the layer-level comparisons
* I appreciate that the authors conduct extensive analyses, but this work would be more novel if they can provide more in-depth studies. I am not surprised that all LLMs have classical NLP pipelines since they use the same architecture and training objectives. The section 4 should be more succinct and concise. I find the lexical identity and inflectional morphology are under-explored by prior work. I would suggest focusing on this point and exploring if LLMs are capable of handling the two tasks, where do LLMs store knolwedge for them, and finally how to perform interventions (steering)
* Some figures lack readability, which strongly undermines the contribution of this work (see questions). This work would be improved if they can make revisions about the presentation and readability

**Questions:**

* Figure 2 is not easy to understand and cannot support the authors' claims. In the left figure, *"The top row shows MLP probe accuracy, the bottom row shows linear
probe accuracy"*. However, I cannot find the corresponding top and bottom rows for different probes. In the right figure, *"Pearson correlations between models are computed by vectorizing each model’s per-layer, per-task accuracy grid"*. I can only see the pearson correlations between models instead of layers. More importantly, the authors' observations are not well supported by the figure. For example, *"Mid-level semantic tasks such as SRL and SPR peak in middle layers, while later layers capture discourse-level phenomena like Coreference and Relations. "*. I find that these tasks perform similarly across different layers. Additionally, why do BERT families have low correlations in the right figure? Can the authors clarify this?
* Figure 3 has the same issue. For example, *"discourse-level features are encoded most strongly in later layers"*. However, the expected layer of BERT-base for the task of Relations is 4.43, and the BERT-base model has 12 layers. It is confusing that the authors derive this finding from this figure. Since there are no baselines in this figure, it is tricky to judge how significant the selectivity is here
* Figure 4 and 5 have a low readability. These lines are entangled with each other and hard to distinguish between different language models
* In section 5.2. the authors conducted extensive analyses,  but there are no focal points
* I did not fully understand why the authors use linear and nonlinear probes, which should be clarified in section 2.1

---

> ### Author Response · Authors · 2025-11-25
> **Response to Reviewer EjL9**
>
> Thank you for your thorough review!
>
> **Concern 1**
>
> See our response to Question 1 below.
>
> **Concern 2**
>
> > I am not surprised that all LLMs have classical NLP pipelines since they use the same architecture and training objectives.
>
> We studied 25 LLMs, all of which differ in architecture, training objectives and training data. We analyze encoder-decoder models like BERT and DeBERTa trained with masked language modeling loss, as well as decoder-only models like GPT2, Qwen2.5 and Llama3.1 trained via next token prediction via cross entropy loss. While these models are all transformer-based they differ in architectural details: for example BERT and GPT-2 use absolute positional embeddings, while Qwen2.5 and Llama 3.1 use rotary positional embeddings (RoPE). Some models are pretrained only, while others undergo additional instruction tuning (e.g. Llama-3.1-8B-Instruct, Qwen2.5-7B-Instruct). Finally, the scale of training data also varies dramatically, ranging from just 8 billion tokens for GPT-2 to 18 trillion for Qwen2.5 (see Table 1).
>
> Even if the result is not surprising, we still believe that it is useful for the broader community, since it is the first to provide evidence for what others may have just assumed. Before these experiments, we did not think that models differing in architecture, size and training data would show similarities in their representations. Many of our smaller (and early) models cannot perform most tasks that modern LMs are evaluated on nowadays, whereas our best models handle those tasks well. Despite this, we find that the NLP pipeline persists across models, just compressed into fewer layers for stronger models.
>
> > I would suggest focusing on this point and exploring if LLMs are capable of handling the two tasks, where do LLMs store knolwedge for them, and finally how to perform interventions (steering)
>
> We agree these are important questions. In our Section 5.2, we have highlighted our experiments that directly address them: we probe attention heads and the residual stream, finding that lemma and inflection features are more strongly encoded in the residual stream. We also perform steering experiments for inflectional representations, and find that they can be effectively manipulated across most models and layers. We believe these cover the core of your suggestions, but we are happy to provide further clarifications if there are specific types of interventions you felt were missing.
>
> **Concern 3**
>
> See our response to Question 1 below. We agree that figure readability could be improved and have made several revisions to the manuscript.

---

> > ### Author Response · Authors · 2025-11-25
> > **Response to Reviewer EjL9**
> >
> > **Question 1**
> >
> > > Figure 2 is not easy to understand
> >
> > To clarify: in Figure 2, the three plots on the top row show MLP probe accuracy. The three plots on the bottom row show linear (regression) probe accuracy. In addition to the figure caption, this is also indicated by the colorbar labels on the right hand side of the plots. We have updated the figure caption in the revised manuscript to be clearer about this.
> >
> > We will clarify what the heatmap on the right is showing. As the figure text states, the heatmap displays “Pearson correlations between models”. The phrase “vectorizing each model’s per-layer, per-task accuracy grid”  refers to the following procedure: each model $m$ has an accuracy matrix $A_m \in \mathbb{R}^{T \times L_m}$, where $T$ is the number of tasks (8) and $L_m$ is the number of layers in model $m$. We flatten this matrix into a vector $\vec{a}_m$ and compute the Pearson correlation $\rho(\vec{a}_i, \vec{a}_j)$ between all pairs of models. The resulting 25x25 heatmap shows cross-model consistency: high correlations indicate that two models exhibit similar layer-wise patterns across all tasks.
> >
> > Regarding the BERT families having lower correlations: our interpretation is that they are encoder-only architectures trained with masked language modeling, while most other models in our analysis are decoder-only models trained with causal language modeling. So, it makes sense why the BERT family would have lower correlations with other models.
> >
> > Please let us know if the figures are still unclear!
> >
> > > … and cannot support the authors’ claims.
> >
> > This misunderstanding stems from the different information each figure conveys, which can give the impression that our figures do not support our claims. Figure 2 shows raw accuracy at each layer. Identifying peaks directly from these heatmaps is difficult because the color gradients obscure subtle differences. Our claims about the linguistic hierarchy (e.g., "Mid-level semantic tasks such as SRL and SPR peak in middle layers") are based on Figure 3's expected layer and center of gravity metrics, not visual inspection of Figure 2's heatmaps. These quantitative metrics capture where tasks peak relative to model depth even when raw accuracy appears flat.
> >
> > We think that the original Section 4.1 was poorly organized and did not clearly convey our intended analysis. In the revision, we have restructured the presentation to clearly distinguish between the heatmaps as exploratory visualizations and the summary statistics (Figure 3) as our primary evidence for claims about layer-wise organization.
> >
> > **Question 2**
> >
> > We appreciate you highlighting this issue. Our statement about "later layers" refers to the relative ordering of tasks within each model, not absolute layer positions. In Figure 3, Relations (expected layer 4.43) peaks later than syntax tasks like POS (expected layer 1.44) and semantic tasks like SRL (expected layer 2.43) in BERT-base. The preserved hierarchy means discourse tasks consistently appear after syntax and semantics within each model's depth, even though the absolute layer numbers vary across models with different total depths. We realize this was not clear in the paper and have added a more thorough and precise analysis in Section 4 of the revised manuscript.
> >
> > **Question 3**
> >
> > We agree that Figures 4 and 5 can be improved visually. In our revised manuscript, we have averaged accuracies in Figure 4 based on model category (Encoder-only, Small Decoder, Large Decoder) to reduce line overlap and clutter. We have also adjusted the y-axis limits (starting at 0.4) to make the differences between high-performing models more visible.
> >
> > **Question 4**
> >
> > Thank you for bringing this up! We agree that Section 5.2 lacked clear focal points in the original submission.  In the revision, we have restructured this section to use bolded, descriptive headers that summarize the main takeaway of each analysis (e.g., "Inflection is linearly separable; lemma shows limited nonlinearity" and "Residual streams retain more linguistic information than attention outputs").
> >
> > **Question 5**
> >
> > In Section 2.1, we explain why we use linear and non-linear probes. We say that “The linear probe measures how well information is linearly separable in the representation space, while the non-linear probe tests whether a non-linear decision boundary yields better performance. Comparing these probes allows us to infer whether a property is encoded *linearly* or *non-linearly*.” Therefore the purpose of linear and nonlinear probes is to determine how information is encoded (linearly or non-linear) and where it is encoded.
> >
> > Thank you again for taking the time to review this paper and giving valuable feedback! Hopefully the above discussion and revised manuscript address your concerns. Please let us know if you have any questions or clarifications!

---

### Official Review · Reviewer_W1FQ · 2025-11-02

**Soundness:** 2
**Presentation:** 2
**Contribution:** 2
**Rating:** 2
**Confidence:** 4

**Summary:**

This paper examines 25 transformer-based language models (both MLM and generative), from early architectures like BERT and GPT-2 to modern ones such as Llama-3.1 and Gemma-2, to investigate how linguistic information is represented in the multiple layers of these models. The authors claim that these models exhibit a hierarchical organization, with syntax supposedly concentrated in early layers, semantics and entity-level information in middle layers, and discourse phenomena in later ones. They further suggest that larger models show these patterns emerging earlier in the network. Finally, they report that lexical identity becomes increasingly nonlinear with depth, while inflectional morphology remains linearly accessible, arguing that such regularities persist across architectures and training regimes.

**Strengths:**

- the work is relevant and timely: revisiting and updating BERTology to observe how information is represented in more recent, generative LMs is an interesting challenge
- the authors carried out a very large number of experiments on many different LMs of multiple types (MLM, generative [base], generative [instruct] ) and sizes (from BERT base to LLAMA 3.1 8B)
- the authors commit to publishing a GitHub repository with the code necessary to reproduce their experiments, and display a strong

**Weaknesses:**

- the authors should make sure they use the correct terms. For instance, "lemma" and "lexeme" are not interchangeable. "Inflectional morphology" is a subfield of linguistics, not a task (the authors have "morphological analysis" in mind).
- the eight tasks are not defined/described, and the labels used in Figure 1 to name them are not defined in the text. The reader should not have to read Jindal et al. 2022 to understand what "Universal Proposition English-EWT (SRL)" is, and with which other tasks it should intuitively be grouped (is it a syntactic, semantic or discourse task, as per the authors' classification?)
- more importantly, I fail to understand, based only on Figures 2 and 3, how the authors reach the strong conclusion that starts Section 4.3: "our results demonstrate that modern language models consistently rediscover the classical NLP pipeline […]. […] we find that syntactic information is typically represented in earlier layers, and discourse-level features are encoded most strongly in later layers." The authors do not indicate which tasks they consider syntactic (is morphological analysis a "syntactic" task?), semantic and discourse tasks. The content of the two above-mentioned figures does not seem, to my naked eyes at least, to corroborate their conclusions, nor display consistent behaviours with one another. And results for only 3 models are shown anyway. The fact that other results are given in the appendices is not enough: the main part of a paper must be self-contained, and the authors should have given overall quantitative results demonstrating that their conclusion holds for all modern language models, since this is what Section 4.3 implies.
- steering experiments and their results are difficult to understand when they only take 6 lines of the main paper. It might be the case that the authors have tried to include too much content in a single paper, when a separate paper focused on the content of Section 5 could have make sense.

**Questions:**

- why use Random Forests in section 5, after having explained at the beginning of Section 2 that it is important to only use *simple* probabilistic classifiers as probes?

---

> ### Author Response · Authors · 2025-11-25
> **Response to Reviewer W1FQ**
>
> We appreciate your detailed feedback! Thank you for recognizing the timeliness and relevance of our work. We address your question and concerns below.
>
> **Concern 1 (incorrect terms)**
>
> Thanks for spotting this! We have corrected the inaccurate language (we now use "lemma" instead of "lexeme"), which is visible in our revised manuscript.
>
> **Concern 2 (linguistic tasks not defined)**
>
> This is a good point. We have included definitions for each of the eight tasks in our revised submission, including how they are organized within the “classical NLP pipeline” (see Section 2.2 and Appendix D). To summarize, the eight tasks are:
> - POS (Part-of-Speech tagging): Identifying the grammatical category of each word (noun, verb, adjective, etc.)
>
> - Constituents (Constituency parsing): Identifying hierarchical phrase structure, determining which words group together to form constituents like noun phrases and verb phrases
>
> - Dependencies (Dependency parsing): Identifying grammatical relationships between words, such as subject-verb or modifier-head relations
>
> - Entities (Named Entity Recognition): Identifying and classifying named entities such as persons, organizations, and locations
>
> - SRL (Semantic Role Labeling): Identifying the semantic roles that entities play in events (e.g., agent, patient, instrument)
>
> - SPR (Semantic Proto-Roles): Predicting fine-grained semantic properties of arguments (e.g., whether an argument changes state or is sentient)
>
> - Coreference: Determining which expressions in a text refer to the same real-world entity
>
> - Relations (Relation Extraction): Identifying semantic relationships between entity mentions across sentences
>
> Following [1], we group POS, Constituents, and Dependencies as syntactic tasks; Entities, SRL, and SPR as semantic tasks; and Coreference and Relations as discourse-level tasks. In our in-depth analysis (Section 5) we do not consider how lemma identification and morphological analysis fall into the pipeline.
>
> **Concern 3 (unsupported findings)**
>
> > The content of the two above-mentioned figures does not seem, to my naked eyes at least, to corroborate their conclusions,
>
> We appreciate this feedback and as part of our revisions have tightened the language in this analysis. To clarify: we are making claims about relative layer positions, not absolute positions in the model. When we state that "syntactic information is in earlier layers" and "discourse features are in later layers," we mean that syntax peaks earlier than semantics, which peaks earlier than discourse - within each model.
>
> In the revised manuscript, we have restructured the presentation to clearly distinguish between the heatmaps (Figure 2) as exploratory visualizations for task accuracies and the summary statistics (Figure 3) as our primary evidence for claims about the linguistic hierarchy.
>
> > … nor display consistent behaviours with one another.
>
> This is due to the different information each figure conveys. Figure 2 (heatmaps) is difficult to visually interpret in terms of layer-wise differences, since the color changes are subtle. We address this with Figure 3, which shows expected layer and center of gravity metrics in order to quantify relative differences in where tasks peak. These metrics provide our evidence for hierarchical pipeline ordering even when absolute accuracy appears flat in the heatmaps. The figures are complementary, not contradictory.
>
> In the revised manuscript, we have restructured the presentation to clearly distinguish between the heatmaps (Figure 2) as exploratory visualizations for task accuracies and the summary statistics (Figure 3) as our primary evidence for claims about the linguistic hierarchy.
>
> > the authors should have given overall quantitative results demonstrating that their conclusion holds for all modern language models, since this is what Section 4.3 implies.
>
> We do provide overall quantitative results across all 25 models. The right panel of Figure 2 shows Pearson correlations between all pairs of models, where each model is represented by its full task accuracy matrix (layer-wise accuracies across all 8 tasks). We compute the correlation between each pair of models' accuracy matrices to measure cross-model consistency. The high correlations indicate that models exhibit similar layer-wise patterns. Given the scale (25 models with varying depths), we think that this correlation heatmap provides the most interpretable summary of consistency across all modern language models. In our revised manuscript, we provide a more thorough analysis and explanation of this heatmap (see Section 4.1, L248-262).
>
> **Concern 4 (steering experiments)**
>
> Thank you for this feedback. We have, in the revision, expanded Section 5.2 to provide more in-depth discussion of the steering experiments, including a more thorough analysis of the results (specifically discussing the U-shaped steering dips observed in Qwen2.5 variants in Figures 20 and 21).

---

> > ### Author Response · Authors · 2025-11-25
> > **Response to Reviewer W1FQ**
> >
> > **Question 1 (why random forests?)**
> >
> > We originally wanted an additional probe with a different capacity that was more robust to noise. However, it added unnecessary complexity to our methodology, and as a result we have decided to remove our Random Forest analysis from the paper. This is in order to streamline our methodology and make space for more interesting analyses.
> >
> > Thank you again for taking the time to review this paper and giving valuable feedback! Hopefully the above discussion and revised manuscript address your concerns. Please let us know if you have any questions or clarifications!
> >
> > **References**
> >
> > [1] [BERT Rediscovers the Classical NLP Pipeline](https://aclanthology.org/P19-1452/) (Tenney et al., ACL 2019)

---

### Official Review · Reviewer_Vgfd · 2025-11-07

**Soundness:** 3
**Presentation:** 3
**Contribution:** 2
**Rating:** 4
**Confidence:** 3

**Summary:**

The paper evaluates a bunch of older LMs as well as modern LLMs on probing tasks used to measure layer-by-layer learning. The findings correlate strongly with "the classical NLP pipeline" (as in Tenney et al): early layers are syntax heavy, middle layers focus more on semantics.. and so on.

The authors follow the probe design methodology from Tenney et al. The setup is pretty similar to Tenney et al, however, they also use the methodology on more SoTA LLMs. In doing so, they compare the learnings b/w more dated LMs (like BERT) v.s modern open source models (like Qwen). They notice some interesting findings, such as, as model sizes increase, learning is peaked at earlier layers indicating that modern models learn rich representations more quickly and need fewer layers to consolidate linguistic knowledge.

In addition, the paper does a thorough analysis on the lexical and morphological inflection tasks, in various settings across layers, multiple languages and probing setups (to check non linear nature)

**Strengths:**

* The experiments are well designed and thorough. The results are explained well and there is strong rationale for most of the results. Overall, seems like a very sound setup. I must also applaud the thoroughness, there is good level of depth to the experiments (so many variables like layers, model sizes, task, probing design)
* The paper is fairly well written. Its easy to understand and keeps you interested.

**Weaknesses:**

* My biggest issue with the paper is the strong similarities to Tenney et al. Its not clear from the paper what is being introduced  in a novel manner v/s ideas derived from Tenney et al. I'm worried that this isn't a significant scientific contribution beyond what previous papers have introduced.
* While this style of probing (for latent knowledge) is interesting, it would be useful to relate this to more "downstream" tasks like math/reasoning/coding/reading comprehension etc. And perhaps we could then compare linguistic probing (like do earlier layers learn hard math problems) v/s downstream task probing? This would be very useful for the community.

**Questions:**

Please answer questions about differences wrt Tenney et al and how this work is contributing in a novel manner.

---

> ### Author Response · Authors · 2025-11-24
> **Response to Reviewer Vgfd**
>
> Thank you for your detailed review! We appreciate that you found the paper well written and easy to understand. We address your concerns below.
>
> **Concern 1 (strong similarity to Tenney et al.)**
>
> First, we note that while our task setup is the same as Tenney et al. (i.e. edge probing), we have clear differences in methodology. They use scalar mixing to combine information across layers while we train linear and non-linear probes at individual layers to *directly isolate* where and how information emerges. Additionally, we analyze 25 modern models (vs their 2) and extend beyond English to 5 other morphologically diverse languages. Finally, to account for probe complexity as a confounding factor, which prior work did not consider, we apply control tasks to measure selectivity [1].
>
> We believe that the main contribution of our work is our novel findings. We find that modern LLMs show compressed linguistic organization, where larger models encode the same information in progressively earlier layers. This compression pattern wasn't visible in BERT/GPT-2.
>
> Additionally, we present analysis on two new tasks not previously considered by Tenney et al. or other works. Specifically, we probe for lemma and inflectional features across layers, revealing how models separate lemma from lexeme. Our steering experiments demonstrate these representations are causally manipulable, and our pretraining checkpoint analysis (see Figures 16 and 17 in the revised paper) provides intuition for when these features emerge during training.
>
> **Concern 2 (downstream tasks)**
>
> The focus of our paper is on understanding how language models encode linguistic information internally, which we believe is valuable independent of downstream performance. Our findings about compressed representations in larger models and the linear/nonlinear split between morphology and lexical identity provide insights into the geometry of learned representations. That said, we think analyzing how linguistic encoding relates to downstream task performance is promising future work!
>
> **References**
>
> [1] [Designing and Interpreting Probes with Control Tasks](https://aclanthology.org/D19-1275/) (Hewitt & Liang, EMNLP-IJCNLP 2019)

---

### Author Response · Authors · 2025-11-25
**General comment to reviewers**

We would like to thank all our reviewers for their detailed and valuable feedback. In response to the shared feedback across reviews about the clarity and organization of our paper, we have uploaded a revised manuscript. We thought it would be helpful to write a general response briefly summarizing the changes made:

- We have updated the captions for Figures 2 and 3 to improve their readability.
- Section 4 has been rewritten to clarify that our pipeline analysis is in terms of relative layer position, not absolute layer position. We have also included a more thorough explanation of how Figures 2 and 3 support our findings.
- We have expanded Section 5.2 (Analysis) to better organize our findings on linear separability, intrinsic dimensionality, and steering.
- We have added explicit definitions for all eight linguistic tasks in Section 2.2 and Appendix D to clarify how they map to the syntax-semantics-discourse pipeline.
- We have expanded Section 6 (Related Work) to include recent studies on representation dynamics in modern LLMs.
- We have removed the Random Forest analysis to streamline the methodology and emphasize the linear vs non-linear complexity classes for our probes.

We also note our appreciation for the positive comments and strengths, including:

- The experiments are well designed and thorough… Overall, seems like a very sound setup. (Reviewer Vgfd)
- I must also applaud the thoroughness, there is good level of depth to the experiments (so many variables like layers, model sizes, task, probing design) (Reviewer Vgfd)
- The work is relevant and timely (Reviewer W1FQ)
- The authors carried out a very large number of experiments on many different LMs of multiple types (MLM, generative [base], generative [instruct] ) and sizes (from BERT base to LLAMA 3.1 8B) (Reviewer W1FQ)
- It is interesting to see how LLMs handle lexical identity and inflectional morphology, which is worth studying (Reviewer EjL9)
- The investigation takes into account large set of models (Reviewer HQt2)
- The set of metrics employed is large and meaningful (Reviewer HQt2)

We have provided our responses to the comments of each reviewer below. We hope our answers resolve all initial questions and concerns raised by the reviewers. If you have any questions or further concerns, we are happy to answer them\!

---

### Note · Authors · 2026-01-06

I have read and agree with the venue's withdrawal policy on behalf of myself and my co-authors.